# A transcriptomic axis predicts state modulation of cortical interneurons

Stéphane Bugeon[1 ✉], Joshua Duffield[1], Mario Dipoppa[1,2], Anne Ritoux[1], Isabelle Prankerd[1], Dimitris Nicoloutsopoulos[1], David Orme[1], Maxwell Shinn[1], Han Peng[3], Hamish Forrest[1], Aiste Viduolyte[1], Charu Bai Reddy[1,4], Yoh Isogai[5], Matteo Carandini[4] & Kenneth D. Harris[1 ✉]

Transcriptomics has revealed that cortical inhibitory neurons exhibit a great diversity of fine molecular subtypes[1–6], but it is not known whether these subtypes have correspondingly diverse patterns of activity in the living brain. Here we show that inhibitory subtypes in primary visual cortex (V1) have diverse correlates with brain state, which are organized by a single factor: position along the main axis of transcriptomic variation. We combined in vivo two-photon calcium imaging of mouse V1 with a transcriptomic method to identify mRNA for 72 selected genes in ex vivo slices. We classified inhibitory neurons imaged in layers 1–3 into a three-level hierarchy of 5 subclasses, 11 types and 35 subtypes using previously defined transcriptomic clusters[3]. Responses to visual stimuli differed significantly only between subclasses, with cells in the *Sncg* subclass uniformly suppressed, and cells in the other subclasses predominantly excited. Modulation by brain state differed at all hierarchical levels but could be largely predicted from the first transcriptomic principal component, which also predicted correlations with simultaneously recorded cells. Inhibitory subtypes that fired more in resting, oscillatory brain states had a smaller fraction of their axonal projections in layer 1, narrower spikes, lower input resistance and weaker adaptation as determined in vitro[7], and expressed more inhibitory cholinergic receptors. Subtypes that fired more during arousal had the opposite properties. Thus, a simple principle may largely explain how diverse inhibitory V1 subtypes shape state-dependent cortical processing.

The cerebral cortex contains a rich diversity of neurons, particularly amongst inhibitory cells. Although this diversity was visible to early anatomists[8,9], its full complexity has emerged only with the advent of transcriptomics[1–6]. Single-cell RNA sequencing (scRNA-seq) and Patch-seq analysis suggest that cortical inhibitory neurons are divided into five major subclasses, which are named *Pvalb*, *Sst*, *Lamp5*, *Vip* and *Sncg* on the basis of their marker genes[1–7]. However, much finer transcriptomic distinctions exist within these subclasses: cluster analysis has defined 60 fine inhibitory transcriptomic subtypes in visual cortex[3]. Moreover, cortical inhibitory neurons exhibit variations along transcriptomic continua[2,10], which can predict their intrinsic physiological properties[2].

A key open question is whether the molecular diversity of cortical inhibitory neurons is mirrored in vivo by diverse patterns of activity, and whether there are simplifying principles that can help understand the relationship between gene expression and activity in these myriad cell groups. Three main methods have been used to characterize the in vivo activity of molecularly identified cells. The first is to record from one cell at a time juxtacellularly, and then apply post-hoc morphological reconstruction and immunohistochemistry[11]. This method has limited throughput. The second is to use electrophysiology or two-photon calcium imaging in transgenic mice[12–20]. However, transgenic lines can only identify one group of cells at a time, and these groups are broad, containing cells of multiple subtypes, types and even subclasses. The third—and potentially most powerful—method is to combine two-photon calcium imaging with ex vivo molecular identification of the recorded neurons[21–27]. This method can record the activity of large numbers of neurons from multiple groups of cells simultaneously, and its ability to assign cells to fine molecular groups is limited only by the methods that are used to subsequently identify the neurons.

Here, we pursued this last approach, using two-photon microscopy to record from large populations of neurons in mouse V1 and applying in situ transcriptomics to the imaged tissue to localize mRNAs for 72 selected genes, a method we term functional neuromics. This is a substantial increase in the number of detected genes compared with previous methods, and allows the identification of fine transcriptomic subtypes. Although most differences in sensory tuning appeared at the level of the five main subclasses, fine subtypes showed significant differences in their modulation by cortical state. These differences in state modulation were explained in large part by a single transcriptomic continuum, which also correlated with the intrinsic membrane properties and morphology of these subtypes as assessed in vitro[7], and with their expression of excitatory and inhibitory cholinergic receptors.

[1]UCL Queen Square Institute of Neurology, University College London, London, UK. [2]Center for Theoretical Neuroscience, Columbia University, New York, NY, USA. [3]Department of Physics, University of Oxford, Oxford, UK. [4]UCL Institute of Ophthalmology, University College London, London, UK. [5]Sainsbury Wellcome Centre for Neural Circuits and Behaviour, University College London, London, UK. ✉e-mail: s.bugeon@ucl.ac.uk; kenneth.harris@ucl.ac.uk

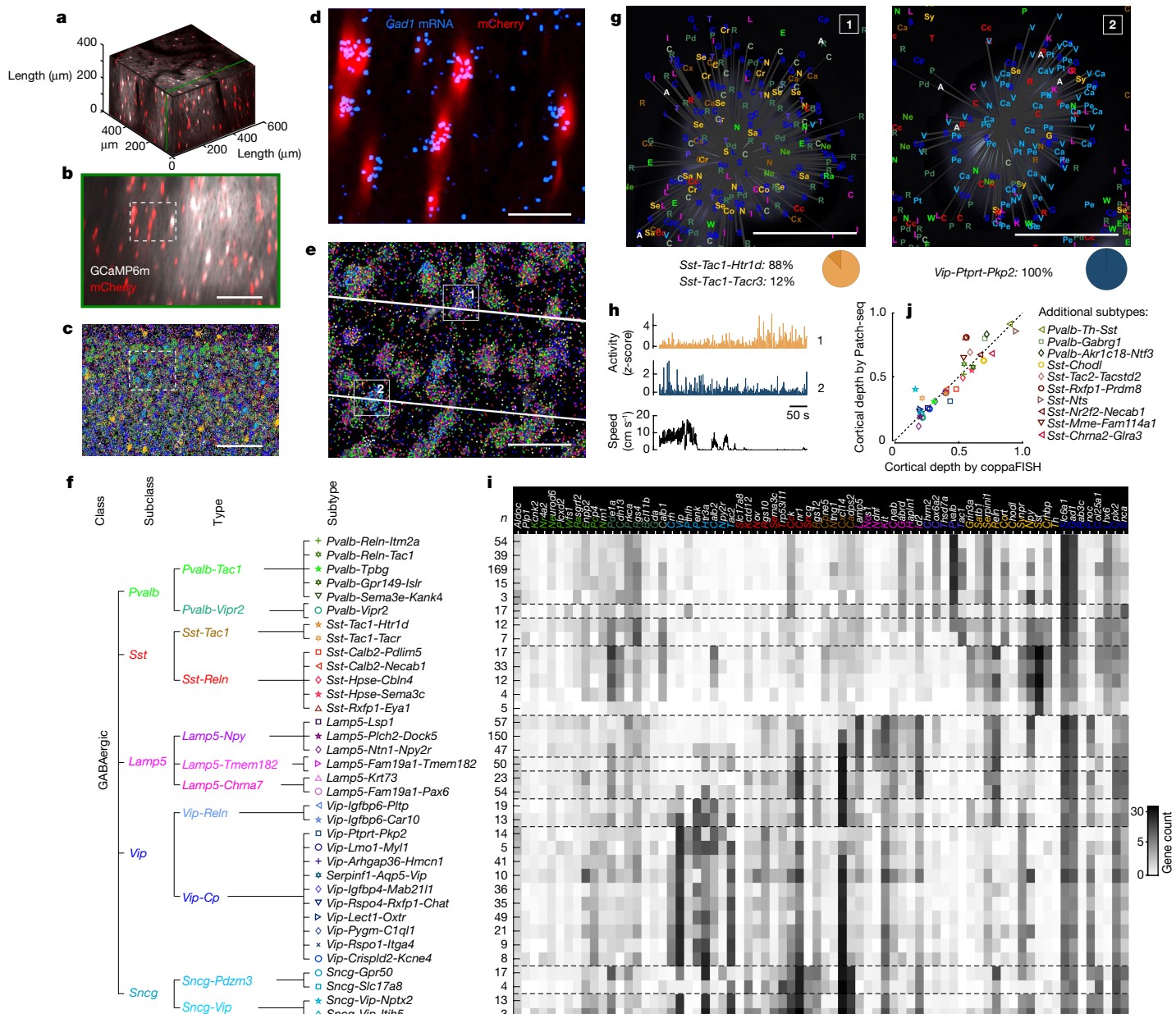

**Fig. 1 | Post-hoc transcriptomic identification of recorded neurons.**
**a**, Three-dimensional (3D) representation of an example reference *z*-stack (white: GCaMP6m, expressed in all neurons, red: mCherry, expressed in inhibitory neurons). **b**, Digital sagittal section of this *z*-stack (maximum intensity projection, 15-μm slice; colours as in **a**. Scale bar, 100 μm. **c**, Portion of ex vivo slice aligned to section in **b** after 72-fold mRNA detection with coppaFISH. Dots represent detected mRNAs (colour code: top of **i**). Scale bar, 100 μm. **d**, Expanded view of dashed rectangle in **b**,**c** showing in vivo mCherry fluorescence (red) and ex vivo *Gad1* mRNA detection (blue). Scale bar, 20 μm. **e**, mRNAs detected in this same region, plotted as in **c**. White lines indicate two functional imaging planes. Grey background: DAPI stain for cell nuclei. Scale bar, 20 μm. **f**, Hierarchical classification of in-vivo-recorded cells into 5 subclasses, 11 types and 35 subtypes. Within each type, subtypes are sorted by their mean first transcriptomic principal component (tPC1) score (see Fig. 5b). *Lect1* is also known as *Cnmd*; *Fam19a1* is also known as *Tafa1*. **g**, Higher-magnification view of cells 1 and 2 from

**e**. Gene detections are indicated by coloured letters (code: top of **i**). Grey background: DAPI image. Below: pie charts indicating probabilities of assignment to subtypes. Scale bars, 5 μm. **h**, Deconvolved calcium traces for the two example cells, together with running speed. **i**, Mean expression of the 72 genes (pseudocoloured as log(1 + gene count)) for the 35 subtypes, ordered as in **f** (*n* = 4 mice). Left: number of unique cells of each subtype. *Nov* is also known as *Ccn3*. **j**, Comparison of the median cortical depth of each subtype found using coppaFISH (as a fraction of total depth; *n* = 14 sections from a brain in which mRNAs were detected down to layer 6), and its median cortical depth found independently using Patch-seq[7] (Pearson correlation: *r* = 0.91, *P* = 1 × 10^−13; analysis of covariance (ANCOVA) controlling for subclasses and types: $F_{(1)} = 163.6$, $P = 6 \times 10^{-12}$). Only subtypes with at least four cells for each dataset were considered. Symbols for subtypes imaged in vivo are shown in **f**; for subtypes too deep to image, symbols are shown on the right.

## Identifying recorded inhibitory subtypes

We performed two-photon calcium imaging in mice expressing mCherry in inhibitory neurons (Gad2-T2a-NLS-mCherry), injected with a pan-neuronal GCaMP6m virus (AAV1-Syn-GCaMP6m-WPRE-SV40), and then applied in situ transcriptomics to sagittal slices of the imaged region. During imaging, mice were free to run on an air-suspended Styrofoam ball, and their behavioural state was monitored through facial videography. Spontaneous activity was recorded in front of a uniform grey screen, and visual responses were elicited by presenting drifting

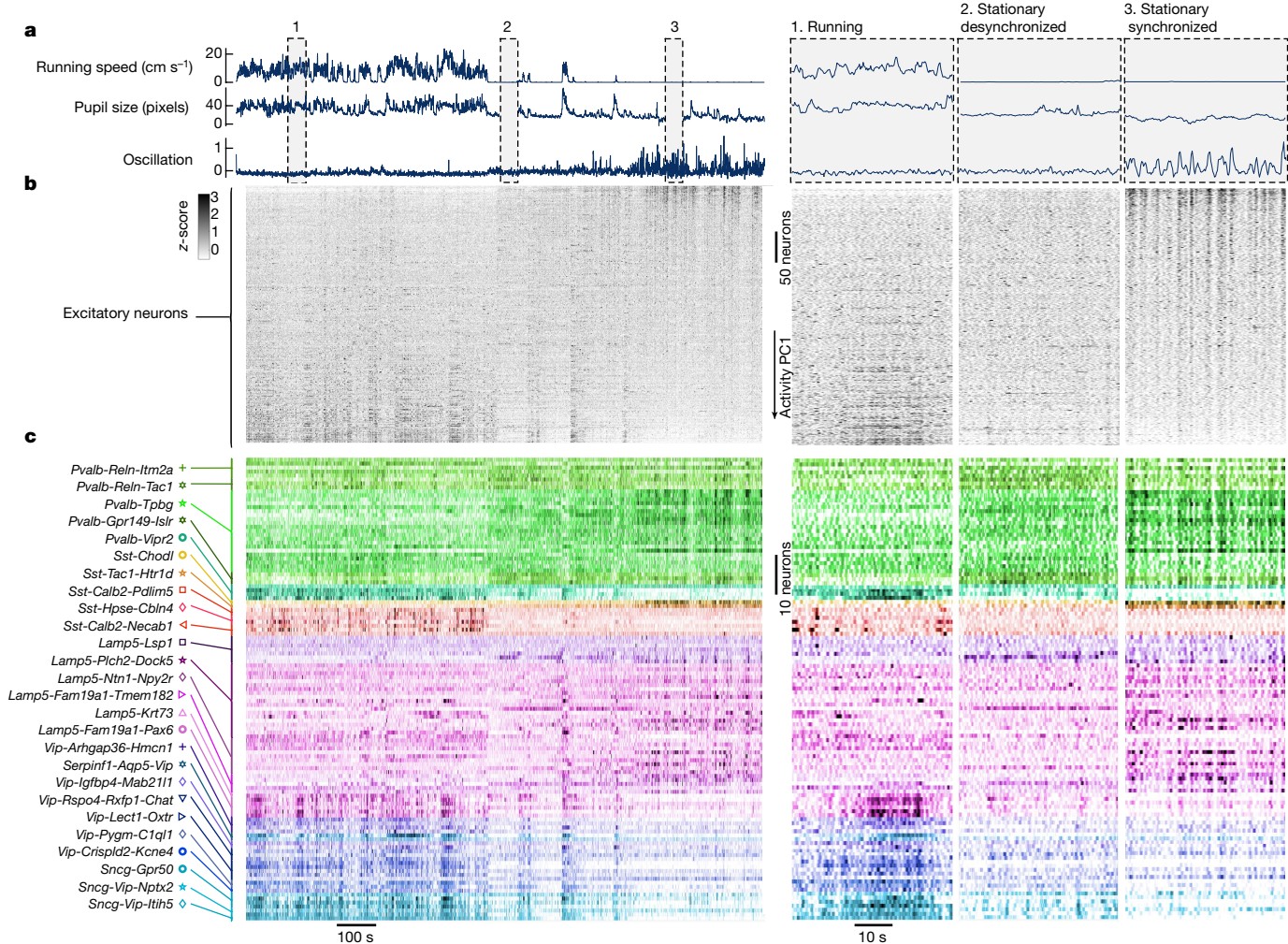

**Fig. 2 | Example raster of spontaneous neuronal activity.** Raster of spontaneous neuronal activity (grey screen) for an example session. **a**, Running speed, pupil size and mean activity of the 10% of excitatory cells (ECs) with most negative principal component weights (oscillation). **b**, Raster of EC activity, sorted by weight on the first principal component of their activity. **c**, Raster of the activity of inhibitory cells (ICs), grouped and coloured by subtype. The three columns on the right show an expanded view of the time windows marked in **a**, illustrating three behavioural states. These rasters show all recorded ECs (413 cells) and molecularly identified ICs (117 cells) in this session (94 ICs not matched to in situ transcriptomics are not shown; note the different scale bars for ECs and ICs). Neuronal activity was z-scored and then smoothed with a 1-s boxcar window.

gratings, natural images and sparse noise. Recordings typically spanned 0–250 μm below the brain surface, targeting cortical layers 1–3. At the end of each session, we obtained a high-resolution two-photon z-stack volume (Fig. 1a). After functional imaging was complete, brains were removed and frozen unfixed, and the imaged volume was cut into 15-μm-thick sections with a cryotome.

To identify the locations of 72 pre-selected genes we built on a previous approach of in situ sequencing[28] to obtain a method termed coppaFISH (combinatorial padlock-probe-amplified fluorescence in situ hybridization). This method amplifies selected transcripts in situ using barcoded padlock probes[29] and reads out their barcodes combinatorially through seven rounds of seven-colour fluorescence imaging (Methods and Extended Data Fig. 1). The method detected $144 \pm 57$ transcripts per cell (mean ± s.d.). The slices were aligned to the in vivo z-stacks with a point cloud registration algorithm using inhibitory neurons as fiducial markers, identified in vivo with mCherry and ex vivo through gene expression (Fig. 1b–e and Extended Data Fig. 2). We applied this method to 17 recording sessions from 4 mice, and obtained $89 \pm 31$ (mean ± s.d.) molecularly identified inhibitory cells together with $393 \pm 173$ pyramidal neurons per session, making a total of 1,090 unique molecularly identified inhibitory cells (some of which were recorded in multiple sessions; Supplementary Data 1).

We classified these inhibitory cells using a three-level hierarchy (Fig. 1f). The lowest hierarchical level ('subtype') comprised the fine transcriptomic clusters defined previously[3], and the top level ('subclass') was the *Pvalb*, *Sst*, *Lamp5*, *Vip* and *Sncg* groupings that were defined in the same previous report. An intermediate level ('type') was suggested by uniform manifold approximation and projection (UMAP) analysis of scRNA-seq data (Extended Data Fig. 3), which revealed collections of clusters that we could putatively associate to morphological cell types (see Methods for full explanation). We named these intermediate-level types *Pvalb-Tac1* (putative *Pvalb* basket cells); *Pvalb-Vipr2* (putative chandelier cells); *Sst-Reln* (putative Martinotti cells); *Sst-Tac1* (putative non-Martinotti *Sst* cells); *Lamp5-Npy* (putative neurogliaform cells); *Lamp5-Tmem182* (putative canopy cells); *Lamp5-Chrna7* (putative layer-1 α7 cells); *Vip-Reln* (putative layer-1 *Vip* cells); *Vip-Cp* (other Vip cells); *Sncg-Pdzrn3* (putative large *Cck* cells); and *Sncg-Vip* (putative small *Cck/Vip* cells). UMAP analysis (Extended Data Fig. 3) suggested that although types were usually discrete, their constituent subtypes often merged continuously into each other, tiling dimensions of continuous variability of inhibitory neurons.

Cells that were functionally imaged in vivo were assigned to a subtype (and thus also a type and subclass) using pciSeq[28], a Bayesian algorithm that computes for each cell a probability distribution over clusters defined

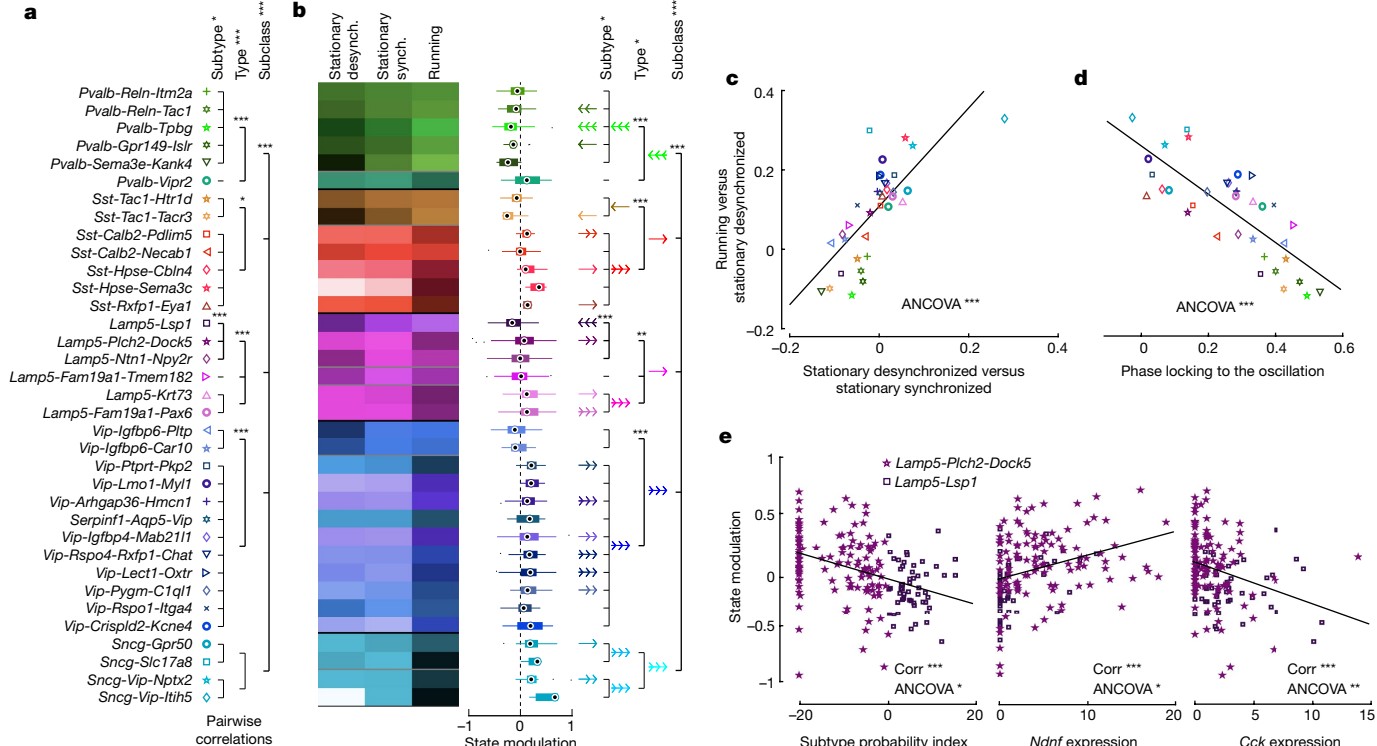

**Fig. 3 | State modulation of inhibitory subtypes. a**, Nested permutation analysis for spontaneous correlations. Top, significance of omnibus test for higher correlations within subclasses ($P < 0.0001$), nested types ($P < 0.0001$) and subtypes ($P = 0.048$). Right bolded brackets: significant post-hoc tests within each grouping (Benjamini–Hochberg-corrected; $P$ values in Extended Data Fig. 6b). **b**, Left, pseudocolour representation of the mean activity of each subtype in each state. Middle, box plots showing the distributions of state modulation (running versus stationary synchronized) for cells of each subtype ($n = 4$ mice, 17 sessions; for box definitions, see Methods). Right, nested permutation analysis, plotted as in **a**. Omnibus test found significantly different state modulation between subclasses ($P < 0.0001$), nested types ($P = 0.022$), and subtypes ($P = 0.014$). Benjamini–Hochberg-corrected post-hoc tests found significant differences within *Pvalb* ($P < 0.0001$), *Sst* ($P < 0.0001$), *Lamp5* ($P = 0.0025$), *Vip* ($P < 0.0001$) and *Lamp5-Npy* ($P < 0.0001$) cell groups. Coloured arrows at each level indicate significant state modulation for each cell group (two-sided *t*-tests, Benjamini–Hochberg-corrected; number of arrowheads indicates significance). **c**, Modulation for running versus stationary desynchronized states, against modulation for stationary synchronized versus desynchronized states. Each glyph shows mean values for a subtype; symbols as in **b** ($F(1) = 375.4$, $P = 3.6 \times 10^{-71}$, ANCOVA controlling for session). **d**, Modulation for running versus stationary desynchronized states against locking to the synchronized state oscillation. Each glyph shows mean values for a subtype ($F(1) = 240.5$, $P = 2 \times 10^{-47}$, ANCOVA controlling for session). **e**, State modulation for cells in the *Lamp5-Plch2-Dock5* and *Lamp-Lsp1* subtypes, against subtype probability index ($\log(P_{\text{Subtype1}}/P_{\text{Subtype2}})$; left), or *Ndnf* (middle) and *Cck* (right) gene expression. These three variables correlated significantly with state modulation (Pearson correlation: subtype probability: $P = 2 \times 10^{-7}$, $r = -0.39$; *Ndnf* expression: $P = 2 \times 10^{-5}$, $r = 0.32$; *Cck* expression: $P = 2 \times 10^{-4}$, $r = -0.29$), even controlling for a common effect of subtype ($F(1) = 4.8$, $P = 0.03$; $F(1) = 6.2$, $P = 0.014$; $F(1) = 10.7$, $P = 0.0014$, respectively, ANCOVA). Black lines: linear fits. *$P < 0.05$, **$P < 0.01$, ***$P < 0.001$; one-, two- or three-headed arrows in **c** indicate the same significance levels; direction indicates the sign of modulation.

by previous scRNA-seq data. Expression levels were sufficient to assign cells with high probability to a single subtype (Fig. 1g,h and Extended Data Fig. 4), and we therefore assigned each cell to a single subtype of maximum a posteriori probability. The algorithm could assign cells to any of the 109 clusters defined by scRNA-seq (representing all neurons and non-neurons), but as expected from the restriction of two-photon imaging to the superficial layers, the imaged cells were assigned to just 35 clusters corresponding to superficial inhibitory neurons. The number of cells recorded varied across transcriptomic groups (Fig. 1i), and subtypes to which fewer than three cells were assigned were excluded from further analysis (eight cells total). The gene expression for the 72 genes in our panel showed consistent differences across the 35 subtypes recorded (Fig. 1i).

To verify the accuracy of our cell-type assignments, we performed two analyses using independent data. First, we took advantage of the fact that different inhibitory subtypes reside at different cortical depths, as established by an independent Patch-seq study[7]. The median depth of subtypes by our method and by Patch-seq matched closely (Fig. 1j). Notably, this did not only reflect depth differences between the top-level subclasses ($P < 0.001$, ANCOVA controlling for subclass) or even types ($P < 0.001$, ANCOVA controlling for type). For example,

whereas *Sst*-expressing neurons are most often found in deep layers, specific subtypes such as *Sst-Calb2-Necab1* were localized in superficial layers by both our method and the independent Patch-seq data. Second, we compared the functional recordings to two-photon calcium imaging studies that identified cells with three transgenic lines (*Sst*, *Pvalb* and *Vip*)[30]. When we analysed our data after grouping together cells that were expected to be labelled in each of these lines, we found results that were consistent with previous studies (Extended Data Fig. 5). We thus conclude that our methods accurately identify subtypes.

## State modulation of inhibitory subtypes

In vivo activity differed between transcriptomic groups, down to the subtype level. We generated raster plots showing the simultaneous spontaneous activity of V1 populations, with all inhibitory neurons identified to subtypes (Fig. 2). Examination of these rasters revealed complex patterns of correlated activity. We tested whether correlations between cells within a single subclass, type or subtype were stronger than correlations across these groupings (Fig. 3a). This analysis requires careful statistics owing to two potential confounds: first, the large

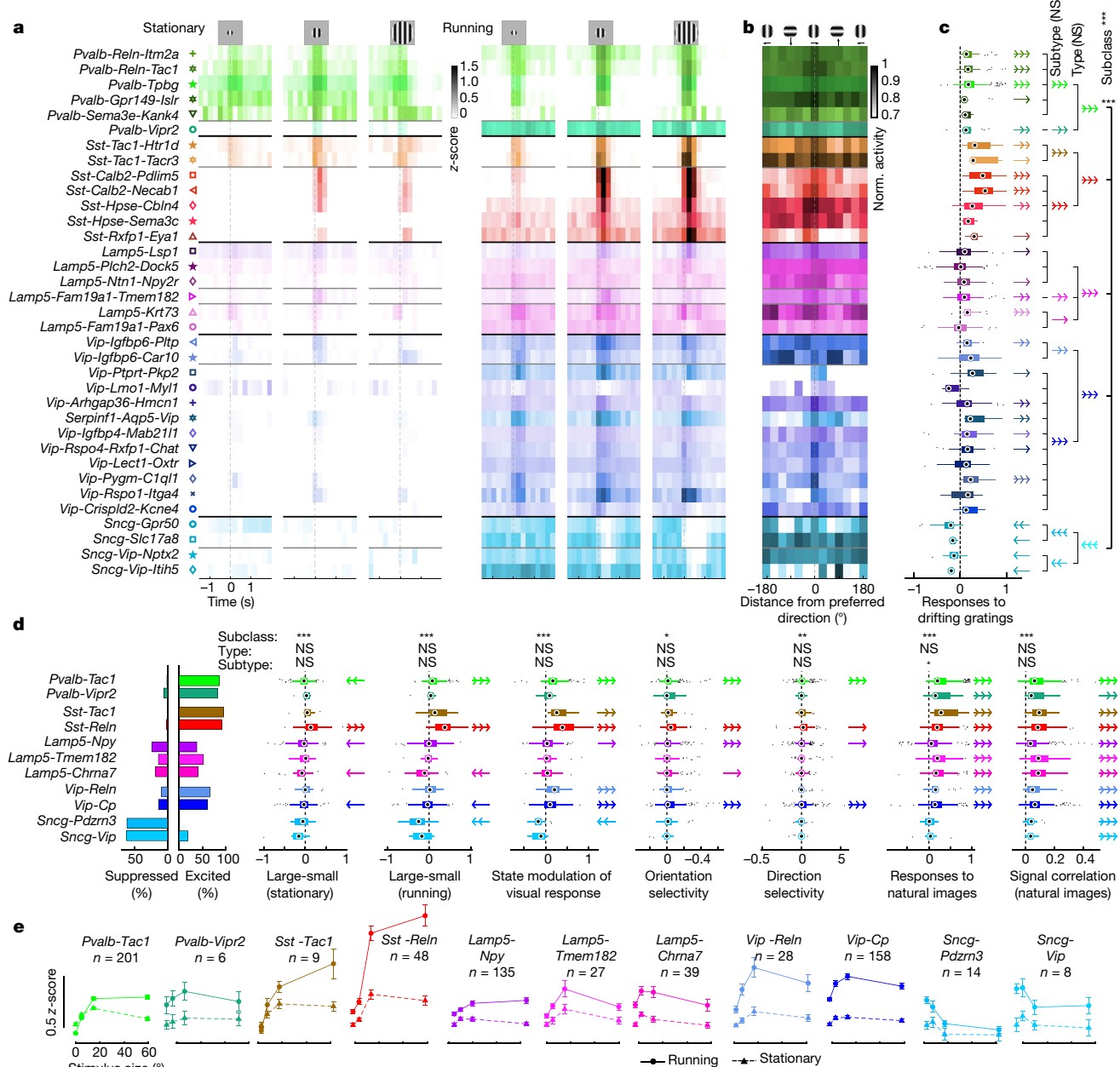

**Fig. 4 | Sensory responses of inhibitory subtypes. a**, Pseudocolour activity rasters trial-averaged on the onset of drifting grating stimuli (duration 0.5 s), for different stimulus sizes (5°, 15° and 60°) and locomotor states. Each row shows the average activity of a subtype. Dashed grey lines: stimulus onset. **b**, Cross-validated direction tuning curves for each subtype, shown in pseudocolour as a function of grating direction. Tuning curves were averaged over odd trials, and shifted relative to the preferred direction found on even trials; thus, a peak will only appear at 0 if the cell is genuinely tuned. **c**, Nested permutation analysis of drifting grating responses (measured at the stimulus size eliciting the largest negative or positive response), plotted as in Fig. 3b. Top, significance of omnibus test for differences between subclasses (*P* < 0.0001) and nested types (*P* = 0.99) and subtypes (*P* = 0.49). **d**, Additional analyses of stimulus responses. From left: fraction of cells of each type

significantly excited or suppressed by grating stimuli; hierarchical analyses of response differences between large and small gratings in stationary and running conditions; state modulation of visual response by running, averaged over all sizes; orientation and direction selectivity; and mean response and reliability (signal correlation) for natural image stimuli. Nested permutation analyses plotted as in Fig. 3b but only to type level; full plots and *P* values are in Extended Data Fig. 7. **e**, Mean size tuning curves for each type, showing the mean responses in stationary (dashed lines, triangles) and running (solid lines, circles) epochs. Only cells with receptive fields < 20° from the stimulus centre are included. Error bars, s.e.m. (*n* = 4 mice, 17 sessions; numbers below the type name on each plot indicate the number of cells). *P* < 0.05, **P* < 0.01, ***P* < 0.001; NS, not significant; one-, two- or three-headed arrows in **c**,**d** indicate the same significance levels; direction indicates the sign of modulation.

number of subtypes presents a multiple comparisons problem; second, different recordings will by chance sample different proportions of each cell group, so variability between recordings could be mistaken for variability between cell groups. To solve these confounds, we developed a nested permutation test, which tests the null hypothesis that

activity amongst cells assigned to the same transcriptomic group is no more similar than that amongst cells assigned to different transcriptomic groups (Methods and Extended Data Fig. 6a). The test operates hierarchically, testing for significant differences between subclasses, between the types that comprise a single subclass, and between the

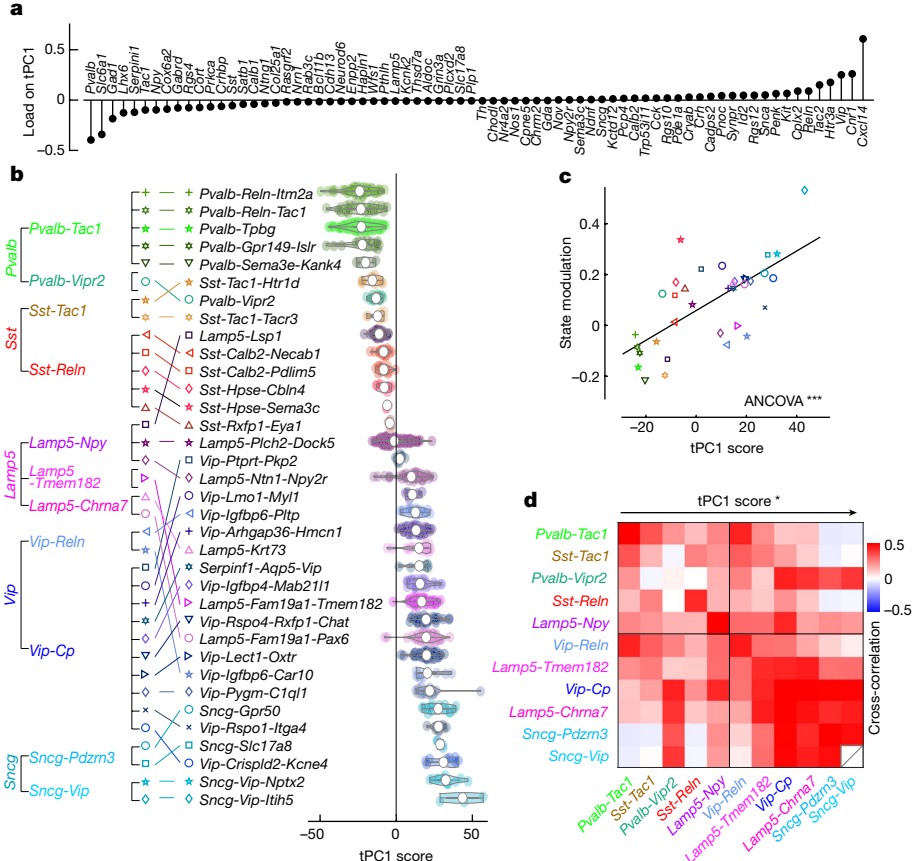

**Fig. 5 | A single transcriptomic axis predicts state modulation. a**, Loading of each gene onto tPC1. **b**, Ordering of subtypes by tPC1. Left, original ordering by subclass and type as in previous figures. Middle, subtypes re-ordered by the mean of tPC1. Right, violin plots showing the distribution of tPC1 values over cells of each subtype. **c**, Correlation between state modulation and tPC1.

Each glyph represents mean values for a subtype; symbols as in **b** ($F(1) = 14.5$, $P = 2 \times 10^{-4}$, ANCOVA controlling for session and subclass). **d**, Matrix of pairwise correlations between simultaneously recorded types. The types are sorted by tPC1, showing a significant effect of tPC1 on the pairwise correlations ($P = 0.014$; one-sided permutation test). *$P < 0.05$, ***$P < 0.001$.

subtypes that comprise a single type. The test showed that correlations between cells within a single subclass, type or subtype were stronger than correlations across these groupings ($P < 0.001$ for subclass and type; $P < 0.05$ for subtype; Fig. 3a and Extended Data Fig. 6a,b).

The activity of different cell groups correlated diversely with ongoing behaviour as measured by two assays of arousal: locomotion and pupil diameter (Fig. 2a,c). Because these assays of arousal in turn correlate with cortical state[31–33], we asked how the activity of the identified cell groups depends on cortical state.

We characterized cortical state using the activity of the excitatory population. As previously described[34], some excitatory cells (positively weighted on the first principal component of population activity) were more active when the mouse was aroused (fast running, large pupil), whereas other excitatory cells (with negative weights) fired during inactive periods (no running, small pupil). In addition, we found that behavioural inactivity was sometimes accompanied by low-frequency oscillations in population activity, which strongly synchronized the excitatory neurons as visible in the mean activity of the negatively weighted cells (Fig. 2a,b; the frequency of these fluctuations is unclear as our two-photon microscope aliases frequencies above 2.15 Hz). We thus used running and cortical synchronization to distinguish three states that correspond to decreasing levels of arousal: running; stationary desynchronized; and stationary synchronized. To quantify the modulation of a cell by cortical state we compared the activity of each cell during the two extreme states: running versus stationary synchronized.

We found significant differences in the way that different subclasses, types and subtypes were modulated by cortical state (Fig. 3b). We modified the nested permutation test to compare activity between states,

and found significant differences between subclasses ($P < 0.001$), with the *Sncg*, *Vip* and *Lamp5* subclasses being on average more active during running and the *Pvalb* subclass more active during oscillation. Significant differences were also seen between the types that constitute individual subclasses ($P = 0.02$). For example, within the *Pvalb* subclass, *Pvalb-Tac1* cells were strongly active during synchronized states and less active during running, whereas *Pvalb-Vipr2* cells showed the opposite behaviour (consistent with previous results[35]). Within the *Sst* subclass, *Sst-Tac1* cells were most active during synchronized states, whereas *Sst-Reln* cells were more active during running. Similar dichotomies were observed in the *Lamp5* subclass, with *Lamp5-Chrna7* cells being more active in running and *Lamp5-Npy* cells mixed. *Vip* and *Sncg* cells were more active during running—except for *Vip-Reln* cells, which showed the opposite behaviour. Significant differences were also seen between the subtypes that comprise a single type ($P = 0.014$). The most prominent of these differences was between the subtypes that comprise the *Lamp5-Npy* (putative neurogliaform) type, with *Lamp5-Plch2-Dock5* cells firing more in running but *Lamp5-Lsp1* cells firing more in synchronized states. A trend toward differences in state modulation was also seen between *Sst-Reln* (putative Martinotti) subtypes ($P < 0.05$; significant on its own but not after Benjamini–Hochberg correction). Activity in the stationary desynchronized state was intermediate between the synchronized and the running states (Fig. 3c and Extended Data Fig. 6c,d). A subtype's state modulation was correlated with its degree of phase-locking to the synchronized state oscillation, with subtypes that were more active during running being less locked to the oscillation during the stationary synchronized periods (Fig. 3d).

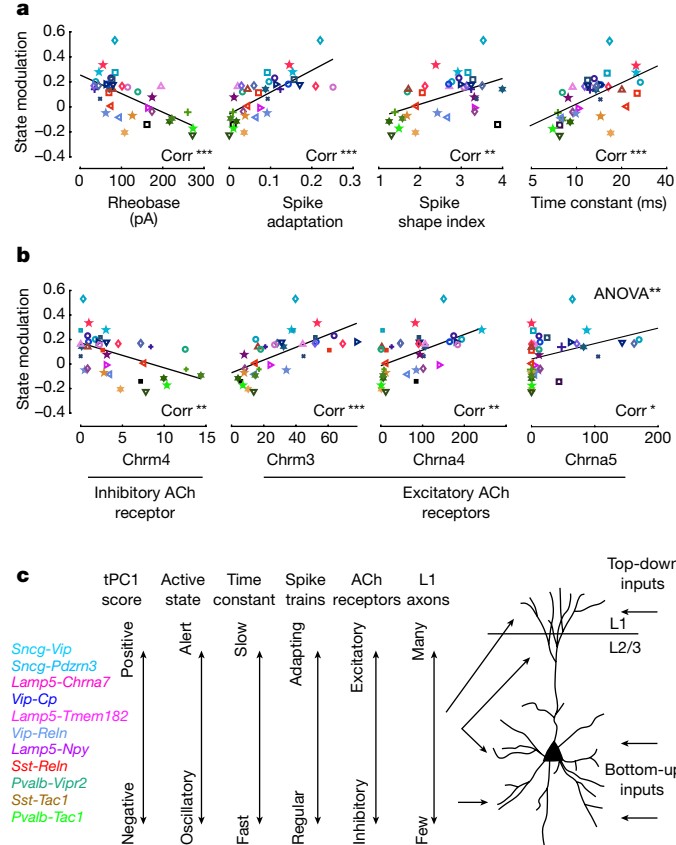

**Fig. 6 | Correlation of state modulation with cellular properties.**
**a**, Correlation between state modulation and electrophysiological properties measured by an independent Patch-seq study[7]. Each symbol represents mean values for a subtype, coded as in Fig. 5b. Rheobase: $r = -0.63$, $P = 5 \times 10^{-5}$; spike adaptation: $r = 0.70$, $P = 3 \times 10^{-6}$; spike shape index: $r = 0.49$, $P = 0.003$; time constant: $r = 0.57$, $P = 4 \times 10^{-4}$ (significance, Pearson correlation). **b**, Correlation between state modulation and cholinergic receptor expression obtained from an independent scRNA-seq study[3]. Each symbol represents mean values for a given subtype, coded as before. $Chrm4$: $r = -0.50$, $P = 0.002$; $Chrm3$: $r = 0.63$, $P = 5 \times 10^{-5}$; $Chrna4$: $r = 0.52$, $P = 0.0014$; $Chrna5$: $r = 0.37$, $P = 0.03$. Correlations of state modulation with excitatory cholinergic receptor expression were higher than with inhibitory receptor expression (including receptors not shown here; $P = 0.008$, $F(1) = 12.2$, two-sided ANOVA; only receptors with more than 2 counts in at least 5 subtypes were considered, making 10 in total). *$P < 0.05$, **$P < 0.01$, ***$P < 0.001$; black lines are linear regression fits. **c**, Schematic summarizing the transcriptomic axis and its functional and cellular correlates. Right, schematic of inputs from inhibitory neurons along the transcriptomic axis to a layer-2/3 (L2/3) cortical excitatory cell. ACh, acetylcholine.

The modulation of individual subtypes by brain state varied smoothly along transcriptomic continua, rather than showing sharp differences between discrete groupings. For example, amongst subtypes of the *Lamp5-Npy* type, *Lamp5-Lsp1* cells were most active in the synchronized state, whereas *Lamp5-Plch2-Dock5* cells fired more during running. The division between these two subtypes, however, reflects a somewhat arbitrary dividing line along a continuous dimension of transcriptomic variability (Extended Data Fig. 3). To test whether such a continuous dimension of transcriptomic variability could explain differences in state modulation, we quantified the position of each imaged cell along the continuum by its ratio of posterior probabilities of assignment to the two subtypes. We observed a smooth dependence of state modulation along this continuum, which ANCOVA analysis showed depended more on this continuous transcriptomic score than on discrete subtype assignment (Fig. 3e). Similar continuous dependence was visible at

the single-gene level, with state modulation within *Lamp5-Npy* cells correlating with expression of *Cck* and *Ndnf* (Fig. 3e) even after controlling for subtype. Similar results were seen for *Sst-Reln* subtypes (Extended Data Fig. 6e).

## Sensory responses of inhibitory subtypes

We next probed responses to visual stimuli: drifting gratings of various sizes and directions, and natural images. Unlike state modulation, visual responses showed significant differences only at the level of subclasses, not types or subtypes.

Most inhibitory subtypes contained neurons that responded to grating stimuli (Fig. 4a–d and Extended Data Fig. 7). *Pvalb* and *Sst* cells that responded to gratings were almost exclusively excited by them. *Lamp5* and *Vip* cells contained a mixture of excited and inhibited cells, with *Vip* cells more often being excited. Notably, *Sncg* cells—whose visual responses have not to our knowledge yet been studied—were almost exclusively inhibited. Orientation and direction selectivity were relatively low for most subclasses[17,22], except for a slight tendency for *Sst* and *Vip* cells to show stronger tuning. Most cells showed significant coding of natural image stimuli, which differed significantly between subclasses, and was weakest for *Sncg* cells. Differences in natural image responses were largely homogeneous between types and subtypes within a subclass, although a trend towards difference was seen among the *Lamp5* subclass (Extended Data Fig. 7).

The most marked difference in the visual responses of different inhibitory cell groups was in their tuning for grating size and the modulation of this tuning by cortical state (Fig. 4a,d,e). Size tuning was significantly modulated at the subclass level: whereas *Sst* cells showed little or no surround suppression, with strong responses to large stimuli[12,30], *Sncg* cells showed a clear opposite pattern, in which they were progressively more suppressed by larger stimuli (Fig. 4e). Modulation of grating response by locomotion was significantly different between subclasses, with locomotion increasing the responses of *Sst*, *Pvalb* and *Vip* cells to various degrees, and decreasing those of *Sncg* cells.

In summary, sensory responses showed significant differences between subclasses, but not between types and subtypes. The most marked differences between subclasses were in size tuning and its modulation by state. A lack of statistical significance of course does not exclude the possibility that the sensory tuning of subtypes may differ in ways too small for our methods to detect; but the fact that the same statistical tests found subtype differences in state modulation suggests that any such differences in sensory responses are likely to be subtle.

## Transcriptomic PC1 and state modulation

The above analyses showed that state modulation, but not visual responses, differ significantly between transcriptomic subtypes. We next returned to the dependence of state modulation on subtype, and found that a portion of this diversity can be explained by a single transcriptomic axis (Fig. 5). This axis was defined independently of the physiological data: we simply computed the first principal component of the gene expression vectors measured in situ (transcriptomic principal component 1, or tPC1). Applying principal component analysis (PCA) to the in situ transcriptome of our cells revealed a continuum (Fig. 5a,b). This continuum did not simply reflect an ordering of the five main subclasses, but a more complex organization of types and subtypes. For example, although *Pvalb-Tac1* (putative basket) cells occupied the negative end of the continuum, *Pvalb-Vipr2* (putative chandelier) cells were positioned amongst a different type—the *Sst* cells. The different *Lamp5* subtypes were widely distributed across the continuum, with *Lamp5-Npy* (putative neurogliaform) cells having more negative values than *Lamp5-Tmem182* and *Lamp5-Chrna7* (putative canopy and α7) cells, which were positioned amongst cells of a different subclass—the *Vip* cells. *Sncg* cells occupied the most positive

end of the continuum. The loading of cell-type marker genes (such as *Pvalb* or *Vip*) on tPC1 reflected the position of the corresponding cell types, but genes that are expressed in all interneurons could also show strong loadings; for example, *Gad1* and *Slc6a1* (Fig. 5a), which are involved in the synthesis and transport of GABA (γ-aminobutyric acid), were strongly negatively loaded. The ordering and gene loading observed here are similar to those seen in a previous analysis of CA1 single-cell transcriptomic data[10], which suggested that cell types at the negative end of this continuum express genes consistent with faster metabolic rates and strong inhibition on the somas or proximal neurites of their targets.

The state modulation of a subtype correlated with its position along the transcriptomic continuum tPC1 (Fig. 5c). Cells with negative tPC1 scores, such as *Pvalb-Tpbg* (putative basket) cells, were most strongly active in synchronized states, whereas cells with a positive tPC1 value, such as *Sncg* cells, were most active during desynchronized and running states. State modulation was significantly correlated with tPC1, even after taking into account differences between subclasses ($P < 0.001$, ANCOVA controlling for session and subclass; Fig. 5c). These effects could be seen at a single-gene level, with a subtype's state modulation negatively correlated with expression of the GABA-processing genes *Gad1* and *Slc6a1* (Extended Data Fig. 8a; $P < 0.001$, ANCOVA controlling for session). This single principal component could predict 70% of the variance of state modulation that is explainable transcriptomically (Extended Data Fig. 8b). Thus, different inhibitory subtypes have diverse relationships to cortical state, but these relationships can be predicted in large part by a single transcriptomic axis, with the side of this axis that is associated with stronger GABA synthesis showing more activity in oscillatory states.

The tPC1 axis also largely predicted correlations between the spontaneous activity of inhibitory types, with positive correlations between types of similar tPC1 values, and negative correlations between types of opposite tPC1 values ($P < 0.05$, permutation test; Fig. 5d). This also held true when considering correlations computed within any of the three states independently (Extended Data Fig. 8c–e).

A cell type's state modulation and position on the tPC1 axis also correlated with many aspects of its intrinsic physiology and morphology (Fig. 6a and Extended Data Fig. 9). To demonstrate this, we compared our measurement of each subtype's state dependence with measurements on the same V1 subtypes that were made by an independent study using Patch-seq[7]. This comparison showed that subtypes that were active during synchronized states (low arousal levels) had faster membrane time constants and spike repolarization speeds, a more hyperpolarized resting potential, lower membrane resistance, a larger rheobase (the minimum current required to drive spiking) and weaker spike frequency adaptation (Fig. 6a and Extended Data Fig. 9a). Subtypes active during running had the opposite properties. For example, *Sst-Tac1* cells, which are faster spiking than *Sst-Reln* cells[7], had the lowest tPC1 values and the greatest preference for oscillatory states amongst the *Sst* subclass (Fig. 5b). This Patch-seq data also revealed a noteworthy correlation of tPC1 and axonal morphology. Within the *Sst* and *Lamp5* subclasses, cells with larger values of tPC1 (which would thus show more activity in alert states in vivo) had a greater fraction of their axonal projections in layer 1, and a smaller fraction in layer 2/3 ($P < 0.001$ and $P < 0.05$ for *Lamp5* and *Sst* respectively; Pearson correlation with Benjamini–Hochberg correction; Extended Data Fig. 9b). This correlation was not seen for the other subclasses, for which axonal projections to layer 1 were rare.

Finally, we asked whether state modulation also correlated with the expression of cholinergic receptors between subtypes. Levels of acetylcholine are higher in active states and contribute to cortical desynchronization[36–38]. Moreover, acetylcholine differentially affects inhibitory neuronal types by acting through different receptors, with nicotinic and $G_q$-coupled muscarinic receptors exciting some inhibitory types, and $G_i$-coupled muscarinic receptors inhibiting others[39–44]. We compared our measurements of each subtype's state dependence with cholinergic receptor expression measured in an independent single-cell transcriptomic study[3], and found positive correlations between state modulation and the expression levels of all nicotinic or $G_q$-coupled muscarinic receptors, and negative correlations between state modulation and the expression levels of $G_i$-coupled muscarinic receptors (Fig. 6b; excitatory receptors significantly more positively correlated than inhibitory receptors; $P < 0.01$, ANOVA). We thus hypothesize that differential expression of cholinergic receptor subtypes might contribute to the continuum of state modulation along the main axis of transcriptomic variation tPC1.

## Discussion

By identifying the transcriptomic types of simultaneously recorded V1 neurons, we discovered functional differences across fine cellular subtypes, ordered along a main axis of transcriptomic variation. These subtype differences were seen not in the sensory responses of the neurons—which differed primarily across high-level subclasses—but rather, in the relation of their activity to cortical and behavioural state. State modulation can vary significantly between fine subtypes within a type, but this appears to reflect continuous transcriptomic variation rather than discrete subtypes. Furthermore, a single axis of transcriptomic variation across inhibitory cells—the first transcriptomic principal component (tPC1)—largely explains the differences in state modulation between subtypes, and predicts their spontaneous correlations. This transcriptomic axis also correlates with a subtype's membrane physiology, layer-1 axon content and expression of excitatory and inhibitory cholinergic receptors (Fig. 6c).

It is notable that a single transcriptomic dimension—derived from patterns of gene expression without regard to functional or physiological properties—correlates with state modulation that we measured in vivo, with intrinsic physiology measured in vitro[7] and with the expression of cholinergic receptors with opposite signs for excitatory and inhibitory receptors[3]. This dimension defined in V1 a continuum that is similar to one previously described in CA1 inhibitory neurons[10], but with one exception: in CA1, *Sncg* subtypes were spread along the continuum, rather than being clustered at the positive end as in V1. This might be related to the existence of fast-spiking CCK basket cell subtypes in CA1 (ref. [45]), and to the fact that CA1 *Sncg* cells can be inhibited by locomotion[46].

The correlation between tPC1, state modulation and cellular physiology is not perfect, and this one axis certainly cannot explain all properties of cortical interneurons. Nevertheless, tPC1 may define an approximate but general organizing principle, that can explain many observations that have previously been made on individual inhibitory groups (Supplementary Discussion). For example, acetylcholine has been shown to have diverse effects on different inhibitory groups[39–43], such as the classical 'cholinergic switch'[44], in which fast-spiking (putative *Pvalb* basket) cortical neurons are inhibited by muscarinic receptors but low-threshold spiking (putative *Sst* Martinotti) neurons are excited by nicotinic receptors. This result is consistent with the receptor expression profile of these types, and with our finding that desynchronized and running states suppress *Pvalb-Tac1* cells and drive *Sst-Reln* cells. In fact, our data suggest that the behaviour of these two types reflects a more general principle: at least in superficial V1, inhibitory cells with lower tPC1 values exhibit physiological properties that are closer to *Pvalb* basket cells, lower levels of nicotinic and excitatory muscarinic receptors, more inhibitory muscarinic receptors and negative state modulation. The reverse is true for cells with larger tPC1 values.

The computational role of this state-dependent switch in the activity of different inhibitory cell types remains an open question. However, our data are consistent with a long-standing hypothesis that alert states and cholinergic modulation biases cortex towards feedforward inputs from primary thalamus, and away from top-down inputs from

elsewhere in cortex[47,48] (Fig. 6c). Indeed, the types that are most suppressed by alert states (putative *Pvalb* basket and *Sst* non-Martinotti) preferentially target thalamorecipient layers 4 and 5b, whereas the *Sncg*, *Lamp5*, *Sst* Martinotti and *Vip* cells, which are more excited in alert states, preferentially target either interneurons, or pyramidal cells in other layers[49,50]. Our data furthermore suggest that the degree of state modulation for *Sst* and *Lamp5* neurons correlates with their axonal innervation of layer 1, which receives top-down input. Opposing cholinergic modulation of these inhibitory types might thus alter the balance between bottom-up and top-down inputs.

In summary, we introduced a functional neuromics approach that revealed that the sensory tuning of V1 inhibitory neurons is determined largely by their top-level transcriptomic subclass, and that their state modulation can be predicted to good approximation from a single transcriptomic axis that also correlates with their intrinsic physiology, morphology and cholinergic receptor expression. As emerging experimental techniques allow for ever-greater amounts of information to be collected on the physiology, connectivity and firing correlates of cortical interneuron types, these simple principles may help to organize this knowledge.

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

# Methods

All experimental procedures were conducted in accordance with the UK Animals (Scientific Procedures Act) 1986. Experiments were performed at University College London under personal and project licences released by the Home Office following appropriate ethics review.

## Mice

Experiments were performed on mice aged between 12 and 15 weeks maintained on a 12-h light–dark cycle, at 20–24 °C and 45–65% humidity, in individually ventilated cages. For post-hoc identification of transcriptomic subtypes, four (two males and two females) Gad2-T2a-NLS-mCherry transgenic mice (stock no: 023140, The Jackson Laboratory), expressing the red fluorescent protein mCherry in the nuclei of *Gad2*-expressing cells, were used. For comparison to transgenic mouse lines (Extended Data Fig. 5), additional experiments were performed as in ref. [30] using one male *Pvalb*[tm1(cre)Arbr] and two males and one female *Sst*[tm2.1(cre)Zjh] crossed with *Gt(ROSA)26Sor*[tm14(CAG-tdTomato)Hze].

## Surgical procedures

On the day of surgery, mice were anaesthetized with isoflurane (1–2% in oxygen), their body temperature was monitored and kept at 37–38 °C using a closed-loop heating pad, and the eyes were protected with ophthalmic gel (Viscotears Liquid Gel, Alcon). An analgesic (Rimadyl, 5 mg kg⁻¹) was administered subcutaneously before the procedure, and orally on subsequent days. Dexamethasone (0.5 mg kg⁻¹) was administered intramuscularly 30 min before the procedure to prevent brain oedema. The exposed brain was constantly perfused with artificial cerebrospinal fluid (150 mM NaCl, 2.5 mM KCl, 10 mM HEPES, 2 mM CaCl$_2$, 1 mM MgCl$_2$; pH 7.3 adjusted with NaOH, 300 mOsm). During the surgery, we first implanted a head plate over the right hemisphere of the cranium for later head-fixation: a stainless-steel head plate with a 10-mm circular opening was secured over the skull using dental cement (Super-Bond C&B, 10 Sun Medical). We then made a circular craniotomy over V1 (3 mm diameter) using a biopsy punch. At this point, six to seven virus injections were made at different positions inside the craniotomy. Finally, the craniotomy was sealed with a glass cranial window, using cyanoacrylate adhesive (Vetbond, 3M) and dental cement.

All mice were injected with an unconditional GCaMP6m virus, AAV1. Syn.GCaMP6m.WPRE.SV40 obtained from the University of Pennsylvania Viral Vector Core. The virus was injected with a bevelled micropipette using a Nanoject II injector (Drummond Scientific Company, Broomall, PA 1) attached to a stereotaxic micromanipulator. Six to seven boli of 100–200 nl virus (2.23 × 10$^{12}$ GC ml⁻¹) were slowly (around 20 nl min⁻¹) injected unilaterally into monocular V1 (ref. [51]) 2.1–3.3 mm laterally and 3.5–4.0 mm posteriorly from bregma and at a depth of L2/3 (200–300 mm).

After virus injection, a small bolus (10 µl) of red fluorescent beads (FluoSpheres Carboxylate-Modified Microspheres, 2.0 µm, red fluorescent (580/605), 2% solids, Thermo Fisher Scientific) was injected at the most rostral part of the craniotomy, to allow orientation of the ex vivo slices but not interfere with V1 imaging in the caudal part. After recovery, mice were habituated for handling and head-fixation for three days before carrying out recordings.

## Recording neuronal activity in V1

**Two-photon calcium imaging.** Each mouse was recorded for at least three sessions. In vivo recordings were performed 15–45 days after the virus injection. We used a commercial two-photon microscope with a resonant-galvo scanhead (B-scope, ThorLabs) controlled by ScanImage 4.2 (ref. [52]), with an acquisition frame rate of about 30 Hz (at 512 by 512 pixels, corresponding to a sampling rate of about 4.3 Hz). The field of view was 550–600 µm. We imaged seven planes at 15–45-µm steps, starting at various positions below the brain surface (from 0 to −150 µm) to sample different cortical depths and therefore subtypes

recorded simultaneously during different sessions. Imaging calcium activity was performed at a wavelength of 920 nm or 980 nm. Three computer screens spanning −135 to +135 visual degrees ($v°$) along the azimuth axis and −35 to +35 $v°$ along the elevation axis were used to display visual stimuli. During the presentation of visual stimuli, we switched off the red gun of the monitors to prevent light from the monitors contaminating the red fluorescent channel.

At the end of each recording session, reference z-stacks were acquired. Starting at the same position as the imaging planes, we acquired two z-stacks of about 400 µm depth, with a 1-µm step between planes. The first one, called the GCaMP z-stack, was acquired at the same wavelength as the calcium imaging (920 or 980 nm). The second one, called the reference z-stack, was acquired at 1,040 nm to image mCherry fluorescence.

Before euthanizing each mouse, we acquired structural z-stacks (ranging from the brain surface to 400 µm deep) at 1,040 nm to get an image of the mCherry cells across the whole craniotomy (including the position where the red fluorescent beads were injected). This structural z-stack was used to select slices on which to perform transcriptomic analysis, and to provide an initialization point for the registration algorithm.

**Initial retinotopic mapping.** All recordings were targeted to the V1 monocular region (>60° azimuth). To find this region, during the first imaging session, we initially mapped the retinotopy of different candidate fields of view, using single-plane imaging. Sparse noise stimuli were presented to the mouse, consisting of black or white squares of width 4.5° visual angle on a grey background at a frame rate of 5 Hz for 10 min. Squares appeared randomly at fixed positions in a 16 by 60 grid, spanning the retinotopic range of the computer screens. 1.5% of the squares were shown at any one time.

**Visual stimulation.** Drifting gratings were centred on the mean receptive field of the microscope's field of view. Gratings had a duration of 0.5 s, temporal frequency of 2 Hz and spatial frequency of 0.15 cycles per degree. The gratings drifted in 12 different directions (from 0 to 330°, separated by 30°) and were of 3 different sizes (5°, 15° and 60° diameter).

Natural scenes from the ImageNet database were contrast-normalized and presented as described previously[34]. Each image was presented for 0.5 s with an interstimulus interval uniformly distributed from 0.3 to 1.1 s. Five per cent of the total presentations was grey stimuli. During each session we presented a given set of 1,000 different natural images twice (corresponding to a subset of the 2,800 images that were originally used[34]).

On each recording session we presented the same random sparse noise stimuli that were used to map retinotopy (see above), for 30 min.

Spontaneous activity was recorded in front of a uniform grey screen, set to a steady cyan level equal to the background of all the stimuli presented for visual responses protocols. The duration of these grey screen presentations was typically between 15 and 20 min.

## Eye-tracking

We used a collimated infrared LED (SLS-0208-B, lpeak = 850 nm; controller: SLC-AA02-US; Mightex Systems) to illuminate the eye contralateral to the recording site. Videos of eye position were captured at 30 Hz with a monochromatic camera (DMK 21BU04.H, The Imaging Source) equipped with a zoom lens (MVL7000; Navitar), and positioned at approximately 50° azimuth and 50° elevation relative to the centre of the mouse's field of view. Contamination light from the monitors and the imaging laser was rejected using an optical band-pass filter (700–900 nm) positioned in front of the camera objective (long-pass 092/52 × 0.75, The Imaging Source; short-pass FES0900, Thorlabs).

## Processing of calcium imaging

Two-photon calcium data were processed using Suite2P (ref. [53]). Neuropil contamination was corrected by subtracting from each region

of interest (ROI) signal its surrounding neuropil signal multiplied by a constant factor of 0.7. Calcium traces were deconvolved using non-negative spike deconvolution[54] with a calcium indicator decay timescale of 1.5 s. ROIs were manually curated to make sure that only cell bodies were considered for further analysis.

## coppaFISH

Many approaches to highly multiplexed mRNA detection have been described[55–73]. The coppaFISH method is a development of an in situ sequencing method[28] (Extended Data Fig. 1). The method uses reverse transcription, padlock probes and rolling-circle amplification to amplify mRNAs to DNA rolling-circle products (RCPs) that contain multiple copies of a 20-nucleotide (nt) barcode sequence, and then detects their location combinatorially in 7 rounds of 7-colour fluorescence imaging.

**Gene selection and DNA probe design.** A panel of 73 genes was selected to allow the identification of cortical cell types. This panel is a subset of a panel of 99 genes described in a previous study[28], which was picked based on scRNA-seq data using an algorithm that predicts which gene combinations are required to identify fine transcriptomic subtypes. Retrospective analysis analysing the contribution that each gene made to classification accuracy revealed that 26 genes in this panel served no purpose in accurately classifying cells (figure S16 of ref. [28]), leading to their removal from the panel. One gene (*Yjefn3*) was detected in our experiments, but could not be used to assign cells to transcriptomic subtypes as it was not present in the reference scRNA-seq dataset[3]. In the main text we therefore refer to a 72-gene panel.

Multiple padlock probes were designed for each gene, spanning the length of the cDNA (Supplementary Data 2). The number of different padlock probes per gene was chosen on the basis of the expression for each specific gene as determined by scRNA-seq. This means that fewer padlock probes were used for genes with high expression and vice versa (for example, four padlock probes were designed for *Sst* but 10 were designed for *Chodl*). All padlock probes consisted of two 15–20-nt recognition sites, a 20-nt gene barcode (unique to each gene) and a 20-nt anchor sequence (identical for all genes and padlock probes).

Padlock probes were designed using previously described software[28]. In brief, this software finds suitable RNA target sequences by restricting the melting temperature of the binding sequence, and by aligning the candidate sequences to the mouse whole transcriptome (RefSeq database) using BLAST+ to check for specificity. Any candidate targets for which another transcript or non-coding RNA sequence matched the target with more than 50% coverage, 80% homology and coverage spanning the central 10 nt of the target sequence were excluded. For each padlock probe, we also designed a specific primer for reverse transcription: a 15-nt-long DNA oligonucleotide that binds the region upstream of the mRNA sequences targeted by the padlock probes (Supplementary Data 3). The use of specific primers greatly improved the number of RCPs obtained per section compared to random primers (Bugeon, S. et al., unpublished observations).

To determine the gene-specific DNA barcode sequences (and the anchor sequence), 240,000 orthogonal 25-mer oligonucleotide sequences[74] were trimmed to 20 nt from the 5′ end and screened for melting temperature (between 55 °C and 56 °C using the SantaLucia method). They were further screened for orthogonality with mouse sequences using BLAST+ with the NCBI mouse genomic plus transcript (Mouse G +T) database. We used the following BLAST parameters: "-reward", 1, "-penalty", -2, "-gapopen", 2, "-gapextend", 1, "-evalue", 10. Any matches in this blast search were removed from the pool. Next, we checked for potential cross-reactivity of the remaining sequences to themselves using the same BLAST parameters, and any hits were removed, resulting in 6,397 possible sequences. The barcode sequences were chosen from this pool.

The combinatorial imaging strategy used two types of DNA probes. Seven 'dye probes' were designed, each consisting of a 20-nt-long DNA oligo conjugated to one of the seven following fluorophores: DY405, AF488, DY485xL, AF532, AF594, AF647 and AF750; the same dye probes were used on each imaging round (Supplementary Data 4). In addition, a set of 40-nt 'bridge probes' were designed for each imaging round, linking each gene's RCP barcode to one of the seven dye probes (Extended Data Fig. 1 and Supplementary Data 5). The bridge probes thus caused each gene to show up in a specific colour channel on each round. This two-part strategy of linking the seven dye probes to the RCPs with bridge probes provides a substantial cost saving over making $N_{genes} \times N_{rounds}$ dye probes, as dye-coupled probes are much more expensive than simple DNA.

Each gene was assigned a sequence of dyes for the seven imaging rounds using a Reed–Solomon coding scheme[75] (Supplementary Data 6), which constructs sequences of minimum possible overlap. Specifically, the genes were numbered by integers $g$, and converted to a base 7 representation $g_2 g_1 g_0$. The dye assigned to gene $g$ on round $r$ was

$$D_{gr} = g_2 r^2 + g_1 r + g_0,$$

where addition and multiplication are understood to happen modulo 7. Codes 0 to 6, which correspond to the same colour in each round, were not used as these codes could not be distinguished from fixed background fluorescence.

All custom DNA oligos (padlock probes, primers, bridge probes and dye probes) were obtained from Integrated DNA Technologies. Padlock probes were ordered as 5′ phosphorylated 4 nmol Ultramer DNA oligos; all other oligos were ordered as classical 25 nmol DNA oligos. The DNA sequences for all 556 primers and padlock probes, 511 bridge probes and 7 dye probes are provided in Supplementary Data 2–5.

**Tissue preparation.** After the in vivo recordings were finished, mice were anaesthetized with isoflurane and then injected with a lethal dose of sodium pentobarbital (0.01 ml g⁻¹). The fresh brains were then dissected out from the skull, taking great care to preserve the integrity of the tissue and avoid warping. The brains were then placed in OCT (Sakura Finetek) and left to freeze on dry ice for 30 min. The samples were then stored at −80 °C until slicing. Sagittal sections (15-µm thick) were then obtained using a Leica Cryostat for each brain and mounted on gelatine-coated borosilicate glass coverslips (22 x 55 mm). Gelatine-coated coverslips allowed tissue section adhesion to the coverslip and RNA preservation throughout the protocol. To make them, coverslips mounted on a rack were dipped for 30 s in a solution of 2% w/v gelatine and 0.2 % w/v chromium potassium sulfate dodecahydrate in distilled water (https://www.rndsystems.com/resources/protocols/protocol-preparation-gelatin-coated-slides-histological-tissue-sections). Two to three brain sections were thaw-mounted on each coverslip and then frozen and stored at −80°C.

It was not necessary to bleach the native fluorescence of mCherry and GCaMP (which might in principle interfere with later fluorescence imaging), as these faded completely during standard tissue processing.

**In situ RCP production.** The RCPs were prepared as described previously[28], with some modifications. First, coverslips were taken out of the freezer and then directly pre-fixed using 4% paraformaldehyde (PFA) for 5 min at room temperature. This pre-fixation was followed by a quick wash with nuclease-free phosphate-buffered saline (PBS), and incubation in 0.1 M HCl for 5 min at room temperature. After one more PBS wash, the sections were incubated in 70% ethanol for 1 min and then in 100% ethanol for 1 min at room temperature. The coverslips were then left to dry in air. To keep the reagents on the tissue sections, a barrier was drawn around each section using a hydrophobic barrier PAP pen (ImmEdge Hydrophobic Barrier PAP Pen H-4000, Vector Laboratories).

The sections were then directly incubated in reverse transcription mix overnight at 37 °C in a humidified chamber (Slide staining system, StainTray M918, VWR). The mix contained 0.5 mM dNTP mix (Thermo

Fisher Scientific), gene-specific primers (10 μM each), 0.2 μg μl$^{-1}$ BSA (NEB), 1 U μl$^{-1}$ RIBOPROTECT RNase Inhibitor (Blirt) and 20 U μl$^{-1}$ TranscriptMe reverse transcriptase (Blirt) in 1× reverse transcription buffer (Blirt). The mix was removed and fresh 4% (w/v) PFA in PBS was added to the sections without any wash in between. This post-fixation step aimed to cross-link newly synthesized cDNA to the cellular matrix and was carried out at room temperature for 30 min, followed by two washes in PBS. RNaseH digestion, padlock hybridization and ligation were then performed using a single reaction mix. The mix contained 0.05 M KCl (Sigma), 20% ethylene carbonate (Sigma), 10 nM of each padlock probe (557 probes), 0.2 μg μl$^{-1}$ BSA, 0.3 U μl$^{-1}$ Tth DNA Ligase (Blirt) and 0.4 U μl$^{-1}$ RNase H (Blirt) in 1× Ampligase buffer (Epicentre). The sections were first incubated at 37 °C for 30 min for RNaseH digestion and moved to 45 °C for 60 min for stringent hybridization and optimal DNA ligase activity. The sections were then washed twice in PBS. Finally, for rolling-circle amplification, the sections were incubated in a mix containing 5% glycerol (Sigma), 0.25 nM dNTP mix, 0.2 μg μl$^{-1}$ BSA, 0.2 U μl$^{-1}$ EquiPhi29 DNA Polymerase (Thermo Fisher Scientific) and 1× EquiPhi29 buffer (Thermo Fisher Scientific) overnight at 30 °C.

RCP production was quickly verified before full barcode read-out by hybridizing a AF750-conjugated oligonucleotide probe (IDT) to the anchor sequence present in all the RCPs. Sections were incubated for 15 min at room temperature in a hybridization mix containing 10 nM of the dye probe, 2× SSC, 20% ethylene carbonate and H$_2$O. They were then washed twice with 2× SSC. The SSC was then removed from the sections and the coverslips were mounted onto SuperFrost plus (VWR) glass slides using 10 μl SuperFrost gold antifade mountant (Life Technologies). Images of the ROI (visual cortex) were then acquired to visualize the RCPs.

**Imaging of the in situ barcodes (read-out).** All seven rounds of imaging occurred in a custom flow cell, using automated fluidics to wash appropriate bridge and dye probes before each round. The flow cell frame was designed using Blender and printed, using an Ultimaker S5 3D printer, in polylactic acid filament (PLA) with polyvinyl alcohol (PVA) support structures. The PVA support was removed after printing by placing the flow cells in water on a rocker overnight. To make the flow cell air-tight, two 22 ×55-mm glass coverslips (one with RCP-containing sections and one bare) and two approximately 40-cm-long EFTE tubes (Tubing Tefzel Nat 0.0625 inch outer diameter x 0.020 inch inner diameter) were securely mounted using UV curing cement (Norland Optical Adhesive 81) and a UV curing LED system with driver unit and a handheld 365-nm light source (ThorLabs, CS20K2). The coverslip with the sections was mounted so that the side with the sections faces the inside of the flow cell.

The Imaging set-up consisted of a Nikon Eclipse Ti2 microscope with a NIR-LDI laser panel and a Zyla sCMOS 4.2 camera (Andor). The fluidics set-up consisted of a Minipuls 3 pump (Gilson) and two linked MVP multivalves (Hamilton), each with 8 ports. Nikon NIS elements software (v.5.20.02, build 1453) was used to acquire the images and communicate with a second computer controlling the fluidic pump and multivalves. The opening of the valves and the speed and the duration of the pump's activity was managed by an edited version of Kilroy software (https://github.com/ZhuangLab/storm-control; edits available at https://github.com/acycliq/storm-control). The imaging and sequencing chemistry were coordinated by NIS elements software (ND sequence acquisition module), which communicates with the computer running Kilroy by sending TTL pulses through a National Instruments NI-USB 6008 board.

Before sequencing, 15-ml falcon tubes containing bridge probe mixtures for each of the seven imaging rounds, as well as one each for dye probe mixture, anchor probe mixture, imaging buffer, distilled water, 2× SSC and 100% formamide, were attached to the multivalves via EFTE tubing and flangeless fittings (1/16 inch Red Delrin, IDEx Health and Science LLC). The mixtures for bridge, dye and anchor probes contained the appropriate oligonucleotides diluted to 10 nM each in 2× SSC, 20% ethylene carbonate and H$_2$O. The bridge probe mix for the final anchor round contained the Cy7-conjugated anchor probe as well as the Gad1 bridge probe that binds to the AF532 dye probe (Gad1_r6 – 10 nM) and DAPI to stain the cell nuclei. A fresh formamide (S4117 Millipore) aliquot was used for every experiment (stored at 4 °C). The flow cell was then mounted onto the multi-slide stage and connected to the pump and multivalves via EFTE tubing. The speed of the pump was adjusted to approximately 0.4 ml s$^{-1}$. To fill the flow cell, each solution was flushed through the fluidics system for 4 min (the flow cell volume is approximately 1 ml).

In total, eight rounds of imaging were done for each imaging experiment: seven rounds to decode the barcodes and one final anchor round to detect the position of every RCP that was used for later image alignment. In each round, sections were first incubated in 100% formamide for 15 min to strip the RCPs from any previous labelling. The formamide was then flushed from the flow cell with water for 4 min and then with 2× SSC for 4 min. The sections were next incubated in that round's bridge probe mix for 15 min and washed with 2× SSC. After this, the sections were incubated in the dye probe mix for 15 min, and again washed with 2× SSC. The flow cell was filled with an imaging buffer consisting of glucose oxidase and catalase containing oxygen scavenging system[76] to protect the fluorophores from photobleaching during imaging.

After each round of sequencing chemistry, 16-bit images were acquired using wide-field epifluorescence excitation, and a 40× magnification air-objective (CFI Plan Apochromat Lambda 40XC, NA 0.95). Images consisted of z-stacks (z-step: 0.5 μm) in seven different colour channels corresponding to the seven fluorophores (Fluorophore – excitation wavelength, emission filters: Dy405 – ex405, 460/50 m; AF488 – ex470, 525/36 m; Dy485xl – ex470, 632/60 m; AF532 – ex520, 560/40 m; AF594 – ex555, 632/60 m; AF647 – ex640, 700/75 m; AF750 – ex730, 811/80 m). Each tile was 2,048 × 2,048 pixels (pixel size: 0.1625 μm). The imaging parameters were adjusted to cover only the ROI (V1) and usually consisted of 10–15 tiles with 10% overlap. The Nikon perfect focus system was used to make sure that the focus stayed relatively constant across imaging rounds. Image files were saved in Nikon's native ND2 format.

### In situ data analysis

The in situ data were analysed with a suite of custom software for image processing, gene calling and cell calling. All code was written in MATLAB, and is freely available at https://github.com/jduffield65/iss. This software was developed from that described previously[28], but has been greatly modified, so is described in full here.

The in situ data consist of eight rounds of multispectral imaging (seven combinatorial rounds, and one reference round in which all RCPs are labelled via the anchor sequence, together with an additional stain for *Gad1* RCPs and a DAPI stain). Because the tissue sample is too large for a single camera image, imaging occurs in overlapping tiles. In each tile, a focus stack of wide-field images was taken for each colour, and flattened into two dimensions (2D) using an extended depth of focus algorithm[77]. The data therefore consist of a set of images:

$$I_{R,C,T}(\mathbf{x}).$$

Here, $I$ gives the pixel intensity for sequencing round $R$, colour channel $C$, tile $T$, and pixel coordinates $\mathbf{x}$ within this tile. The processing pipeline to identify detected genes comprises several steps: initial registration; RCP spot detection and fine registration; cross-talk compensation; and gene calling. These analyses proceed without ever 'stitching' all the tiles into a single large image; this approach allows processing of very large datasets on computers with limited memory, and also easily allows non-rigid alignments. Before the pipeline, all RCP images are linearly filtered by convolving with a difference of Hannings: a Hanning

of radius 0.5 µm minus a Hanning of radius of 1 µm, both normalized to have sum 1. The DAPI background images are filtered with a disk-shaped top-hat filter with radius of 8 µm.

**Initial registration.** The initial registration step finds offsets between all image tiles using the anchor images taken on round 8 (which we refer to as 'reference images'). We use this to define a global coordinate system for the entire tissue sample.

Because we use a square tiling strategy, each tile may have up to four 'neighbours': other tiles with which it has a region of substantial overlap. We denote the set of neighbouring tile pairs as $\mathfrak{N}$.

Spots first are detected in each tile's reference images, as local maxima of the filtered image exceeding a fixed detection threshold. To align the reference images, we loop over all pairs of neighbouring tiles, and compute an offset to register the overlapping regions of the filtered reference images of these two tiles. The offset between two tiles $T_1$ and $T_2$ is found by exhaustive search over all 2D shifts in a range around to the shift expected from the microscope's position sensor. For each shift, we find for each spot $s$ on $T_1$ the pixel distance $D_s$ to the nearest spot on $T_2$ after the shift has been applied. A score is computed as $\sum_s e^{-D_s^2/8}$, and the final shift vector $\mathbf{\Delta}_{T_1, T_2}$ is taken as the one that maximizes this score; that is, the one with the most near neighbours.

We define a single global coordinate system by finding the coordinate origin $\mathbf{X}_T$ for each tile $T$. Note that this problem is overdetermined as there are more neighbour pairs than there are tiles. We therefore compute the offsets by minimizing the loss function

$$L = \sum_{(T_1, T_2) \in \mathfrak{N}} |\mathbf{X}_{T_1} - \mathbf{X}_{T_2} - \mathbf{\Delta}_{T_1, T_2}|^2$$

Differentiating this loss function with respect to $\mathbf{X}_T$ yields a set of simultaneous linear equations, the solution of which yields the origins of each tile on the reference round. The results of this step suffice to define a global coordinate system, but do not provide pixel-level alignment of images from multiple colour channels on multiple rounds, owing to the occurrence of chromatic aberration and small rotational or non-rigid shifts. The latter will be dealt with in the next step, through point cloud registration.

**Spot detection and fine registration.** The second processing step detects spots in all images of the seven sequencing rounds, performs fine alignment of colour channels and sequencing rounds, and computes for each spot a position in global coordinates and an intensity vector summarizing that spot's detected fluorescence in each round and channel.

The most intricate part of this step is fine image registration. Even though the same tile layout is used for all sequencing rounds, the precise positions of the tiles may differ owing to slight shifts in the placement and rotation of the sample. Thus, a single spot might be found on different tiles in different sequencing rounds. Furthermore, owing to chromatic aberration, a spot may be in slightly different positions (although not different tiles) in different colour channels. Because most spots are only a few pixels in size, even a one-pixel registration error can compromise accurate RNA reads.

A global coordinate is defined for each of the spots detected in the reference images using the initial registration described above. In regions where tiles overlap, duplicate spots are rejected by keeping only spots which are closer in global coordinates to the centre of their original tile than to any other.

Next, spot positions are detected in images from all sequencing rounds and colour channels. These are used to align each round and colour channel to the corresponding tile's reference image, using point cloud registration. Specifically, we fit an affine transformation from each reference image to the images of the corresponding tile for all rounds and colour channels, using the iterative-closest point (ICP)

algorithm with matches further than 3 pixels away excluded. These affine transformations can include shifts, scalings, rotations and shears, but we did not find it necessary to introduce nonlinear warping transformations within tiles (nonlinear transformations can still occur globally by variation of the affine transformation across tiles). As the ICP algorithm is highly sensitive to local maxima, it is initialized from a shift transformation computed by the same method used to find the overlap between reference images; that is, the shift that maximizes the number of near neighbours as measured by $\sum_s e^{-D_s^2/8}$. When spots are located on neighbouring tiles on different rounds, the corresponding images are again registered with ICP.

Finally, a seven-dimensional intensity vector $\mathbf{V}_{s,r}$ is computed for each spot $s$ in each round $r$, by reading the intensity from the aligned coordinate of each filtered image.

**Cross-talk compensation.** The last step associating spots to genes consists of transforming the intensity vectors to gene identities.

An important consideration in this stage is that cross-talk can occur between colour channels. Some cross-talk may occur owing to optical bleedthrough; additional cross-talk can occur owing to chemical cross-reactivity of probes. With the current hybridization chemistry (unlike previous sequencing-by-ligation chemistry), the degree of cross-talk tends to be constant within a round, so we learn a single $7 \times 7$ cross-talk matrix and apply it to all rounds.

To estimate the cross-talk present, we first collect a set of seven 7-dimensional vectors $\mathbf{V}_{s,r}$ containing the intensity in each colour channel of all well-isolated spots $s$ in all rounds $r$. Only well-isolated spots are used to ensure that cross-talk estimation is not affected by spatial overlap of spots corresponding to different genes; a spot is defined as well-isolated if the reference image intensity averaged over an annular region (4–14 pixel radius) around the spot is less than a threshold value. Cross-talk is then estimated by running a scaled $k$-means algorithm[78] on these vectors, which finds a set of seven vectors $\mathbf{c}_d$ ($d$ refers to one of the seven dyes), such that the error function:

$$\sum_{s,r} \min_{d_{s,r}, \lambda_{s,r}} |V_{s,r} - \lambda_{s,r} \mathbf{c}_{d_{s,r}}|^2$$

is minimized; in other words, it finds the seven intensity vectors $\mathbf{c}_d$ such that each well-isolated spot on round $r$ is close to a scaled version of one of them.

The cross-talk matrix is used to predict the colour profile expected for an RCP of each gene $g$, for each colour channel and round. If gene $g$ is assigned the dye $d_{g,r}$ in round $r$, the predicted 49-dimensional intensity vector is obtained by concatenating the corresponding cross-talk vectors.

**Gene calling.** Improvements in tissue processing and in situ chemistry mean that our current methods produce substantially more RCPs than the previous in situ sequencing method[28]. Consequently, the fluorescence of neighbouring RCPs often overlaps, which would render the previous detection method unable to find them. To allow resolution of overlapping spots, we therefore developed a gene-calling algorithm, based on orthogonal matching pursuit (OMP)[78]. This algorithm also allows for subtraction of background autofluorescence. Essentially, OMP repeatedly tests whether the 49-dimensional fluorescence vector of a pixel overlaps with the predicted fluorescence vector of each gene; if so, a gene is detected at that location, its code is projected out from the fluorescence vector, and the process repeats.

The OMP algorithm fits a 49-dimensional image (one dimension for each combination of round and colour channel) as a sum of 49-dimensional code vectors. There is one code vector $\mathbf{a}_g$ for each gene, and one 'background' code $\mathbf{a}_c^B$ for each colour channel, which has equal intensity for all rounds in one colour channel only. These background codes account for tissue autofluorescence, which will affect all imaging rounds equally.

The gene codes $\mathbf{a}_g$ are derived from using knowledge of the Reed–Solomon assigned dyes $d_{g,r}$ for each gene in each round and the cross-talk matrix columns $\mathbf{c}_d$. These codes take into account the fact that different genes can have consistently different intensities in different rounds, which may arise from non-uniformity in the synthesized concentrations of the bridge probes. To account for this non-uniformity, we learn a scale factor $\varepsilon_{g,r}$, and predict the 49-dimensional gene code for gene $g$ as a concatenation:

$$\mathbf{a}_g = [\varepsilon_{g,1}\mathbf{c}_{d_{g,1}}; \varepsilon_{g,2}\mathbf{c}_{d_{g,2}}; \varepsilon_{g,3}\mathbf{c}_{d_{g,3}}; \varepsilon_{g,4}\mathbf{c}_{d_{g,4}}; \varepsilon_{g,5}\mathbf{c}_{d_{g,5}}; \varepsilon_{g,6}\mathbf{c}_{d_{g,6}}; \varepsilon_{g,7}\mathbf{c}_{d_{g,7}}]$$

We will describe the general algorithm before specifying how $\varepsilon_{g,r}$ is chosen.

The OMP algorithm expresses the 49-dimensional fluorescence vectors $\mathbf{v}_p$ for each pixel $p$ as a weighted sum of code vectors: $\hat{\mathbf{v}}_p = \sum_{i=1}^{n_p} \alpha_{g_{i,p}} \mathbf{a}_g + \sum_{c=1}^{7} \beta_{c,p} \mathbf{a}_c^B$. Each step of the algorithm can add a code to the set of code vectors $\{g_{i,p} : i = 1...n_p\}$ used to approximate pixel $\mathbf{v}_p$; the 7 background codes are always included. The gene set is initialized to be empty, and to choose which gene code, if any, should be added on each step, the algorithm computes how much the residual $|\mathbf{v}_p - \hat{\mathbf{v}}_p|^2$ would decrease for each possible addition to the set, and picks the gene giving maximum decrease, provided that this decrease is above a threshold of 0.0612 multiplied by the second largest absolute value of $\mathbf{v}_p$ (clamped by a minimum threshold of 0.01 and a maximum threshold of 3.0), up to a total of 6 genes per pixel. After this iterative process has terminated for all pixels, an image is made for each gene, containing the gene's weight for each pixel or zero if that gene is not in the pixel's gene set. RNA detections are found as local maxima of this image, subject to a thresholding criterion; the criterion takes into account several factors and is best understood by examining the source code (https://github.com/jduffield65/iss).

To choose the scale factors $\varepsilon_{g,r}$, a single iteration of the OMP algorithm is run with all $\varepsilon_{g,r} = 1$. Local maxima are detected as just described, but with a more stringent threshold (see source code for details) to ensure only unambiguous gene detections are used. We then compute a 7-dimensional mean intensity vector $\bar{\mathbf{v}}_{g,r}$ of all detected spots for each gene in each round. We then find the scale factors $\varepsilon_{g,r}$ for each round and gene as the least-squares solutions of

$$\bar{\mathbf{v}}_{g,r} \approx \varepsilon_{g,r}\mathbf{c}_{d_{g,r}}$$

**Cell calling.** The DAPI image was used to segment the cells. This was performed by detection of the local maxima in each cell followed by watershed segmentation. The segmentation of matched cells and their close neighbours was manually curated.

To assign cells to transcriptomic subtypes, we used the previously described pciSeq algorithm[28], a Bayesian method that assigns each in situ cell a posterior probability of belonging to each of a set of cell classes defined by prior scRNA-seq. As we recorded from V1, we used the transcriptomic clusters defined in a previous study[3], using only cells from V1 to compute the mean expression of each cluster. These clusters are similar to those produced from other cortical and hippocampal regions, but may differ, particularly for fine subtypes (see Fig. S1 of ref. [6] and Extended Data Fig. 3 of ref. [1] for the probable relationships between these clusters and other classification schemes). The read counts of the scRNA-seq data were divided by 100 to predict the expected in situ RNA count; a further gene-dependent efficiency factor was estimated by the algorithm. The pciSeq algorithm produces a probability for each cell to belong to each class, which we converted to a 'hard' classification by assigning each cell to the subtype of maximum a posteriori probability; cells for which this maximal probability was less than 0.5 were not analysed further (around 2% of matched cells; Extended Data Fig. 4c). We assigned cells using all 109 transcriptomic clusters defined previously[3], including inhibitory neurons of all layers

and non-GABAergic cells. Nevertheless, the algorithm assigned the imaged inhibitory cells to just 35 of these clusters, corresponding to superficial-layer inhibitory subtypes. To evaluate the accuracy of this algorithm, we subsampled the read counts from the scRNA-seq dataset using a Poisson distribution and then estimated the a posteriori probability of belonging to each subtype similarly to in situ cells (Extended Data Fig. 10). This showed that our 72-gene panel yielded an estimated assignment accuracy of 98.1%, 96.6% and 76.4% at the subclass, type and subtype levels, respectively.

### Registration of the in vivo and ex vivo cells

We used inhibitory cells, labelled in vivo by mCherry (Gad2-mCherry mice), as landmarks to perform the registration between the in vivo Gad-mCherry volume and the ex vivo brain sections (Extended Data Fig. 2). This alignment made use of two high-resolution reference z-stacks taken for each subject following each imaging session. The 'GCaMP z-stack' was taken using the same wavelength as functional imaging (920 or 980 nm), covering the same volume but at higher resolution. The 'mCherry z-stack' was acquired in the same volume with a 1,040 nm excitation wavelength to detect inhibitory neurons in Gad2-mCherry mice, but also provided some GCaMP signal in the green channel (although this signal was much lower than for the GCaMP z-stack taken at 920 nm). The different excitation wavelength of these two z-stacks led to a small chromatic aberration, which was only significant in depth. To correct this aberration, we used the green channel found in both imaged volumes, registering planes of the GCaMP z-stack to the mCherry z-stack using fast Fourier transform (FFT) convolution. This was achieved by finding the best matching plane from the later z-stack for each GCaMP z-stack plane as the z position that gave the highest FFT cross-correlation. In addition, a 'global z-stack' was made following the final functional imaging session, which covered the entire region under the craniotomy; this was used for coarse initial registration of the in situ slices.

**Aligning calcium ROIs to the mCherry z-stack.** To align the imaging planes of one functional two-photon session to the GCaMP z-stack, we first obtained their theoretical position using the measured position of the objective for each line scanned (for both the functional imaging planes and the GCaMP z-stack). We then estimated the z-drift during the recording session: the position of the calcium imaging planes over time in comparison to this GCaMP z-stack. To do so, a mean image of each functional imaging plane was obtained for 1 min every 7 min of the recording. These mean images were then aligned to the z-stack using FFT convolution. We then took the median of this z-drift over time and used it to correct the theoretical imaging plane position. We then performed FFT-based registration to correct for a small shift in $X$ and $Y$ between the actual mean image and the reconstructed image. We thus found the position of the imaging planes (and therefore of each functional ROI) in the GCaMP z-stack. These were then aligned to the mCherry z-stack using the transformation described above (chromatic aberration in depth).

**Aligning brain slices to the mCherry z-stack.** To register the positions of the in-situ-detected inhibitory neurons to the 3D mCherry z-stack, we used a custom point cloud registration method, using inhibitory neurons as landmark points. MATLAB code and an example pipeline script can be found at https://github.com/ha-ha-ha-han/NeuromicsCellDetection/, and at https://github.com/sbugeon/NeuromicsCellDetection.

During slicing, the latero-medial order of the sagittal brain sections was carefully recorded. To find the sections corresponding to the imaged region, we first screened them by generating RCPs for every 20th section, and staining with the *Gad1* bridge probe and its corresponding dye probe to label inhibitory neurons. The position of the fluorescent bead injection was usually visible on one of the sections, allowing us to infer the approximate position of every slice (based on the known order and thickness of slicing).

Fine registration of screened sections to the in vivo reference $z$-stack started with cell detection in vivo and ex vivo. To detect cells in the in vivo mCherry $z$-stack, each plane was contrast-normalized to correct for the loss of brightness with depth using the following MATLAB GUI https://github.com/nadavyayon/Intensify3D/blob/master/User_GUI_Intensify3D.m (which performs background and signal estimation based on user defined thresholds), and the $z$-stack was then filtered using a 3D median filter of radius 2 μm to reduce background noise. The mCherry-positive cells were automatically detected on these images using a 3D difference-of-Gaussians filter followed by watershed segmentation. Manual curation was performed to correct for missed or false positive detections. To detect inhibitory cells in the ex vivo slices, we used the in situ expression of *Gad1* in the reference round, as native mCherry fluorescence was not preserved in the fresh-frozen sections. *Gad1* detections formed clusters on GABAergic cells (Extended Data Fig. 2), which were detected by Gaussian smoothing of the *Gad1* RCP images and applying a difference-of-Gaussian filter and watershed segmentation to detect individual clusters. Finally, we manually curated these detections using the full in situ expression of the 72 genes to determine putative interneurons on the basis of the main inhibitory cell markers such as *Vip*, *Sst*, *Pvalb* and so on.

The slices were first coarsely registered using brain structures (hippocampus, brain surface and so on) visualized using the anchor and nuclear staining. Next, they were finely registered using an algorithm to register a 2D point cloud, corresponding to inhibitory neurons in the *ex vivo* slice, into a 3D point cloud, corresponding to inhibitory neurons in the in vivo volume. To align these point clouds, we used rigid registration with 6 degrees of freedom ($\alpha, \beta, \gamma, x, y, z$), where $\alpha, \beta, \gamma$ are the rotation angles, and $x, y, z$ are translational shifts (non-rigid point cloud registration is possible, but we found it to be unnecessary). The registration algorithm searched for the parameters ($\alpha_{max}, \beta_{max}, \gamma_{max}, x_{max}, y_{max}, z_{max}$) that maximize the match of the 2D slice to the corresponding section of the 3D volume.

Because this registration problem has a large number of local maxima, we performed an exhaustive grid-search over these six parameters. Because Fourier convolution of 3D arrays is fast, but rotation of them is not, we used a hybrid point and Fourier method. An outer loop searches over all combinations of rotation angles ($\alpha, \beta, \gamma$), with an initial step size of 1°, refined to 0.5° for finer alignment, and rotates the 3D point cloud accordingly. A 3D volumetric image is then synthesized from these rotated points by adding a Gaussian peak at the location of each point. Each plane $z$ of this image is Fourier convolved with a fixed 2D array synthesized similarly for the 2D cloud, and the resulting 3D correlation map is stored, to accumulate a correlation score function $c(\alpha, \beta, \gamma, x, y, z)$. The top local maxima of this 6D array are found and ranked using both the intensity of the cross-correlogram peaks and the percentage of cells matched within a tolerance of 15 μm (to account for small non-rigid deformations). Finally, the match validity for each section was assessed manually by looking at the overlay between the interpolated cut from the reference $z$-stack and the *Gad1* RCP image. The rotation and translation parameters were manually adjusted to provide the best overlay between the two datasets. Typical rotation angles were found between −10° and 10° of the coarse manual registration, enabling us to save computation time by searching only this range.

**Aligning individual neurons.** Finally, a custom MATLAB GUI was used to curate the match between inhibitory cells in the in vivo recordings and the ex vivo sections. The GUI allowed us to visualize the in vivo mCherry image of each cell (obtained from the reference $z$-stack), the position of the ROIs on the reference $z$-stack and the overlap between the reference $z$-stack cross-sections and the in situ gene expression for the different genes. For each slice, we displayed all mCherry-positive ROIs that were less than 10 μm away from the found position of the slice in the reference $z$-stack. Each assignment of in vivo and ex vivo *Gad*-positive cells was curated manually on the basis of these data. At this stage, the boundaries that were initially found using automatic segmentation of the DAPI image were also manually adjusted for the matched cells and their neighbours, to correct for errors in DAPI segmentation that could affect the gene and cell-type assignment. This correction was based both on the DAPI image and on the in situ gene expression, which provided information that could indicate under-splitting in the DAPI segmentation of adjacent cells. We took a conservative approach to this manual curation process, discarding all imaged cells for which the match to ex vivo slices was not absolutely clear (around 50% of cells).

## Cell selection

We recorded a total of 3,469 (204 ± 42 per session) inhibitory cells and 6,684 (393 ± 173 per session) excitatory cells. Of these inhibitory cells, 1,515 (89 ± 31 per session) cells could be aligned to the ex vivo slices with good confidence, and were thus assigned a transcriptomic identity (see Supplementary Data 1). Some ex-vivo-identified cells were recorded in multiple imaging sessions. In all figures, a unique session was picked for each matched cell (except for Fig. 2, in which we show all cells in a single session, and for pairwise correlations that used all cells in all sessions: Figs. 3a and 5d and Extended Data Figs. 6b and 8c–e). The session assigned was chosen according to the percentage of time that the mouse spent running during this session, to maximize variability of behaviour while the cell was recorded. After removing these duplicates, we obtained 1,090 unique cells. Of these cells, 17 cells were removed because their maximal posterior probability was less than 0.5. Finally, 8 cells that were assigned to subtypes with fewer than 3 cells in total were discarded. The final population of 1,065 cells belonged to 35 transcriptomic subtypes.

For hierarchical analysis, the 35 subtypes were grouped into 11 types corresponding to putative anatomical or physiological cell types based on the previous literature. For *Pvalb* neurons, the grouping was unambiguous: the *Pvalb-Vipr2* subtype is genetically very different to all other *Pvalb* subtypes, and several studies have identified molecular markers of this subtype with chandelier cells[3,4,7,79]. For *Sst* cells, UMAP analysis (Extended Data Fig. 3) suggests that the two *Sst-Tac1* subtypes bridge a continuum between the two *Sst-Calb2* subtypes (identified as superficial-layer Martinotti cells[7,80,81]) and the *Pvalb-Tpbg* subtype (identified as superficial-layer *Pvalb* basket cells[7]). Patch-seq analysis confirms that *Sst-Tac1* cells have less axon in L1 and faster-spiking phenotypes than classical Martinotti cells[7]. We therefore identify the two *Sst-Tac1* subtypes as non-Martinotti *Sst* cells, acknowledging that these two *Sst* types are likely to tile a continuum, rather than truly being discrete cell groups. For *Lamp5* cells, we grouped subtypes on the basis of the results of previous studies[82,83]. The three subtypes that comprised the *Lamp5-Npy* group were identified as neurogliaform cells on the basis of their strong expression of *Npy*. The *Lamp5-Fam19a1-Tmem182* subtype was identified as canopy cells owing to expression of *Ndnf* but not *Npy*; the two remaining subtypes were identified as α7 cells owing to their strong expression of *Chrna7* and weak expression of *Ndnf* and *Npy*. For *Vip* cells, we divided subtypes by transcriptomic methods: UMAP analysis suggested a clear discrete distinction between two *Vip* subtypes characterized by expression of *Reln* as well as weaker expression of *Vip* itself. We are not aware of any specific study on these *Vip-Reln* cells; however, on the basis of their weak *Vip* expression and the fact that *Reln* is usually a L1 marker, we provisionally identify this type with the L1 VIP cells described previously[82]. *Serpinf1* subtypes were included with the *Vip* category as we do not see strong evidence for this as a discrete subclass. Finally, *Sncg* subtypes were divided into two types according to *Vip* expression, with *Sncg-Vip* and *Sncg-Pdzrn3* identified as small and large *Cck* cells, respectively[84,85].

## Data analysis

**Modulation index.** When comparing activity in two conditions (for example, visual stimulus versus grey screen; large versus small grating; running versus stationary synchronized), we used a modulation index computed as

$$\text{Mod index} = \frac{R - B}{R + B},$$

in which $R$ is the mean activity in the first condition (for example, during the response time window) and $B$ is the mean activity in the second condition (for example, during the baseline time window).

**Cell depth comparison to a previous Patch-seq study.** For the analysis validating coppaFISH subtype calling using cell depth (Fig. 1j), we used cells of all layers, not just the in-vivo-imaged cells of L1–L3. We used 14 sections for which gene expression was obtained from L1–L6 (all taken from the same mouse). DAPI segmentation was manually curated (see above) in all layers, and cell calling was performed on these sections using the standard method. This provided the cortical depth for about 47,000 cells, among which 2,130 were assigned to a GABAergic subtype. We normalized the measured cortical depth by the maximum cortical depth in these sections (750 μm) and computed the median cortical depth for each subtype with at least 4 cells (46 such subtypes were found). We then did the same thing for the Patch-seq data of a previous study[7], which gave 42 subtypes with more than 4 cells. We then compared the cortical depth of the subtypes with at least 4 cells in both datasets (33 subtypes in total; Fig. 1j).

**Determining behavioural states.** To distinguish the three main behavioural states during spontaneous behaviour, we used the running speed of the mouse as well as the strength of cortical oscillations. Running speed was measured by optical sensors facing the air-suspended ball[86], and was smoothed with a 2-s moving average filter. We considered the mouse stationary if this smoothed speed was less than 0.3 cm s$^{-1}$, and running otherwise. To distinguish between the synchronized and desynchronized stationary states, we first computed the first principal component (PC) of excitatory cells' activity using PCA, which revealed cells more active in passive or alert states, as previously described[34]. The activity of the 10% of cells with most negative weight on this PC was averaged, which provided a clear summary of the oscillation that appeared in some stationary periods (Fig. 2). Periods of synchronized activity were segmented manually based on the periods in which this average was clearly oscillating. To measure the oscillatory coupling of each inhibitory neuron, we then computed the Pearson correlation between each cell's z-scored activity and the average of this excitatory subpopulation during the synchronized periods.

**Comparison to transgenic mouse line data.** To validate our cell-type assignment, we compared the results obtained with post-hoc transcriptomic data with recordings performed using transgenic mouse lines (Extended Data Fig. 5). We analysed recordings from 18 transgenic mice (5 for *Pvalb*, 8 for *Sst* and 5 for *Vip*; 14 mice were re-analysed from ref. [30] and 4 new mice were added) and 23 sessions (6 for *Pvalb*, 9 for *Sst* and 8 for *Vip*) for a total of 2,589 identified cells (1,023 *Pvalb*, 572 *Sst* and 994 *Vip* cells).

For this analysis (Extended Data Fig. 5), we first deconvolved the calcium traces to inferred firing rates $f_i(t)$ for each neuron $i$ at time $t$ (ref. [53]). We considered two measures of neural activity for each cell $i$ and trial $n$: the average neural activity $r_i(n) = \langle f_i(t) \rangle_{t \in [t_n, t_n + \triangle T]}$ during stimulus presentation from the trial onset time $t_n$ to time $t_n + \Delta T$, and the average neural response $d_i(n) = r_i(n) - b_i(n)$, obtained after subtracting the pre-stimulus baseline activity $b_i(n) = \langle f_i(t) \rangle_{t \in [t_n - \triangle T, t_n]}$. The time window parameter $\Delta T$ took the value 1 s for the data from ref. [30] and 0.5 s for the new transgenic data and the post-hoc transcriptomic data, corresponding to the whole duration of the stimulus. We then computed the average activity and response for a given stimulus $s$ and locomotion condition $v$ ($v = 0$: stationary, $v = 1$: running): $\bar{r}_i(s, v) = \langle r_i(n) \rangle_{n \in \{s, v\}}$ and $\bar{d}_i(s, v) = \langle d_i(n) \rangle_{n \in \{s, v\}}$. We estimated the responsiveness of each neuron $i$ to visual stimuli by computing the $P$ value $p_i$ of

a paired $t$-test comparing $r_i(n)$ with $b_i(n)$ for all trials $n$ (pooling all different stimulus types to obtain one $P$ value per cell). For all subsequent analysis, we selected only cells with $P$ values < 0.05. Modulation of visual response by running was computed as follows: first, we computed the average responses $\bar{d}_i(v) = \langle d_i(n) \rangle_{n \in v}$ and standard deviation $\sigma_i(v) = \sigma_{i, t \in v}(d_i(t))$ across all stimuli for running and stationary trials ($v = 0$ and $v = 1$, respectively). We then computed a modulation index as follows: $M_i^{(3)} = \frac{\bar{d}_i(1) - \bar{d}_i(0)}{\sqrt{\sigma_i(1) + \sigma_i(0)}}$. This index was then normalized for each recording session $k$ as follows $M_i^{(3)} \to M_i^{(3)} / \sigma_{i \in k}(M_i^{(3)})$. We plotted the average modulation of visual responses by running versus the Pearson correlation coefficient of spontaneous activity and running speed $\rho_i$ (Extended Data Fig. 5a). Before computing the Pearson correlation coefficient, we smoothed the activity $f_i(t)$ and running speed $v(t)$ with a time average of 5 s. For this analysis, we selected only cells that had a cortical depth more superficial than −300 μm.

For estimating size tuning curves (Extended Data Fig. 5b), we z-scored the activity of each neuron as follows $\bar{z}_i(s, v) = [\bar{r}_i(s, v) - \langle \bar{r}_i(s, v) \rangle_{s, v}] / \sigma_{s, v}(\bar{r}_i(s, v))$ before averaging over cells of a given type.

To evaluate consistency between the physiological features identified with transgenic and transcriptomic cell-type identification, we trained a classifier to predict cell type from physiological features of each cell in the transgenic lines, and asked whether it generalized to the transcriptomic data (Extended Data Fig. 5c). We trained the classifier using 1,230 training cells (410 examples per cell type for the three cell types). The prediction was based on 14 features, which included normalized values of neural activity during different stimulus size and running condition $\bar{z}_i(s, v)$ (features 1–8); skewness of the calcium trace computed across the whole recording session (feature 9); the correlation of spontaneous activity with running speed $\rho_i$ (feature 10); the ROI diameter (feature 11); the cortical depth (feature 12); and two different measures of the difference in modulation by running: $M_{i, s=60°}^{(1)} - M_{i, s=60°}^{(1)}$ between large and small stimuli, in which $M_{i, s}^{(1)} = [\bar{r}_i(s, v = 1) - \bar{r}_i(s, v = 0)] / \sigma_{s, v}(\bar{r}_i(s, v))$; and $M_{i, s=60°}^{(2)} - M_{i, s=5°}^{(2)}$, in which $M_{i, s}^{(2)} = [\bar{d}_i(s, v = 1) - \bar{d}_i(s, v = 0)] / \langle \bar{r}_i(s, v) \rangle_{s, v}$ (features 13 and 14). We normalized features 9–14 by z-scoring them using the mean and standard deviation for each neuron of the transgenic mice, whereas features 1–8 were already normalized as in Extended Data Fig. 5b. We used cell types: $y = \{Pvalb, Sst, Vip\}$ as training labels. Using the 10 different randomized splits of training and test transgenic data, we applied three different linear classifiers: linear discriminant analysis; logistic regression (regularization parameter $C = 10$); and linear support vector classification (regularization parameters $C = 0.1$). The regularization parameters were chosen after a fourfold cross-validation over the different randomized training sets scanning over $C = \{10^{-3}, 10^{-2}, \ldots 10^2\}$. Applying the classifier to transcriptomic data gave equivalent performance to test-set transgenic data, indicating that the two methods are consistent.

**Response to drifting gratings.** Responsive cells (either activated or suppressed) were defined using a repeated measures ANOVA model (fitrm in MATLAB) with the stimulus direction (12 levels) and size (3 levels) as between-subject factors, and the presence of stimulus as a within-subject factor. A cell was defined as responsive if there was a significant effect of stimulus presence after performing a repeated measures analysis of variance (ranova in MATLAB). Significant cells were classified as activated if the mean activity in the response window was above baseline, or suppressed otherwise.

Orientation selectivity Index (OSI) was computed using a cross-validation method. Each cell's preferred orientation was computed from even trials; selectivity was computed as:

$$\text{OSI} = \frac{(R_{\text{pref}} - R_{\text{ortho}})}{(R_{\text{pref}} + R_{\text{ortho}})},$$

in which $R_{\text{pref}}$ is the mean response on the odd trials to the preferred orientation and $R_{\text{ortho}}$ is the mean response on the odd

trials to the orthogonal orientation (preferred orientation + 90°). This cross-validation was used because non-cross-validated selectivity indices can show large values for sparse neural activity, even if the cells are untuned. The cross-validated measure can take negative values, which indicate inconsistent responses, and will have an expected value of 0 for untuned cells.

Direction selectivity Index (DSI) was obtained similarly. Each cell's preferred direction was computed from even trials, selectivity was computed as:

$$DSI = \frac{(R_{pref} - R_{anti})}{(R_{pref} + R_{anti})},$$

in which $R_{pref}$ is the mean response on the odd trials to the preferred direction and $R_{anti}$ is the mean response on the odd trials to the direction opposite to the preferred ($R_{pref} + 180°$).

Size tuning curves and the state modulation of visual response (Fig. 4d,e and Extended Data Fig. 7c) were computed using the methods of a previous study[30]. Analysis was restricted to cells that had receptive field locations close to the centre of the grating stimuli (<20°). Size tuning curves were obtained for running and stationary states by averaging the z-scored activity of all centred cells of that type (z-scoring was computed relative to the entire drifting grating presentation). Baseline activity (shown as response to size 0 stimuli) was estimated as the average of the z-scored activity during the interstimulus intervals. For both the stimulus response and the baseline, we determined whether the mouse was running or stationary by taking the average running speed during the stimulus presentation. If this speed exceeded 1 cm s$^{-1}$, we considered the mouse as running, and stationary otherwise.

Cross-validated direction tuning curves (Fig. 4b) were computed for all cells using the average across all sizes. A cell's preferred direction was estimated as the direction providing the largest response on even trials. Direction tuning curves were computed by averaging the z-scored activity of each cell on odd trials, for each direction relative to this preferred direction. The curve was normalized by dividing by the mean response to the preferred direction (on the even trials). These normalized curves were then averaged over all cells in a subtype (Fig. 4b). The use of cross-validation means that tuning curves do not automatically have a peak at zero; for a cell with no sensory tuning the preferred direction measured on even trials would have no relationship to odd trial responses, and so the tuning curve would appear flat or random.

**Pairwise correlations between types.** To compute spontaneous correlations between the mean activity of cell groups (Figs. 3a and 5d and Extended Data Figs. 6b and 8c–e), we first normalized each cell's deconvolved activity by dividing it by its maximum. For each experiment, we then averaged the normalized activity of each cell within a group during grey-screen periods, smoothed with a 1-s boxcar window, and decimated the sampling rate to 1 Hz. When the number of cells in a type was less than 2, the correlation was not computed for that experiment. We computed the Pearson correlation between each group's mean activity and averaged over experiments. For the intra-type correlations, we randomly split the cells of each group in two halves and applied the same method, to avoid trivially obtaining a correlation of 1. When the number of cells in a type was less than 4, the correlation was not computed for that experiment.

**Response to natural images.** We summarized a cell's response to natural image stimuli with two numbers (Fig. 4d and Extended Data Fig. 7f,g). Responsiveness was defined as a modulation index between activity during the stimulus presentation period and the activity just before stimulus onset. Signal correlation was defined by correlating the responses to the first repeat of the 1,000 images with the responses to the second repeat of these same images. This metric characterizes a cell's selectivity to these image stimuli[87].

**Transcriptomic PCA.** To compute tPC1, we used the in situ gene expression of the 72 genes for each of the 1,065 unique cells that were imaged in vivo and transcriptomically identified. We performed PCA on this 72 by 1,065 matrix, and took the score of the first component to get tPC1 for each cell. To obtain tPC1 values for cells in Patch-seq (Extended Data Fig. 9b), the same weight vector was used and read counts were transformed by $\log(1 + x)$.

**Multiple linear regression using transcriptomic PCs.** To assess the fraction of variance explained by transcriptomic PCs (Extended Data Fig. 8b), we performed multiple linear regression, using leave-one-out cross-validation to quantify how well each cell's state modulation could be predicted from increasing numbers of PCs. The fraction of variance explained by this multiple linear regression was then compared to the fraction of variance explainable by a cell's subtype, type or subclass assignment, assessed again by leave-one-out cross-validation, predicting a cell's state modulation value as the average modulation of its subclass, type or subtype on the training set.

**UMAP analysis of previous scRNA-seq data.** We performed a UMAP analysis on a previous scRNA-seq dataset[3], separately for caudal ganglionic eminence (CGE) (*Vip*, *Sncg* and *Lamp5*)- and medial ganglionic eminence (MGE) (*Pvalb* and *Sst*)-derived inhibitory subclasses from V1 only (Extended Data Fig. 3).

To do so, we used methods that have previously been described for CA1[10]. First, a set of 150 genes was found using the ProMMT clustering algorithm. Then 150-dimensional expression vectors were made for each cell, applying a $\log(2 + x)$ transform to the scRNA-seq expression levels of these genes. UMAP analysis was performed using a MATLAB toolbox[88], initialized by placing the classes around a unit circle in order of similarity.

The genes automatically selected to perform the UMAP analysis were: *Vip*, *Tac2*, *Sst*, *Pdyn*, *Lamp5*, *Tac1*, *Crh*, *Calb1*, *Penk*, *Calb2*, *Th*, *Cxcl14*, *Ndnf*, *Spp1*, *Htr3a*, *Cplx3*, *Pvalb*, *Crhbp*, *Npy*, *Npy2r*, *Chodl*, *Crispld2*, *Prss23*, *Nov*, *Cbln2*, *Cartpt*, *Akr1c18*, *Atp6ap1l*, *Cadps2*, *Ppapdc1a* (also known as *Plpp4*), *Sncg*, *Tnfaip8l3*, *Unc13c*, *Pdlim3*, *Scgn*, *Pcp4*, *Tcap*, *Lgals1*, *Serpine2*, *Moxd1*, *Pthlh*, *Cd34*, *Cck*, *Sostdc1*, *Spon1*, *Gm39351*, *Mia*, *Slc5a7*, *Pde1a*, *Adarb2*, *Mybpc1*, *Car4*, *Cbln4*, *Gabrg1*, *Fmo1*, *Slc18a3*, *Grpr*, *Lypd6*, *Pde11a*, *Rxfp1*, *Tnnt1*, *Nxph2*, *Lpl*, *Cryab*, *Cp*, *Npy1r*, *Id3*, *Myl1*, *Id2*, *Kit*, *Serpinf1*, *Bcar3*, *Aqp5*, *Scrg1*, *Gpd1*, *Rxfp3*, *Prox1*, *Col25a1*, *Chat*, *Vwc2l*, *Amigo2*, *Myh8*, *Synpr*, *Grm8*, *Igfbp5*, *Gpx3*, *Rgs12*, *Lypd1*, *Cd24a*, *Reln*, *Hapln1*, *Sln*, *Chrm2*, *Ostn*, *Igfbp7*, *Prox1os*, *Atf3*, *Lect1*, *Gpc3*, *Ptprk*, *Teddm3*, *Il1rapl2*, *Col6a1*, *Nek7*, *Crispld1*, *Wif1*, *Wnt5a*, *Bmp3*, *Thrsp*, *Syt2*, *Pcdh20*, *Sfrp2*, *Myh13*, *Efemp1*, *Rprm*, *Cacna2d1*, *Lypd6b*, *Meis2*, *Lhx6*, *Angpt1*, *Rspo1*, *Sema3c*, *Itih5*, *Nfix*, *Sema3a*, *Stk32a*, *Ecel1*, *Jam2*, *Igfbp6*, *Sox6*, *Nfib*, *Sall1*, *Sema5b*, *Shisa8*, *Tacr3*, *Chst7*, *Frmd7*, *Gm31465*, *Rspo4*, *Chrna2*, *Lmo1*, *C1qtnf7*, *Ndst4*, *Ccdc109b* (also known as *Mcub*), *Npas1*, *Egfr*, *S100a10*, *Gpr6*, *Slit2*, *Lsp1*.

**Correlation with electrophysiological and morphological properties.** We examined electrophysiological and morphological correlates of our results by relating them to a previously published Patch-seq dataset[7], which provided electrophysiological, morphological and gene expression data from a set of V1 inhibitory cells analysed in vitro. These cells had been genetically assigned to the same transcriptomic clusters that we used[3], which allowed us to correlate electrophysiological and morphological properties to the state modulation measured in our own dataset. Valid electrophysiological recordings were available for 4,391 cells and included long and short pulses of current injection as well as current ramps. We used the electrophysiological parameters calculated by the original authors using the ipfx software, renaming 'up/down ratio' (the absolute ratio of the slopes of the upward and downward components of the action potential) as 'spike shape index'. Adaptation index was the rate at which spiking changed during a long depolarizing

square stimulus. During a hyperpolarizing square current, the membrane time constant tau is the rate of approach of steady state, and sag is the downward deflection before steady state is reached. Capacitance was calculated as the ratio between measured tau and resistance.

We quantified the ratio of axon in each layer using morphological reconstructions obtained following Patch-seq. To enable comparison to our two-photon data, we only examined reconstructed cells with somas in L1–3 that belonged to one of the 35 subtypes we recorded from, for a total of 163 cells. Morphology was represented as an acyclic undirected graph with a position and radius associated with each node. A pair of adjacent nodes (a segment) fell within a layer if both nodes had cortical depths within the layer boundary. Segments that fell on a layer boundary (less than 4% of segments for each cell) were not classified into a layer, and segments that entered the white matter or pia were excluded. The surface area of all within-layer segments was computed using the distance between nodes and their radii. The within-layer surface area ratio is the sum of the surface area of segments within a layer divided by the total surface area of all segments.

tPC1 was computed for each Patch-seq cell using the same 72 genes and weightings found from our coppaFISH data, with gene expression transformed as $\log(1 + x)$.

**Processing of eye video (pupil detection).** Eye videos were processed using facemap (https://github.com/MouseLand/facemap). An ROI was drawn manually around the pupil of the mouse. The pupil area was defined as the area of a Gaussian fit on thresholded pupil frames, in which pixels outside the pupil were set to zero.

### Statistical analyses

Statistical analysis of differences between cell types faces two potential confounds. First, different experiments will by chance record different proportions of each cell type, and may also by chance show other experiment-to-experiment differences such as overall alertness levels. Second, the large number of subtypes presents a potential multiple comparisons problem.

To solve these problems, we devised a nested permutation test. First, an omnibus test asks whether subclass, type and subtype have a significant main effect on our quantity of interest $y$; there is no multiple comparisons problem for this omnibus test, and all shuffling occurs within an experiment to avoid conflating experiment-to-experiment variability with differences between cell types. The omnibus test is conducted at each of the three levels in a nested manner (Extended Data Fig. 6a): the first asks whether there is a main effect of subclass; the second whether there is a main effect of type beyond that predicted by subclass; and the third whether there is a main effect subtype beyond that predicted by type. After the omnibus test, post-hoc tests are used to ask whether significant differences between types exist within each individual subclass, and whether significant differences between subtypes exist within each individual type (Extended Data Fig. 6a). Additional post-hoc tests are used to ask whether the quantity is significantly different to zero for each subclass, type and subtype. All post-hoc tests are corrected for multiple comparisons using the Benjamini–Hochberg procedure.

To test for a main effect of subclass on a quantity $y$, the omnibus test computes its mean value of $\bar{y}_f$ for each subclass $f$, and uses as test statistic the variance of $\bar{y}_f$ across subclasses. To obtain a $P$ value, this test statistic is compared to a null ensemble obtained after 10,000 random shufflings of the subclass label of each cell, separately within each experiment. To test for a main effect of type, we compute the mean $\bar{y}_c$ of $y$ for each type $c$, and use as test statistic the variance of this mean across types. A null distribution is obtained by 10,000 shufflings of type labels separately within each experiment and subclass. To test for a main effect of subtype, we use as test statistic the variance of $\bar{y}_s$ over subtypes $s$. A null distribution is obtained by recomputing this statistic after shuffling subtype labels 10,000 times, separately within each type and experiment.

To perform the post-hoc test for significant differences between the types within a specific subclass (indicated by $P$ values on the far right of Fig. 3b and similar), or for significant differences between subtypes within a specific type (indicated by stars second to right in Fig. 3b and similar), we performed the same shuffle test inside individual subclasses and types. For example, to obtain the $P$ value for significant differences of subtypes within the *Pvalb-Tac1* type, we used as test statistic the variance of $\bar{y}_s$ across the 5 subtypes inside this type, and compared it to 10,000 shufflings of the subtype labels inside this same type. These post-hoc $P$ values were then corrected using the Benjamini–Hochberg procedure. For post-hoc tests of whether a subclass, type or subtype is significantly different to zero, we used Benjamini–Hochberg-corrected $t$-tests.

For the nested permutation test on pairwise correlations (Fig. 3a and Extended Data Fig. 6b), we used the same shuffling procedure, using as test statistic the difference between the mean of intra-group correlations and the mean of inter-group correlations across all experiments and cell groups.

All $P$ values for the nested permutation test are one-tailed.

For linear correlations (Figs. 1j, 3e and 6a,b and Extended Data Figs. 6e and 9a,b), we show the $P$ value for the Pearson correlation coefficient. To exclude the possibility of conflating experiment-to-experiment variability with differences between cell types, we used ANCOVA controlling for a discrete effect of recording session (Figs. 3c,d and 5c and Extended Data Fig. 8a) quoting the significance of a main effect of the continuous variable. ANCOVA was also used to test whether a continuous transcriptomic variable assigned to each cell correlated significantly with state modulation even after controlling for subtype and recording session (Fig. 3e and Extended Data Fig. 6e) and for subclass and recording session (Fig. 5c), and whether cortical depths of each subtype measured by coppaFISH and Patch-seq were correlated even within a subclass or type (Fig. 1j).

To test for the effect of tPC1 on pairwise correlations (Fig. 5d and Extended Data Fig. 8c–e), we sorted types by tPC1 and computed their pairwise correlation matrix as described above. We used a permutation test to ask whether values close to the diagonal were larger than values far from the diagonal. As test statistic we used the difference between the mean correlation values one or two steps away from the diagonal, and the mean of all other type pairs (Extended Data Fig. 8f). We constructed a null distribution by recomputing this statistic after permuting the order of the types 10,000 times. Again, the $P$ values are one-tailed for this shuffling test.

### Box plots

To show the distributions of physiological properties within a cell population, we used box plots (Figs. 3b and 4c,d and Extended Data Figs. 6c,d and 7a–g). In these plots, the central black circles represent the median; the left and right edges of boxes represent the 25th and 75th percentiles; and the whiskers extend to the most extreme data points (excluding outliers more than 1.5 times the interquartile range from the box, which are plotted as small black dots).

### Statistics and reproducibility

The cell shown in Extended Data Fig. 1c is a representative example of the 1,090 cells that were recorded and processed with coppaFISH. The registration example shown in Fig. 1b–e is a representative example of the 99 slices (over $n = 4$ mice) that we have successfully aligned to the in vivo $z$-stacks. The nine cells shown in Extended Data Fig. 4a,b are representative examples of the 117 molecularly identified cells recorded in the same session as shown in Fig. 2. All experiments were performed on four independent mice, and the results were reliably replicated across all mice. We did not use statistical methods to pre-determine sample sizes. However, our sample sizes are similar to those reported in previous publications using a similar approach[25,26]. Our study did not contain experimental groups, so randomization and blinding do not apply.

**Reporting summary**
Further information on research design is available in the Nature Research Reporting Summary linked to this paper.

**Data availability**
Processed data (calcium activity traces, gene detections and so on) are available at https://doi.org/10.6084/m9.figshare.19448531. The raw data (two-photon movies, transcriptomic images and so on) will be made available upon reasonable request. Natural scenes were obtained from the ImageNet database (https://www.image-net.org/). Source data are provided with this paper.

**Code availability**
Code for analysis of in situ transcriptomic data analysis is available at https://github.com/jduffield65/iss; and for registration of in vivo and ex vivo slices at https://github.com/ha-ha-ha-han/NeuromicsCellDetection and https://github.com/sbugeon/NeuromicsCellDetection. Custom code written in MATLAB (R2019b) to analyse and plot the processed data is available upon reasonable request to the authors. This research used the freely available packages Suite2P (https://github.com/cortex-lab/Suite2P), Kilroy (https://github.com/ZhuangLab/storm-control, with our modifications available at https://github.com/acycliq/storm-control) and Facemap (https://github.com/MouseLand/facemap).

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

**Acknowledgements** We thank T. Hauling for work developing the coppaFISH method; M. Nilsson and X. Qian for advice with in situ transcriptomics; N. Gouwens for help with Patch-seq data; F. Rossi for discussion; M. Krumin for support with microscopy; and L. Funnell for help with histology. This work was supported by the Wellcome Trust (108726, 205093, 090843), the European Union's Marie Skłodowska-Curie program (835489), the Royal Society (NIF\R1\180184) the Chan-Zuckerberg initiative (2018-182811) and the Gatsby Charitable Foundation (GAT3361). M.C. holds the GlaxoSmithKline/Fight for Sight Chair in Visual Neuroscience.

**Author contributions** S.B., M.C. and K.D.H. conceived and designed the experiments. S.B., J.D., H.P., I.P., A.R., H.F., Y.I. and K.D.H. refined the techniques. S.B., M.D., C.B.R., D.O. and A.R. performed the experiments. S.B., J.D., M.D., Y.I., D.N., H.P. and K.D.H. wrote the software for data analysis. S.B., M.D., M.S., A.V. and K.D.H. analysed the data. S.B., M.D., M.C. and K.D.H. made the figures. S.B., J.D., I.P., A.R., Y.I., M.C. and K.D.H. wrote the manuscript with input from M.D., H.P. and M.S. All work was supervised by M.C. and K.D.H.

**Competing interests** The authors declare no competing interests.

**Additional information**
**Correspondence and requests for materials** should be addressed to Stéphane Bugeon or Kenneth D. Harris.

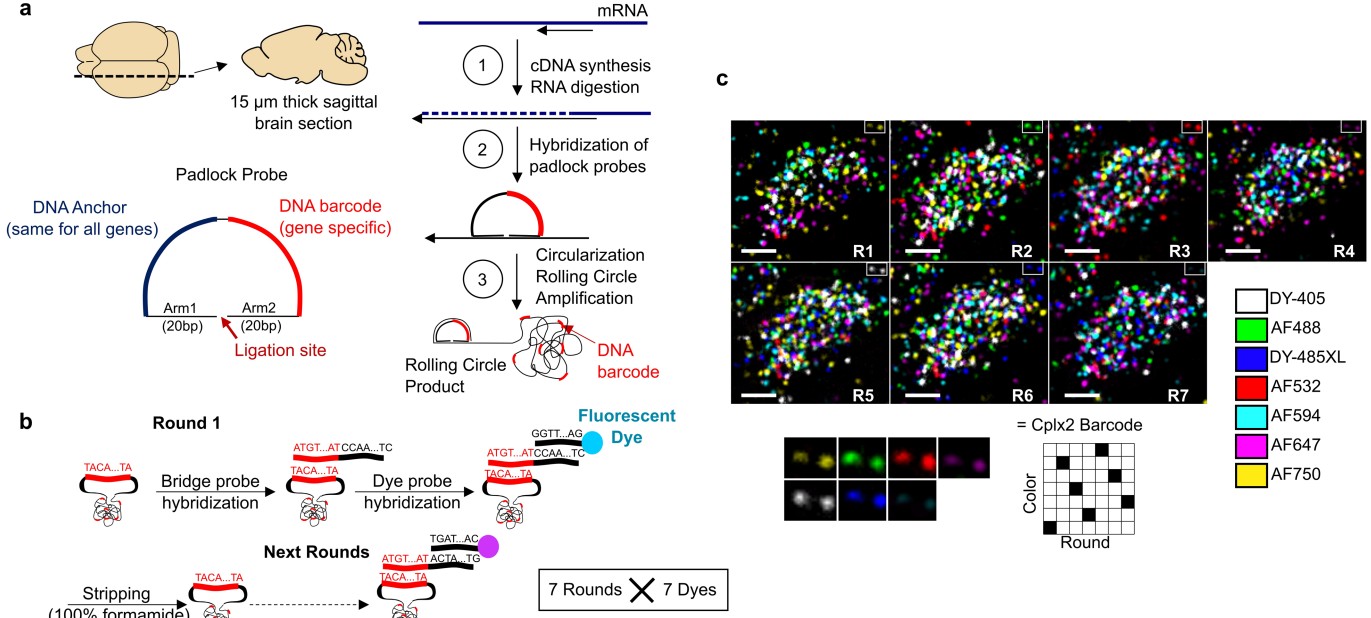

**Extended Data Fig. 1 | Detection of 72 genes using coppaFISH. a**, Sagittal 15 μm brain sections are cut using a cryostat. Local mRNAs are reverse transcribed to cDNA, and the mRNAs digested to free the cDNAs for hybridization with padlock probes. Padlock probes have two 15-20-nt arms complementary to the target site, a 20-nt anchor sequence (identical for all probes) and a 20-nt barcode sequence (unique for each gene). After hybridization to the target site, a DNA ligase enzyme circularizes the padlock probe, but only when it matches the target perfectly. Next, a DNA polymerase enzyme amplifies the circularized padlock probes, producing rolling-circle products (RCPs), which contain many repeats of the padlock sequence including the barcode. **b**, The barcodes are read out by 7 rounds of 7-colour fluorescence imaging. On each round, RCPs are hybridized with custom designed bridge probes, which in turn hybridize to specific dye probes (conjugated to one of 7 fluorophores). The sections are then imaged in 7 colour channels, then all DNA is removed with formamide treatment, and the next round begins. Different sets of bridge probes on each round result in each barcode showing up in a different colour channel using a Reed–Solomon code for minimum overlap. After the 7 combinatorial rounds, a final round images the anchor probe (used for image alignment) and DAPI to visualize cell nuclei. **c**, Example raw data for one cell imaged with the 7 fluorophores and 7 rounds. Each fluorescent spot is an RCP, and the sequence of colours across 7 rounds allows gene identity to be determined. Bottom: magnification of 2 RCPs (top right corner of main images) which corresponded to Cplx2 barcode (6135024). Scale bars: 5 μm.

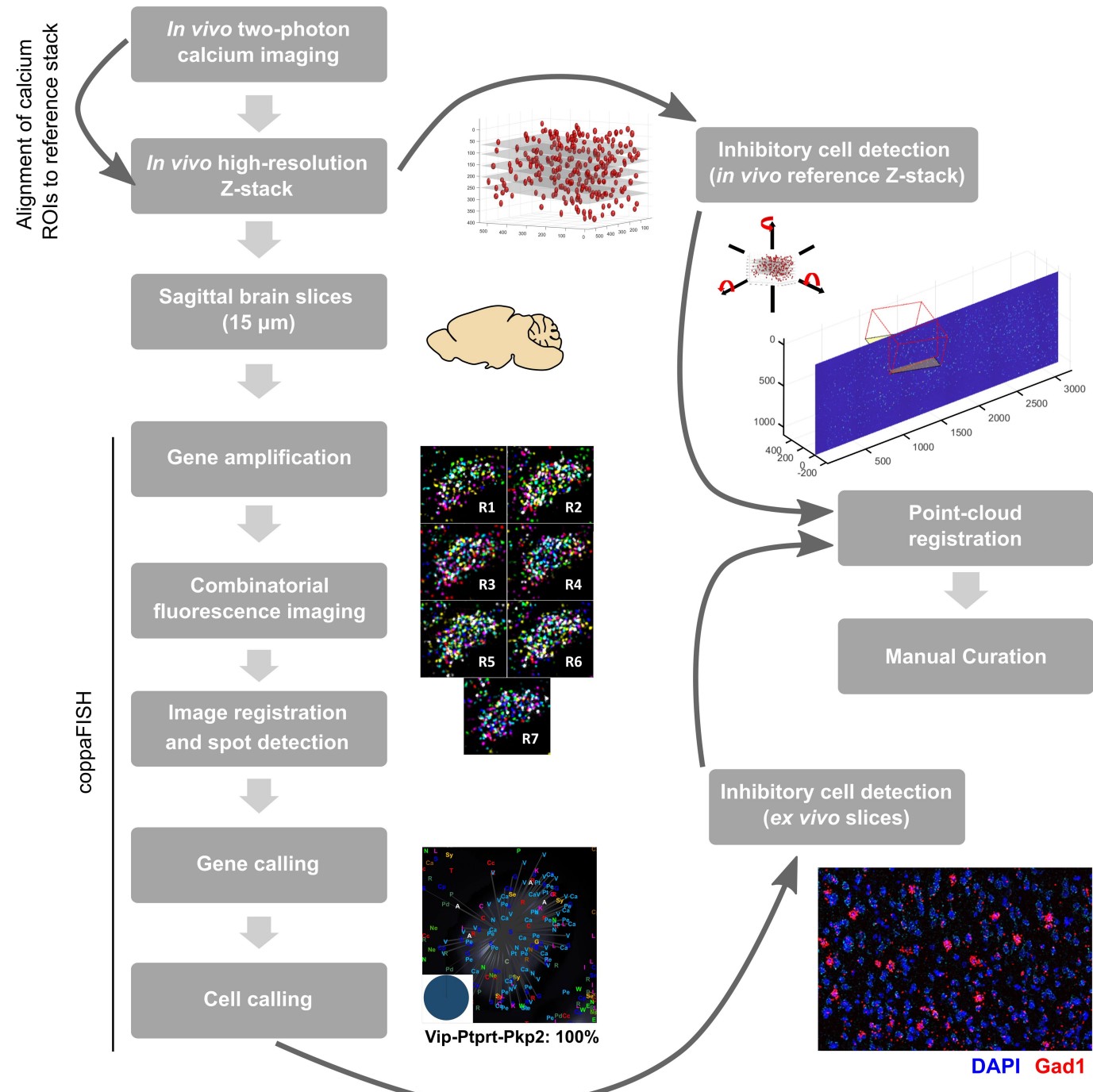

**Extended Data Fig. 2 | Experimental pipeline.** Neural activity was recorded *in vivo* over multiple sessions from each subject (Gad2-mCherry mice with viral GCaMP6m expression in all neurons). At the end of each session, a high-resolution reference Z-Stack was acquired and used to detect interneurons in the Z-stack volume using mCherry fluorescence, and cells recorded during calcium imaging were registered to this Z-Stack. After all imaging sessions, the brain was extracted from the skull without fixation and frozen in OCT. A block from under the imaging window was sliced into 15 µm sagittal sections, which were thaw-mounted on gelatine-coated coverslips. Each section was then processed using coppaFISH:

RCPs were produced *in situ* for the selected genes, and their barcodes were read using 7 rounds of imaging (+1 round of anchor and DAPI staining). The resulting images were then registered across rounds, colour channels, and image tiles and individual spots detected. Gene identity for each RCP was decoded from the 49-dimensional images, and pciSeq[28] was used to determine the subtype probabilities for each cell. To align the images, interneurons detected *in vivo* and *ex vivo* were used as fiducial markers for point cloud registration, which finds the best alignment of the 2D *ex vivo* slice in the 3D volume. Finally, individual cell matches were manually curated, and a subtype assigned to the recorded cells.

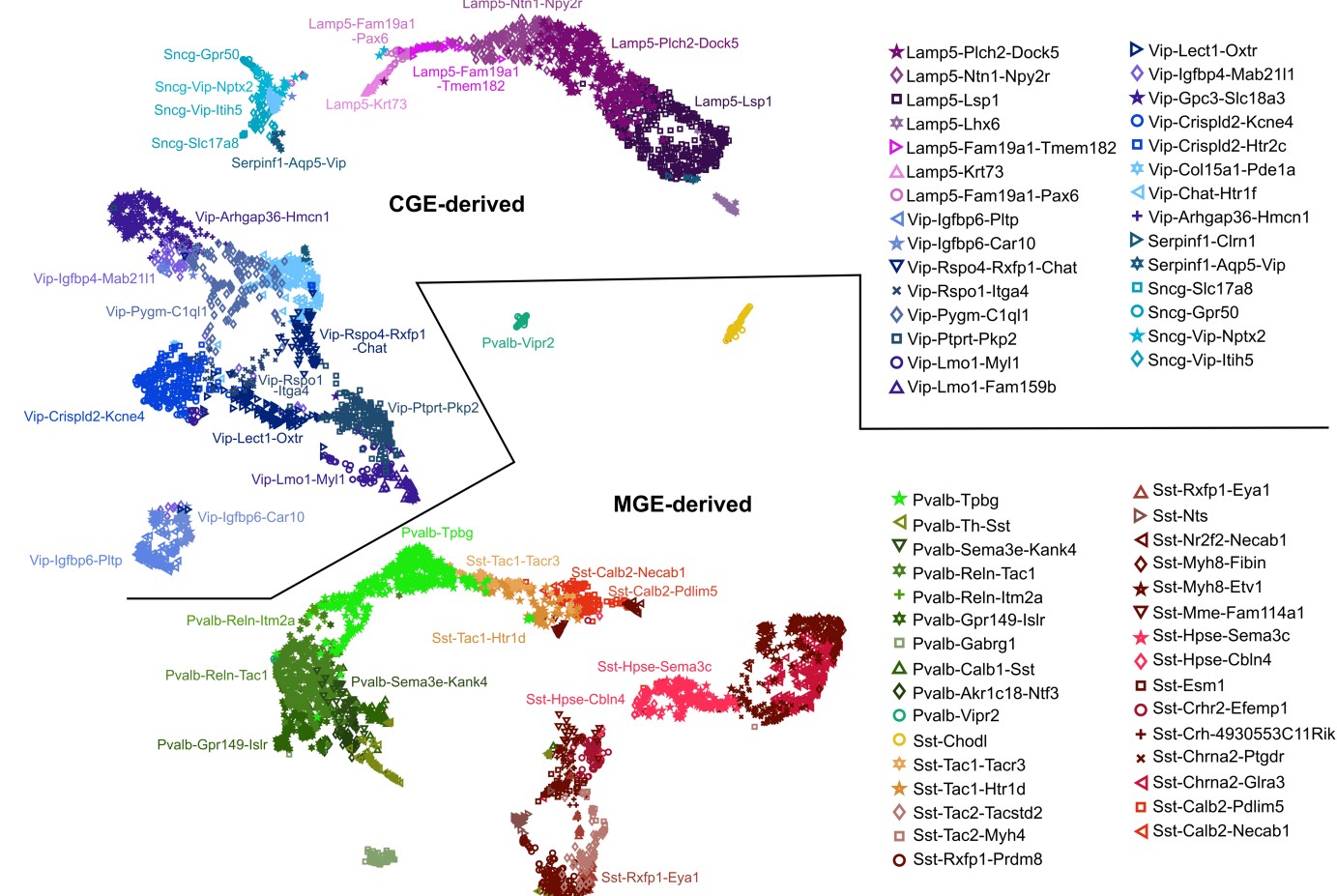

**Extended Data Fig. 3 | UMAP analysis of scRNA-seq data.** Each dot represents a V1 inhibitory cell, from the Tasic et al.[3] data, with glyph representing its assigned subtype. UMAP analysis was performed separately for MGE and CGE derived interneuron subtypes, using 150 log-transformed genes selected by the ProMMT algorithm[10]. This analysis reveals both highly discrete subtypes such as Pvalb-Vipr2 (putative chandelier cells) and smoothly varying continua where boundaries between subtypes appear arbitrary, such as Lamp5-Ntn-Npy2r, Lamp5-Plch2-Dock5, and Lamp5-Lsp1 (putative neurogliaform subtypes). Also note the smooth transition between Sst-Calb2 (putative Martinotti subtypes), Sst-Tac1 (putative Sst non-Martinotti), and Pvalb-Tpbg (putative superficial basket cell subtypes). Text on main plots indicates location of *in vivo* imaged subtypes.

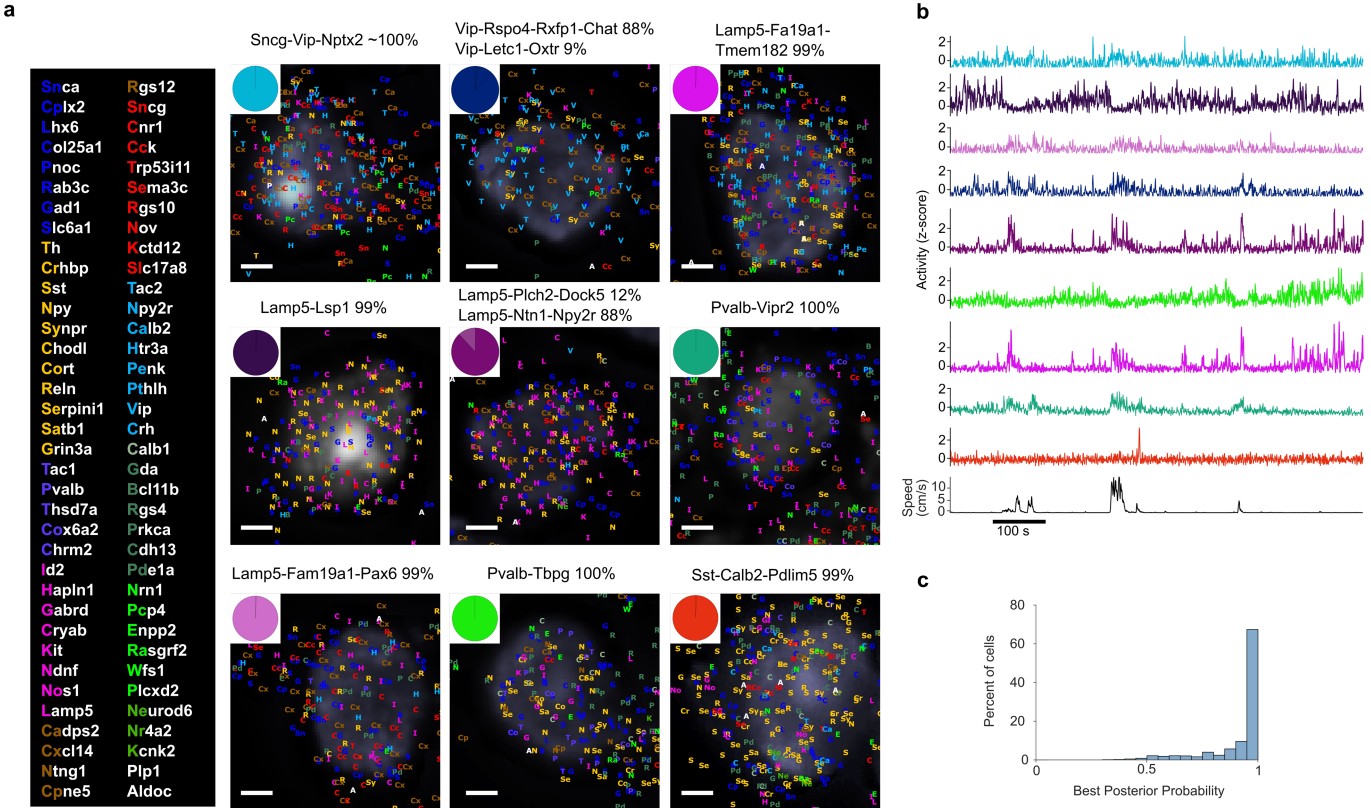

**Extended Data Fig. 4 | Example cells. a**, Nine example cells which were recorded during the same session as in Fig. 2. Pie plots indicate the posterior probabilities of each cell's subtype assignment. Grey background images show DAPI-stained nuclei. Each gene detection is represented by coloured letters (key to the left). Scale bars: 2 μm. **b**, Activity of these 9 cells during spontaneous behaviour, together with the running speed of the mouse. The traces are colour coded according to the assigned subtype for each cell (pie plots in **a**). **c**, Analysis of Bayesian classification confidence. Histogram shows posterior probability for a cell to belong to its assigned subtype, for *in vivo* imaged cells. About 2% of cells for which confidence was below 50% were not analysed further.

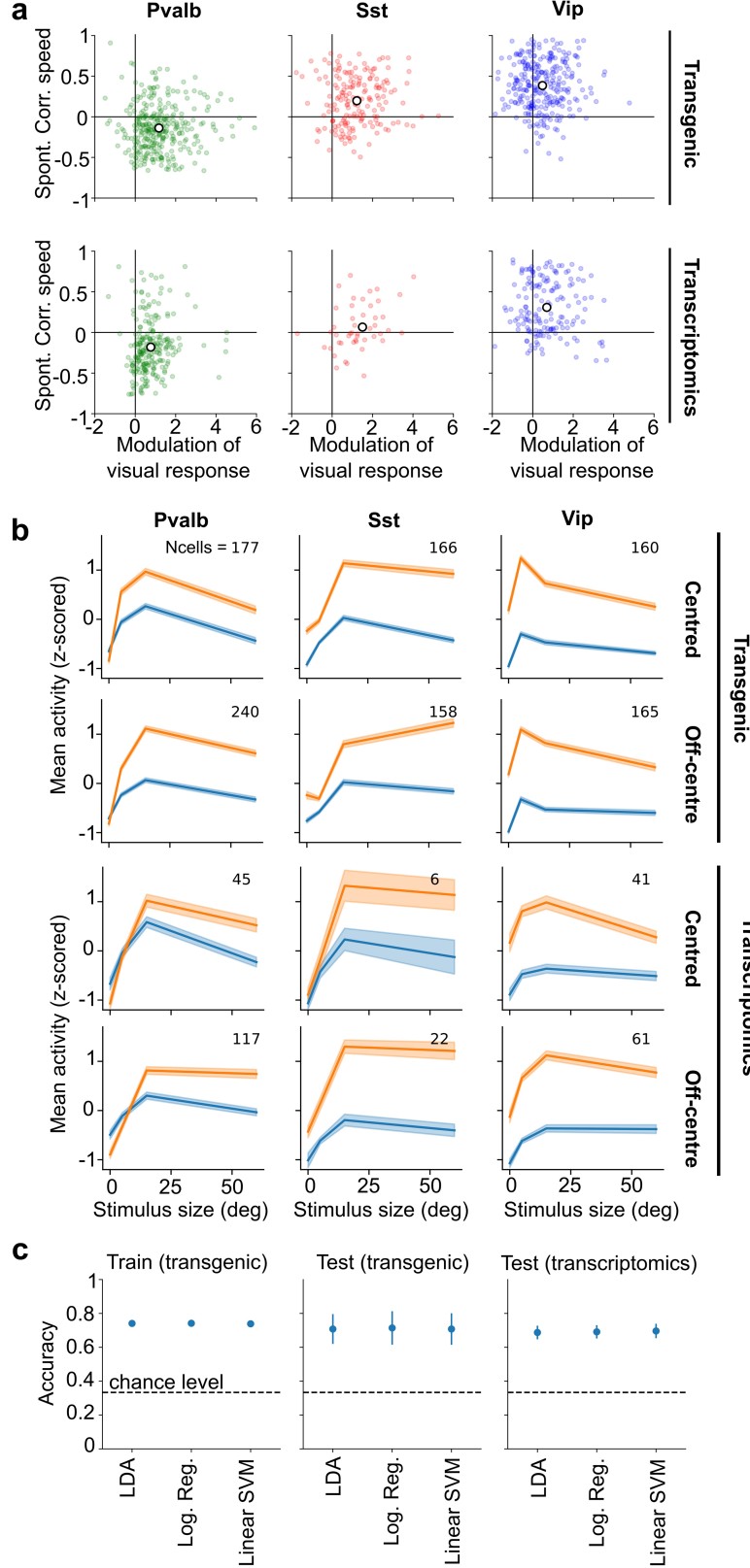

**Extended Data Fig. 5 |** See next page for caption.

**Extended Data Fig. 5 | Comparison with results in transgenic mice. a**, Top row: modulation of visual responses by running vs. correlation to running speed during spontaneous behaviour, for Pvalb, Sst, and Vip interneurons identified in transgenic mouse lines. Data re-analysed from Ref. [30] and including 4 new mice. Bottom row: same analysis using interneurons identified by post-hoc transcriptomic analysis (data from this study; the Vip group included Vip-positive Sncg cells which are likely to be labelled in the Vip-Cre transgenic line). **b**, Size tuning curves of Vip, Pvalb and Sst cells for both datasets. Top row: responses measured in transgenic mice for centred stimuli (0-10° offset from receptive field centre); second row: response to off-centre stimuli (10-20° offset from receptive field) in transgenic mice; bottom two rows, same from post-hoc transcriptomics. Orange curves: responses during running; blue curves, responses during stationary epochs. Numbers at the top right corner of each plot indicate number of cells. Data are given as mean ± s.e.m. **c**, Classification of cell type from physiological features was identical for the two cell typing methods. Each cell was assigned to either Sst, Pvalb or Vip based on 14 physiological features (such as correlation to running speed, size tuning curves, skewness), using one of 3 different linear classifiers trained on a training set randomly selected from the transgenic recording sessions. Left: training-set classifier accuracy averaged over multiple random selections of the training set. Centre: accuracy of the classifiers averaged over the held-out transgenic sessions (test sets). Right: out-of-sample accuracy of the linear models on data with interneurons identified by post-hoc transcriptomics. Note the similar performance on transgenic and transcriptomic test sets. Error bars: s.d. over divisions into training and test set.

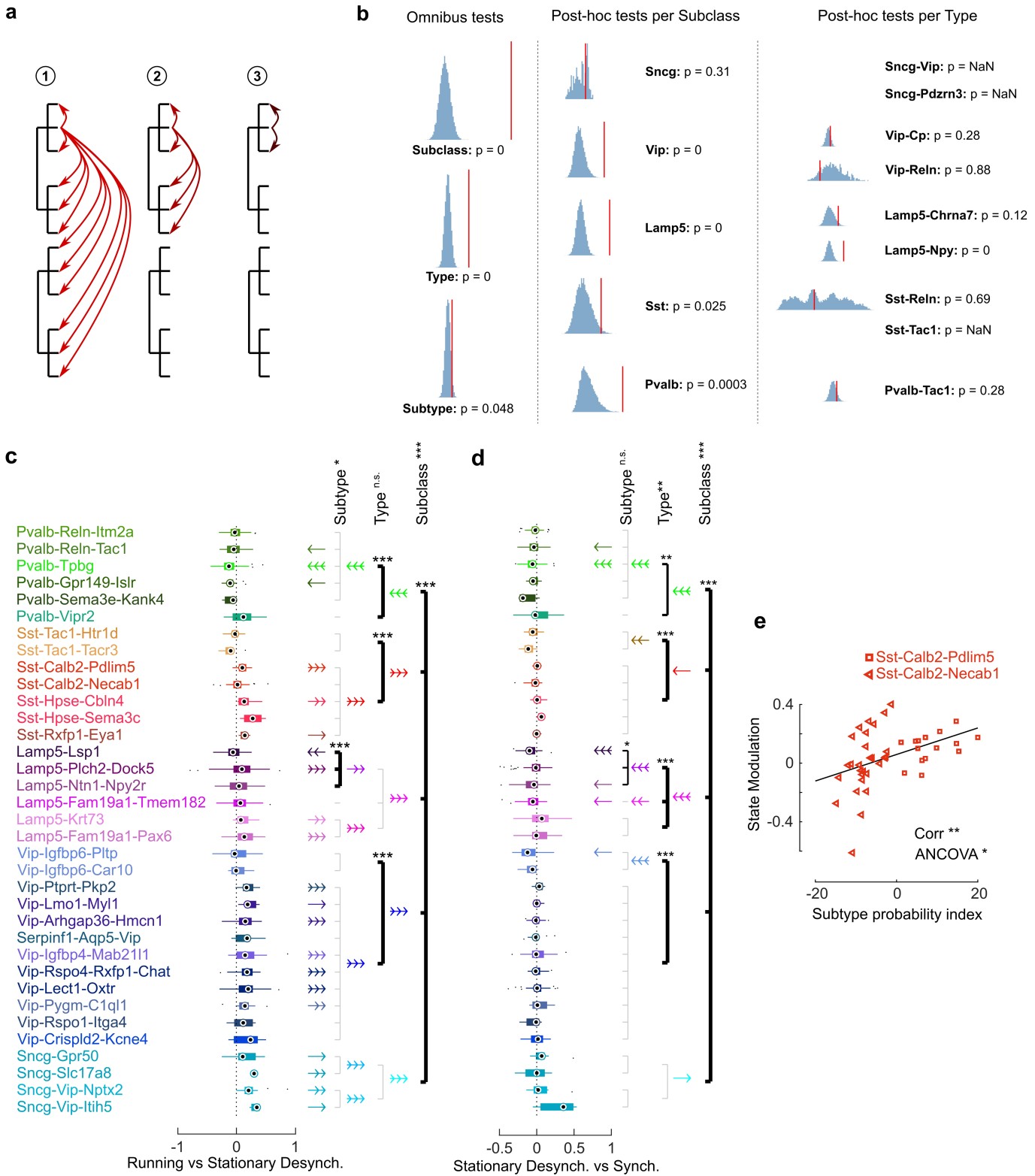

**Extended Data Fig. 6** | See next page for caption.

**Extended Data Fig. 6 | Further analyses of state modulation during spontaneous behaviour. a**, Illustration of nested permutation test method. The test asks whether a quantity of interest differs significantly between cell groups, at each level of the classification hierarchy: whether it differs between subclasses, between types belonging to a single subclass, and between subtypes belonging to a single type. To use the test, one first computes a test statistic (such as the mean correlation coefficient of cells assigned to the same group). To obtain a p-value, this test statistic is compared to a null distribution obtained by shuffling the cells' transcriptomic labels within the appropriate hierarchical level, in a one-sided manner. To test for a difference between the top-level subclasses, cells are shuffled without restriction (1). To test for a difference between types within subclasses, cells are shuffled separately within each subclass (2). To test for a difference between subtypes within types, cells are shuffled separately within each type (3). In all three cases, cells are only shuffled within experiments, to avoid conflating variability between experiments with variability between cell types. **b**, Nested permutation test results for pairwise spontaneous correlations. Left: blue histograms represent the probability of observation obtained by shuffling transcriptomic labels 10,000 times at three hierarchical levels (see **a**). Red lines: observed value of the test statistic. Middle: post-hoc tests for each subclass. Right: post-hoc tests for each type containing at least 2 subtypes. All post-hoc p-values were adjusted with Benjamini–Hochberg correction for multiple comparisons. **c**, Nested permutation analysis of modulation between running and stationary desynchronized states, plotted as in Fig. 3b. Top: significance of omnibus test for differences between subclasses (p < 0.0001) and nested types (p = 0.21) and subtypes (p = 0.038). Post-hoc p-values were adjusted with Benjamini–Hochberg correction (Pvalb: p < 0.0001, Sst: p < 0.0001, Vip: p < 0.0001, Lamp5-Npy: p < 0.0001) **d**, Nested permutation analysis of modulation between stationary desynchronized and stationary synchronized states, plotted as in Fig. 3b. Top: significance of omnibus test for differences between subclasses (p < 0.0001) and nested types (p = 0.007) and subtypes (p = 0.088). Post-hoc p-values were adjusted with Benjamini–Hochberg correction (Pvalb: p = 0.0075, Sst: p < 0.0001, Vip: p < 0.0001, Lamp5: p < 0.0001 Lamp5-Npy: p = 0.04) **e**, State modulation vs. subtype probability index for Sst-Calb2-Necab1 and Sst-Calb2-Pdlim5 cells, plotted as in Fig. 3e (Pearson correlation: r = 0.43, p = 0.005; ANCOVA accounting for effects of subtype: F(1) = 7.3, p = 0.011). *, p < 0.05, **, p < 0.01, ***, p < 0.001; 1, 2, or 3-headed arrows in **c** and **d** indicate same significance levels, direction indicates the sign of modulation.

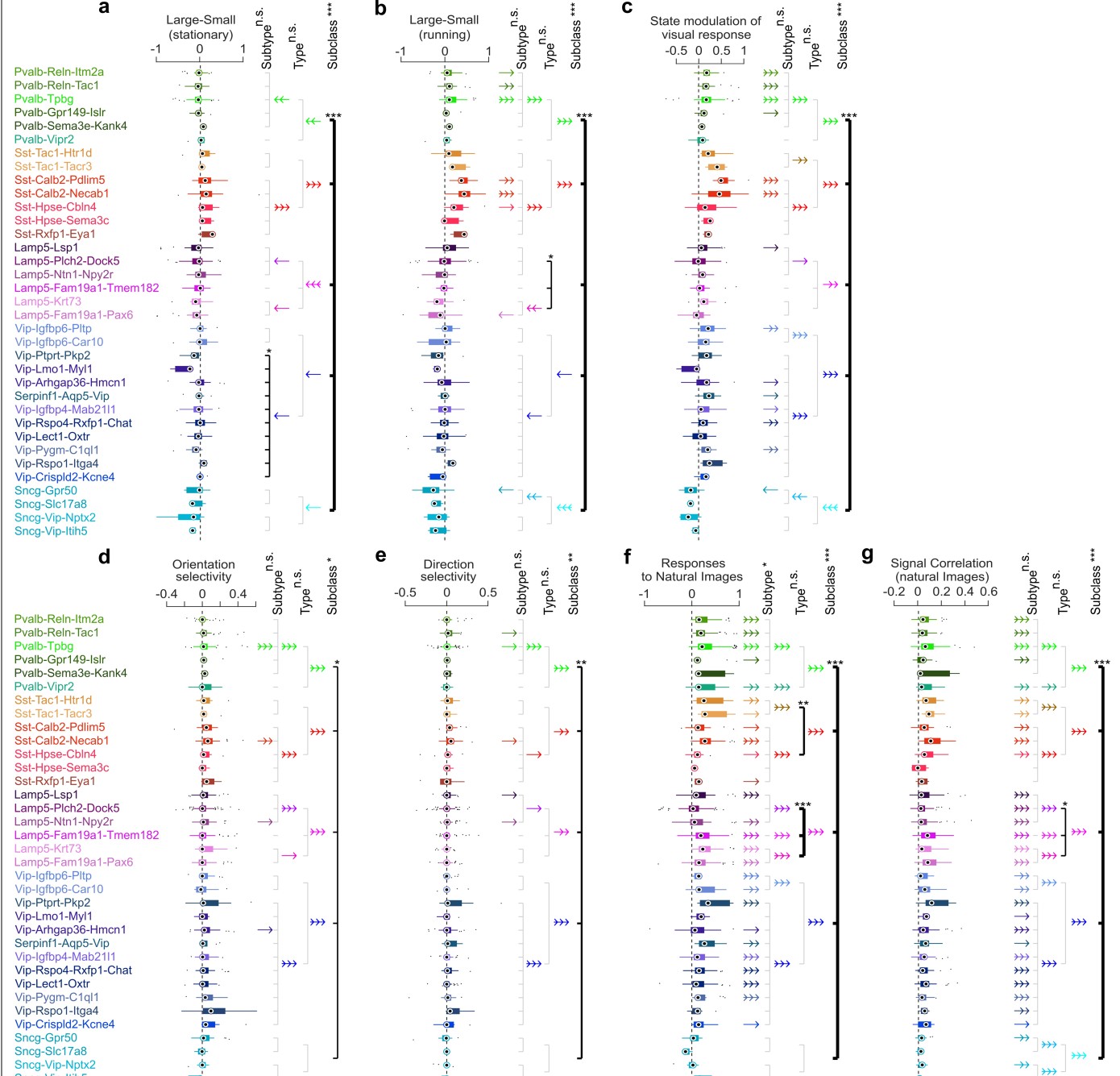

**Extended Data Fig. 7 | Further analyses of visual responses.** Each panel shows a nested permutation analysis for the visual variables analysed in Fig. 4d, but extended to the subtype level. All panels plotted as in Fig. 3b. Post-hoc comparisons of multiple cell groupings are Benjamini–Hochberg corrected within each of these plots. Omnibus and post-hoc tests hierarchical permutation tests are one-sided, one-sample post-hoc t-tests are two-sided. **a**, Response differences between large and small gratings in stationary periods (subclass: p < 0.0001; Post-hoc tests: Vip-Cp: p = 0.036). **b**, Response differences between large and small gratings during running (subclass: p < 0.0001; Post-hoc tests:

Lamp5: p = 0.04). **c**, State modulation of visual response by running, averaged over all sizes (subclass: p < 0.0001). **d**, Orientation selectivity (subclass: p = 0.02). **e**, Direction selectivity (subclass: p = 0.001). **f**, Mean response for natural image stimuli (subclass: p < 0.0001; subtype: p = 0.037; Post-hoc tests: Lamp5: p= < 0.0001, Sst: p = 0.01). **g**, Reliability (signal correlation) for natural image stimuli (subclass: p < 0.0001; Post-hoc tests: Lamp5: p = 0.015). *, p < 0.05, **, p < 0.01, ***, p < 0.001; 1, 2, or 3-headed arrows indicate same significance levels, direction indicates the sign of modulation.

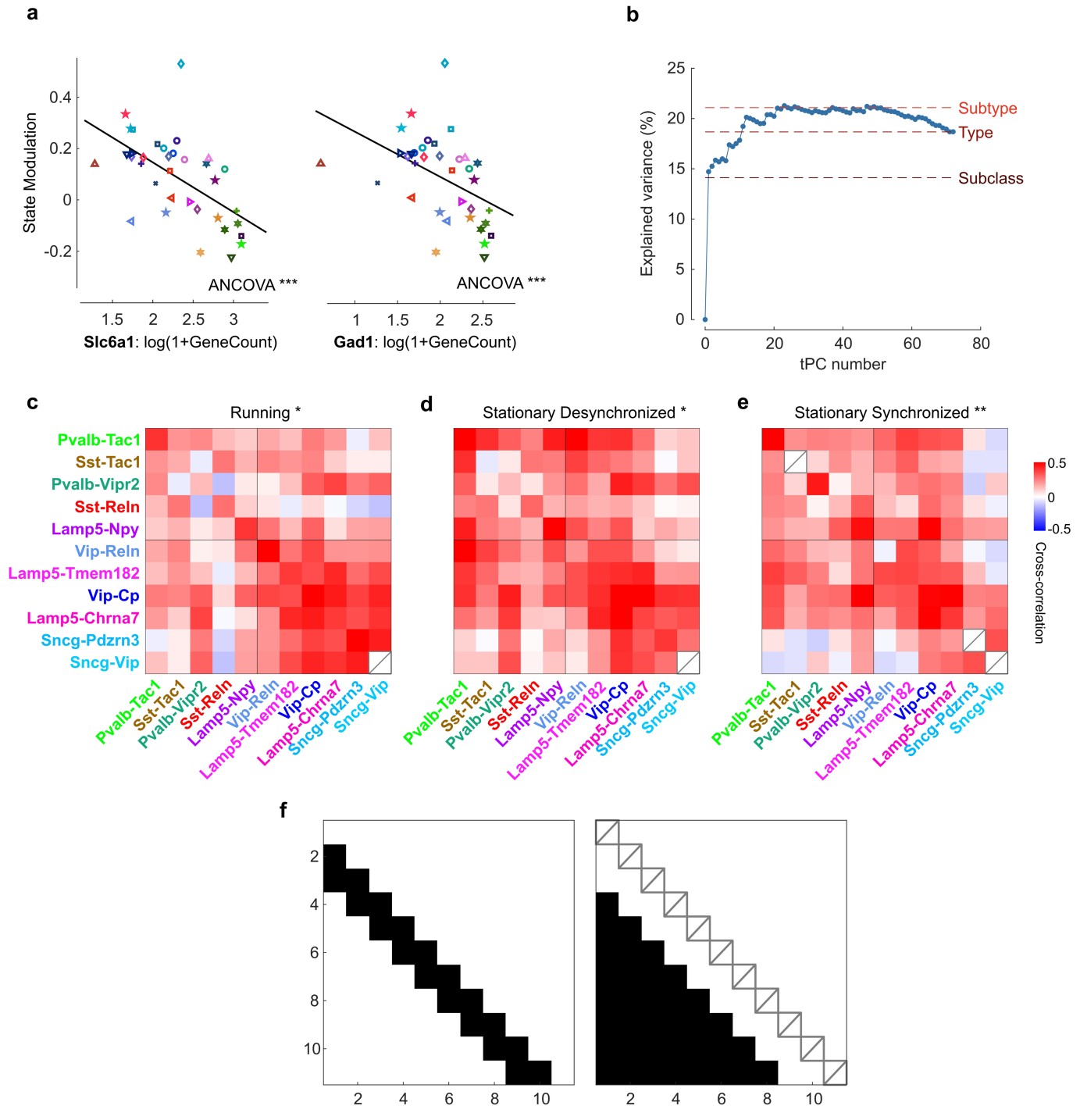

**Extended Data Fig. 8 | Additional analyses of the relationship between tPC1 and state modulation or pairwise correlations. a**, Correlation of state modulation with natural log expression of two individual genes, Slc6a1 and Gad1, plotted as in Fig. 5c. (ANCOVA controlling for session, F(1) = 138.2, p = 6×10$^{-30}$; F(1) = 50.7, p = 2×10$^{-12}$, respectively) **b**, Variance fraction of a cell's state modulation explainable by successive transcriptomic dimensions. Blue points: fraction of cross-validated variance explainable by multiple linear regression from successive transcriptomic PCs. Dashed lines: fraction of variance explainable by discrete classification according to a cell's subtype, type or subclass assignment. The first transcriptomic PC explains respectively 70%, 79% and 108% of the variance explainable by subtype, type and subclass

assignment. **c**, **d**, **e**, Pairwise correlations between simultaneously recorded types, plotted as in Fig. 5d, but separately for periods within each of the three states (running, stationary desynchronized, and stationary synchronized). The types are sorted by tPC1; types with similar tPC1 values have significantly higher correlations (one-sided permutation test, p = 0.025, p = 0.038, p = 0.005 respectively). **f**, The permutation test showing higher correlations amongst cells of similar tPC1 used as test statistic the difference between the average of correlation coefficients close to the diagonal (left), and the average of all other off-diagonal coefficients; intra-type correlations were not used. This test statistic was compared to a null ensemble obtained after shuffling tPC1 values 10,000 times. *, p < 0.05, **, p < 0.01, ***, p < 0.001.

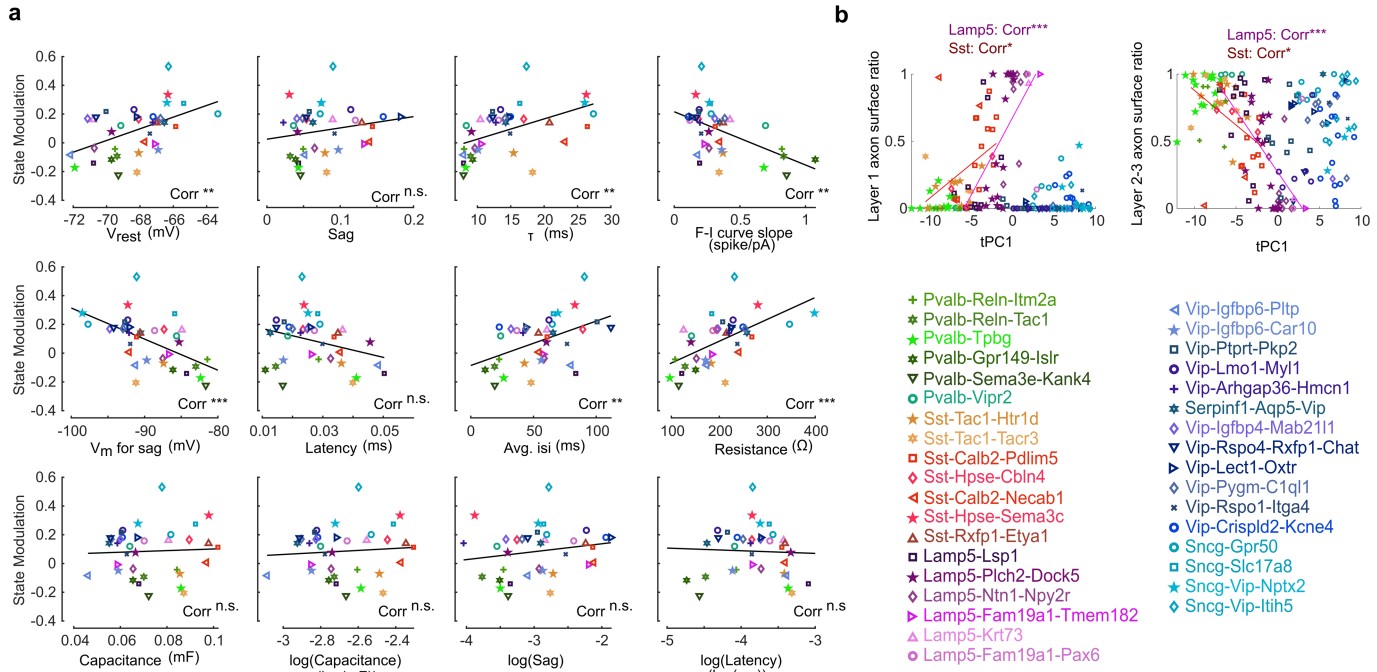

**Extended Data Fig. 9 | Additional analyses of Patch-seq data. a**, Additional electrophysiological properties vs. State modulation plotted as in Fig. 6a. $V_{rest}$: r = 0.49 p = 0.003, Sag: r = 0.19 p = 0.27, τ: r = 0.5 p = 0.002, F-I curve slope: r = 0.53 p = 0.001, $V_m$ for Sag: r = -0.56 p = 5×10$^{-4}$, Latency: r = -0.31 p = 0.07, Avg. isi (inter-spike interval): r = 0.46 p = 0.005, Resistance: r = 0.59 p = 2×10$^{-4}$, Capacitance: r = 0.05 p = 0.78, log(Capacitance) : r = 0.1 p = 0.64, log(Sag) : r = 0.19 p = 0.3 and log(Latency): r = -0.04 p = 0.8. Stars show significance assessed by Pearson correlation (two-sided tests). Black lines are linear fits.

**b**, Fraction of axonal arborization (measured by surface area) in layer 1 (left) and layer 2-3 (right) vs. tPC1 computed for each Patch-seq neuron. Each symbol represents a cell. Pearson correlation (two-sided tests) was computed individually within each subclass, and p-values were adjusted with Benjamini–Hochberg correction (Layer 1 Lamp5: r = 0.63 p = 4×10$^{-5}$; Layer 1 Sst: r = 0.41 p = 0.046; Layer 2-3 Lamp5: r = -0.59 p = 3×10$^{-4}$; Layer 2-3 Sst: r = -0.44 p = 0.03). Coloured lines show linear fit for each subclass with significant Pearson correlation. *, p < 0.05, **, p < 0.01, ***, p < 0.001.

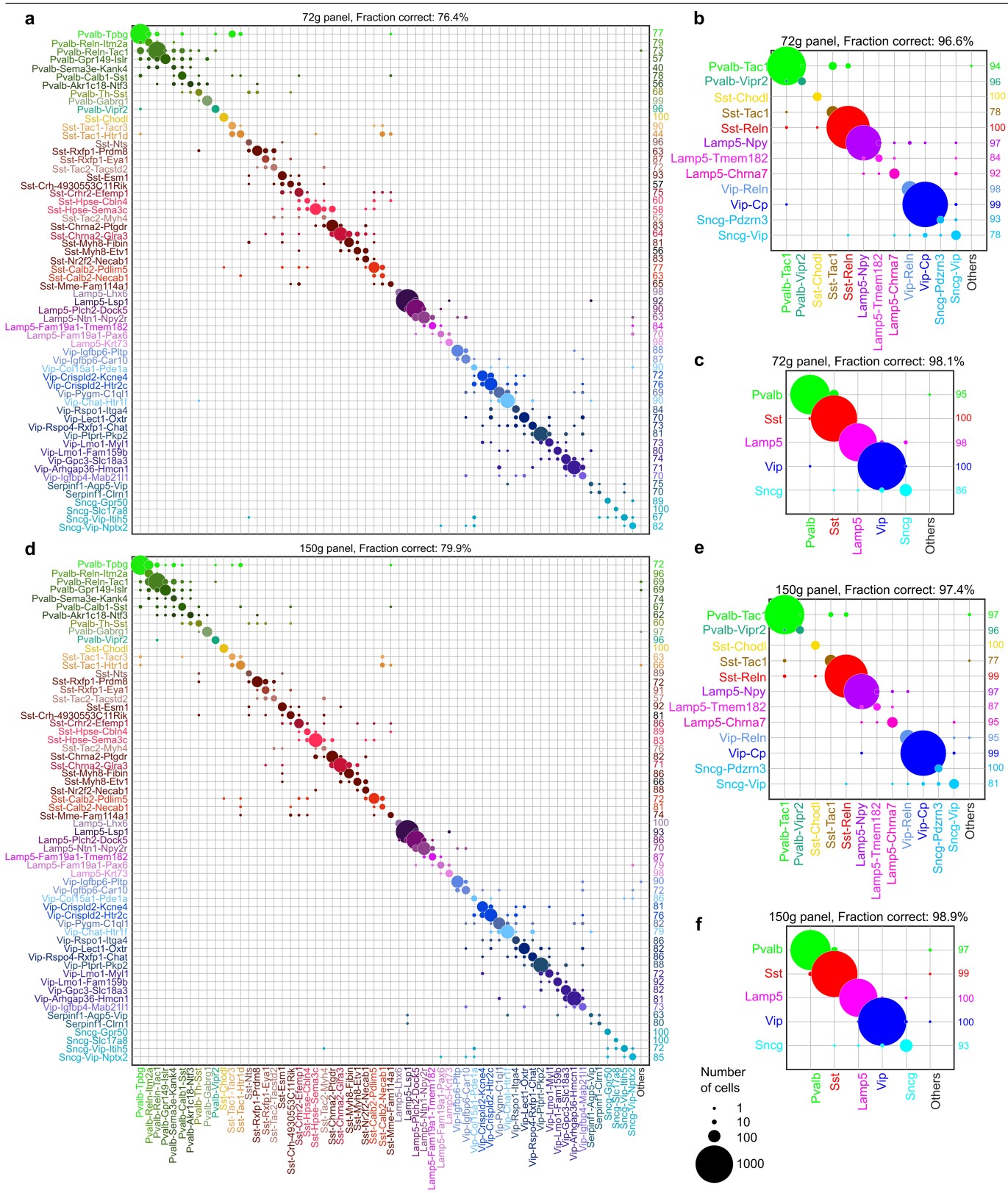

**Extended Data Fig. 10** | See next page for caption.

**Extended Data Fig. 10 | Confusion matrices for cell-type classification on subsampled scRNA-seq data.** To evaluate the accuracy of our cell classification algorithms, we generated simulated ground-truth by random subsampling from scRNA-seq data. A simulated coppaFISH gene count was obtained for each cell in the V1 scRNA-seq data of Ref. [3] (5680 cells from 60 GABAergic clusters) by drawing each gene's expression from a Poisson distribution with mean equal to the scRNA-seq read count, divided by a factor of 100 to account for the relative inefficiency of *in situ* detection. 10-fold cross-validation was used, sequentially using 90% of the cells to compute the mean gene expression per subtype used for classification, and the remaining 10% for evaluation. Cells were classified using the approach taken for coppaFISH data. This procedure was repeated 10 times to estimate the classification accuracy for all 5680 simulated cells. **a**, **b**, **c**, Confusion matrices for classification accuracy at the level of subtypes, types and subclasses using our standard 72-gene panel. Each row shows the results for cells initially classified by Ref. [3] to one subtype, type or subclass, with the size of the circles on that row showing the number of subsampled cells assigned to each subtype, type or subclass using our approach. Subtypes, types and subclasses are assigned correctly with 76.4%, 96.6% and 98.1% accuracy respectively. **d**,**e**,**f**, Using a 150-gene panel (selected by the ProMMT algorithm[10], same panel used to generate the UMAP of Extended Data Fig. 3) increases performance by only 3.5% over the 72-gene panel for subtype assignment and gave very similar performance for type and subclass. Using a yet larger panel of 6000 genes leads to worse performance than the 72-gene panel, owing to overfitting (not shown; 76.8%, 93.6% and 95.4% accuracy for subtype, type and subclass assignment respectively).

# Reporting Summary

## Statistics

For all statistical analyses, confirm that the following items are present in the figure legend, table legend, main text, or Methods section.

| n/a | Confirmed | |
|---|---|---|
| ☐ | ☒ | The exact sample size (*n*) for each experimental group/condition, given as a discrete number and unit of measurement |
| ☐ | ☒ | A statement on whether measurements were taken from distinct samples or whether the same sample was measured repeatedly |
| ☐ | ☒ | The statistical test(s) used AND whether they are one- or two-sided<br>*Only common tests should be described solely by name; describe more complex techniques in the Methods section.* |
| ☐ | ☒ | A description of all covariates tested |
| ☐ | ☒ | A description of any assumptions or corrections, such as tests of normality and adjustment for multiple comparisons |
| ☐ | ☒ | A full description of the statistical parameters including central tendency (e.g. means) or other basic estimates (e.g. regression coefficient) AND variation (e.g. standard deviation) or associated estimates of uncertainty (e.g. confidence intervals) |
| ☐ | ☒ | For null hypothesis testing, the test statistic (e.g. $F$, $t$, $r$) with confidence intervals, effect sizes, degrees of freedom and $P$ value noted<br>*Give P values as exact values whenever suitable.* |
| ☒ | ☐ | For Bayesian analysis, information on the choice of priors and Markov chain Monte Carlo settings |
| ☐ | ☒ | For hierarchical and complex designs, identification of the appropriate level for tests and full reporting of outcomes |
| ☐ | ☒ | Estimates of effect sizes (e.g. Cohen's *d*, Pearson's *r*), indicating how they were calculated |

*Our web collection on statistics for biologists contains articles on many of the points above.*

## Software and code

Policy information about availability of computer code

| | |
|---|---|
| Data collection | NIS-Elements (v5.20.02, build 1453, Nikon), ScanImage v4.2 (written in Matlab), Kilroy toolbox for Matlab (01/09/2019 version, https://github.com/ZhuangLab/storm-control; edits available at https://github.com/acycliq/storm-control) |
| Data analysis | The custom code written in Matlab (R2019b) to analyse and plot the processed data will be available upon reasonable request to the authors.<br>Suite2P toolbox for Matlab (01/05/2018 version, https://github.com/cortex-lab/Suite2P)<br>In situ data analysis code can be found at (16/07/2019 version, https://github.com/jduffield65/iss)<br>Registration code can be found at https://github.com/ha-ha-ha-han/NeuromicsCellDetection/, edits available at https://github.com/sbugeon/NeuromicsCellDetection, version 26/11/2021)<br>Facemap toolbox for Matlab (18/11/2018, https://github.com/MouseLand/facemap)<br>Intensify3D toolbox for Matlab (version 20/07/2020, https://github.com/nadavyayon/Intensify3D/blob/master/User_GUI_Intensify3D.m) |

For manuscripts utilizing custom algorithms or software that are central to the research but not yet described in published literature, software must be made available to editors and reviewers. We strongly encourage code deposition in a community repository (e.g. GitHub). See the Nature Portfolio guidelines for submitting code & software for further information.

## Data

Policy information about availability of data

All manuscripts must include a data availability statement. This statement should provide the following information, where applicable:

- Accession codes, unique identifiers, or web links for publicly available datasets
- A description of any restrictions on data availability
- For clinical datasets or third party data, please ensure that the statement adheres to our policy

Processed data (cellular calcium traces, gene detections, etc) are available at https://doi.org/10.6084/m9.figshare.19448531.v1.
The raw data (2-photon movies, transcriptomic images etc.) will be made available upon reasonable request.
Natural scenes were obtained from the ImageNet database (https://www.image-net.org/).

# Field-specific reporting

Please select the one below that is the best fit for your research. If you are not sure, read the appropriate sections before making your selection.

☒ Life sciences  ☐ Behavioural & social sciences  ☐ Ecological, evolutionary & environmental sciences

For a reference copy of the document with all sections, see nature.com/documents/nr-reporting-summary-flat.pdf

# Life sciences study design

All studies must disclose on these points even when the disclosure is negative.

| | |
|---|---|
| Sample size | We did not use statistical methods to pre-determine sample sizes. However, our sample sizes are similar to those reported in previous publications using a similar approach (Xu et al., Science, 2020; Condylis et al., Science, 2022). |
| Data exclusions | Eight cells were excluded as they were assigned to a Subtype with less than 3 cells in total to avoid statistical analyses comparing groups with less than 3 observations. This exclusion criteria was not pre-established. |
| Replication | All experiments were performed on 4 independent animals, the results were reliably replicated across all animals. |
| Randomization | Our study did not contain experimental groups so randomization does not apply. |
| Blinding | Our study did not contain experimental groups so blinding does not apply. |

# Reporting for specific materials, systems and methods

We require information from authors about some types of materials, experimental systems and methods used in many studies. Here, indicate whether each material, system or method listed is relevant to your study. If you are not sure if a list item applies to your research, read the appropriate section before selecting a response.

### Materials & experimental systems

| n/a | Involved in the study |
|---|---|
| ☒ | ☐ Antibodies |
| ☒ | ☐ Eukaryotic cell lines |
| ☒ | ☐ Palaeontology and archaeology |
| ☐ | ☒ Animals and other organisms |
| ☒ | ☐ Human research participants |
| ☒ | ☐ Clinical data |
| ☒ | ☐ Dual use research of concern |

### Methods

| n/a | Involved in the study |
|---|---|
| ☒ | ☐ ChIP-seq |
| ☒ | ☐ Flow cytometry |
| ☒ | ☐ MRI-based neuroimaging |

## Animals and other organisms

Policy information about studies involving animals; ARRIVE guidelines recommended for reporting animal research

| | |
|---|---|
| Laboratory animals | Mice (Mus musculus) from 3 transgenic mouse lines: Gad2-T2a-NLS-mCherry (2 males and 2 females), Pvalb<tm1(cre)Arbr> (1 male), Sst<tm2.1(cre)Zjh> (2 males and 1 female), were used in this study. All mice were maintained on a C57BL/6 genetic background. Mice were aged between 12 and 15 weeks. |
| Wild animals | No wild animals involved. |

| Field-collected samples | No samples collected from field. |
| --- | --- |
| Ethics oversight | All experimental procedures were conducted in accordance with the UK Animals (Scientific Procedures Act) 1986. Experiments were performed at University College London under personal and project licences released by the Home Office following appropriate ethics review. |

Note that full information on the approval of the study protocol must also be provided in the manuscript.

