## [Peer Review File · Nature]

Manuscript Title: A transcriptomic axis predicts state modulation of cortical interneurons

Reviewer Comments & Author Rebuttals

Reviewer Reports on the Initial Version:

Referees' comments:

Referee #1 (Remarks to the Author):

This manuscript by Bugeon and colleagues explores how changes in brain state influence the activity of different types of GABAergic interneurons in superficial layers of the primary visual cortex of the mouse. To this end, the authors combine in vivo 2-photon calcium imaging in head-fixed mice with high-resolution in situ transcriptomics ex vivo, with the subsequent registration of the two imaging datasets. This approach provides the most comprehensive dataset to date of the in vivo activity of a large diversity of cortical interneurons, and as such, this is a very important resource that will be of great value to the community. The analysis of this dataset led the authors to propose that the modulation of the activity of interneurons by brain states correlates with their transcriptional profile, as identified by the first principal component of gene expression. This general principle may have limited practical value, but it represents a useful paradigm to describe the interaction between specific patterns of brain activity and neuronal diversity.

I only have a few comments and suggestions, which may contribute to improving the manuscript by clarifying some aspects of the study. I think this might be important to make the manuscript more accessible to a larger audience.

1. Why did the authors use 72 genes? What is the minimum number required to distinguish among all 35 t-types found in layers 1-3?
2. Line 90: "the imaged cells were assigned to just 35 of the 60 total t-types defined by the original scRNA-seq study". Were the 35 t-types preselected based on their laminar location in Tasic et al, or were all 60 t-types used as the reference when using pciSeq?
3. Line 133: "The hierarchical permutation test revealed significant differences in state modulation at all three levels: Family, Class, and t-type". It's unclear what the authors refer to as "significant differences" here. Significant differences in the way that families, classes, and t-types are modulated by brain state? Most of the description that follows this statement corresponds to interneuron classes and t-types. What's the point of looking at the behaviour of interneuron families when the authors can achieve this level of granularity? Shouldn't be the point of this analysis to highlight the fact that lumping all PV (or SST or VIP...) cells together (as done in many in vivo studies over the past few years) does not make sense from a functional perspective?
4. Line 149: "The dependence of state modulation on t-type". It's unclear what the authors meant here. Do they mean "the modulation of individual t-types by the brain state..."? Please rephrase.

5. It's been suggested that some of the 60 t-types previously described in the adult mouse cortex may correspond to the same cell type in different brain states. Does the dataset described here support that claim? This would be a useful addition to the current analyses.

6. Line 165: "Most inhibitory cell types contained neurons responding to grating stimuli. Do the authors mean "most inhibitory families contained neurons..."? Otherwise, I do not understand this sentence. More generally, the results on the responses to visual stimuli are surprising. I would have assumed that both the modulation by brain state and the response to sensory stimuli is largely determined by the intrinsic properties of interneurons and their specific connectivity. Why would then the sensory stimuli elicit so little differences (relative to the brain state modulation) in different classes/t-types?

Minor

1. Line 107: "We next asked to what extent fine transcriptomic types affect a neuron's in vivo activity patterns". It's unclear what the authors meant to say here, please rephrase.

2. The term "genetic" as in "a genetic axis" is perhaps too general and overarching to be used in this context. I suggest that the authors describe this as "a transcriptomic axis" instead.

3. A comment about the nomenclature used throughout the paper. The five major Families remind me of the New York mafia... Seriously, the use of families/classes/t-types is yet another variation of the always confusing nomenclature employed to describe cortical interneurons. Considering that they are organised in a hierarchical structure, perhaps using a nomenclature that more evidently reflect that would be useful (class/subclass/type/subtype), as proposed for example by Zeng and Sanes. In that nomenclature, interneurons would be a class of cortical neurons, the five families would be subclasses of interneurons, morphologically defined interneurons would be types, and t-types would be subtypes. Just a suggestion.

Referee #2 (Remarks to the Author):

This manuscript presents a powerful new approach to combine in vivo functional imaging with post-hoc cell type decoding. The authors developed the coppaFISH method that allows cellular resolution detection of the expression of dozens of genes simultaneously from thin sections, which they can use to decode fine-grained cell type identities by mapping to scRNA-seq data. They used 2P Ca imaging to record from large populations of neurons in mouse primary visual cortex, and then applied coppaFISH to the imaged tissue to localize mRNAs for 72 genes chosen to identify fine inhibitory t-types in layers 1-3. To achieve high-precision in vivo – ex vivo correspondence for each imaged cell, they performed a series of image alignment tasks on multiple imaging volumes. This approach is among the first few reported in the field, and the most comprehensive and rigorous one I have seen so far. Given the vast amount of cell type information that has been gained from single-cell transcriptomics, the approach reported here will be of tremendous value to the neuroscience field to begin to relate detailed gene expression and cell type information of individual cells with

their in vivo activities and function.

The study allowed simultaneous imaging and direct comparison, for the first time, of nearly all the inhibitory cell types in L1-3 for their visual response properties in different brain states, including types that were not identified in vivo before (e.g., Sncg). The authors assigned the cells to a three-level hierarchy of 5 Families, 11 Classes, and 35 t-types. They found that visual responses differed significantly only across Families, but modulation by brain state differed at all three hierarchical levels. They identified a single transcriptomic principal component which could predict the brain state modulation of different cell types. These findings are of great interest as a significant addition of the rich literature on visual cortical physiology and begin to reveal the deeper complexity of the roles the diverse inhibitory cell types may play.

Major points:

1. The authors used Tasic et al (2018) transcriptomic taxonomy as the starting point and grouped 35 t-types (with known substantial continuous variations among them) into intermediate-level classes based on their own UMAP analysis, and compared with Gouwens et al (2020) Patch-seq study which also derived intermediate-level triple-modality MET types. There are more recent transcriptomic studies that also attempted to discretize the continuity by defining intermediate-level supertypes or groupings, including Yao et al (Cell 2021) isocortex-hippocampus study and BICCN (Nature 2021) multimodal primary motor cortex study. It would be important to have a more comprehensive comparison between the current study and these other studies to see how the classes defined here relate to the other definitions and present a consistent view about the intermediate inhibitory cell types, in cortical L1-3, and their associated markers, which will be very useful for the field.
2. Related to the above point, for Extended Data Fig 3, it will be really helpful to actually label all relevant t-types on the UMAP itself. Currently with the crowded glyphs and similar colors, it is nearly impossible to tell which t-types are where, so the point of continua and the rationale of class grouping can't be clearly understood.
3. The differential state modulation of different cell classes/types is intriguing. It is somewhat surprising and disappointing to see that it could be explained by a single genetic PC, and not entirely convincing to me. Given that this gPC1 is strongly correlated with the well-known major difference of excitability and firing pattern of the interneuron classes, e.g., fast spiking for Pvalb cells, regular spiking for Sst cells and delayed spiking for neurogliaform cells, it could have a dominant effect. Could the relationship between state modulation and gPC1 be just a simple correlation? Similar question goes to the correlation with the expression of cholinergic receptors, which is rather weak. More compelling justification to explain the specific aspects of the state modulation is needed.

Minor points:

1. To make cell type labels consistent, for all figures (e.g., Fig 1, 3, 4, EDFig 8), change all capital labels of PVALB, SST, LAMP5, SNCG and VIP to Pvalb, Sst, Lamp5, Sncg and Vip. Also, according to the initial definition of classes (lines 78-82), there should be a dash between gene symbols, e.g., Pvalb-Tac1, not PVALB Tac1.
2. Fig 1k, please clarify what the scaled cortical depth on each axis corresponds to. What micrometer of depth, or fraction of the total depth, or something else, does 1 correspond to?

3. Fig 2a, this may be obvious but please still point out that this shows an example session.
4. Fig 3b legend says that “Tuning curves were averaged over odd trials, shifted and normalized according to the preferred direction on even trials, and averaged across cells of the same t-type.” So for the direction tuning plot please clarify if preferred direction is always at 0. For some cell types it doesn’t seem to be the case, why?
5. Refs 45 and 54 have now been published.
6. Line 553, “This means that fewer padlock probes were used for genes with low expression and vice versa” seems to be wrong. It should be fewer padlock probes for genes with high expression.
7. Extended Data Fig 1a legend, retro-transcribed should be reverse-transcribed.
8. Extended Data Fig 5b, please label what X-axis stands for, and what the numbers in the plots stand for.
9. Extended Data Fig 6a-b, provide cell type labels as in Fig 2b.

Referee #3 (Remarks to the Author):

This study by Bugeon et al examines the role of fine transcriptional interneuron types and their higher order groupings in the visual cortex. The study is, for the most part, technically impressive, and the findings are a much needed contribution for better insight into the functions of these cell types predicted from scRNA-Seq. Specifically, Bugeon et al show different responses of transcriptomic types to brain states related to arousal and locomotion. For visual processing they mostly show that differences in fine transcriptomic types are not evident. Instead, interneurons show differences at the level of broadly defined families. Notably these family definitions largely correspond to a decade of prior work on cortical interneuron classification and function. This has two implications. First, it provides some validation of their methodology. Second, classifying the cells at the family level has been done a variety of ways in the past and it is not clear from the paper how the added gene expression information improves understanding of sensory response over past single gene transgenic or IHC results (by emphasizing the similarity of results, the authors mostly indicate that it doesn’t). A main finding that the authors put forward is the genetic principal component (gPC), which correlates nicely with the property differences that are observed in arousal state and visual processing. Some of the conclusions are overly speculative but are presented as empirical facts instead of as untested hypotheses, thus they need to be better qualified and concurrently de-emphasized as primary findings of the study (see below). Overall, the authors should better highlight the concrete differences of their observations with this powerful method relative to extensive prior work by the field.

1. The method for retrospective FISH after calcium imaging in vivo. This is an incremental advance over past methods (previously called either CaRMA imaging or MultiMap imaging) with the advance primarily being based on the number of genes that have been examined (72 genes) which is considerably greater than the 12 genes in Xu et al., the 6 genes in Lovett-Baron, and the 6 genes in Chesler/Ryba. Note that Chesler/Ryba also demonstrated this method (neuron 2020), and they should be cited by the authors. Because of this prior work, the main advance of incorporating a greater number of genes into the CaRMA/multimap method should be mentioned.
2. Biologically, the advance is to look at fine transcriptional types and their grouping into classes and

families in the context of one another. There are some confirmatory results here as well as some new observations. This is definitely valuable and unique data. The results are important but also a little disappointing because there doesn't appear to be much detectable significance to the fine interneuronal transcriptomic types in visual processing. This is surprising because there do appear to be differences in responsiveness at the level of internal states. I think many readers will be left wondering if there are tasks or stimuli that would potentially draw out a more specific role for interneuron types.

3. Cell type assignment. There appear to be substantial undiscussed technical issues with the gene set, the gene detection method, and characterization of the reliability with which these types (defined from the Allen Institute (Tasic et al)) can be predicted by the gene set selected by Bugeon et al. The genes used for cell type assignment in this paper often do not correspond to the cell type designators from Tasic et al. This is confusing and never really discussed. Moreover, analysis of the reliability with which the chosen gene set predicts the scRNA-seq clusters is also not presented in this paper. The authors do show a supplemental file reporting that typically less than 50% of interneurons can be assigned to a cell class. This is surprising given their large number of genes. Does this reflect RNA degradation or a methodological limitation of padlock probes, which are not efficient for gene detection, or is it a limitation of the gene set? This raises questions about the robustness of the methodological advance associated with the larger number of genes.

4. Similarity of responses within a type. One of the hoped for advantages of transcriptomic types is that they will correspond to groups of cells that respond similarly, as implied on line 32-34 of the manuscript. However, there is not very clear quantification of this. There is some information about proportions excited or inhibited, but it is hard to keep track of. The authors should consider an approach outlined in ref. 40, which defined a metric called purity that could be used here. Regardless, there should be much more extensive documentation of the similarity of the responses within transcriptomic types under different conditions (states, stimuli, etc). For example, Fig. 2a could be reproduced as the purity of the individual types/groups/families while maintaining the temporal axis of the figure. Other purity temporal averages should also be provided.

5. Genetic PCA. The authors use this dimensionality reduction approach as a convenient descriptor of the gene expression continuum of interneuron cell types. This is an appropriate choice, but it seems to primarily reflect the already established family differences that are widely used in the literature (e.g., PV, SST, VIP). This is somewhat difficult to fully evaluate because not all the family markers used to describe the cell types were actually used in the gene set (e.g., *Sncg* is missing). Thus, the emphasis on this analysis is appropriate but the level of surprise communicated in the manuscript (line 263) at its usefulness without putting it in the context of known interneuron family differences seems unwarranted (unless the authors can explain other reasons to be surprised by this). It is also odd that the authors emphasize the weighting of *Slc6a1* and *Gad1* for the negative PC weights instead of *Pvalb*, which is the most negative weight. The authors also introduce unsupported claims that *Slc6a1* is reflective of elevated metabolic activity and *Gad1* reflects more GABA release. Possibly, but the only citation (13) is to a paper by some of the authors on this study that just proposes this interesting hypothesis but does not test it. This speculation appears to have the aim of trying to provide some cell biological meaning to their data, but it is not sufficiently well supported to be considered a conclusion of this paper as opposed to an interesting observation to be tested later.

6. The role of cholinergic signaling. The authors get more speculative by introducing a correlation with the expression level of excitatory or inhibitory cholinergic or nicotinic receptors. This data is

derived from level of expression in scRNA-Seq (Tasic et al), and it is a relationship that was never empirically determined by experiments from the neurons recorded in vivo by Bugeon et al. This is not clearly stated in the main text, and I was surprised to discover the lack of in vivo testing of these transcripts while I was trying to understand the discrepancy of the gene sets from the transcript-type naming convention. The role of cholinergic modulation is an attractive hypothesis, but the correlation coefficients are low, and the authors should have included these genes in their gene set to establish if these low correlation coefficients are weaker or stronger when measured from the actual neurons recorded in vivo. It is important to note that these neuromodulatory receptors are not necessarily expressed in every cell of a particular type and thus it needs to be shown directly that their expression level predicts the neuronal response type of the interneurons. It also should be kept in mind that, if authors are going to go back to the scRNA-Seq data without pre-planning the hypotheses that they will test, then all sorts of correlations might be observable with different gene sets. Finally, these findings appear to be similar to the tested hypothesis in reference 40 that the neuromodulatory gene *Npy1r* is expressed across multiple hypothalamic cell types and is the most predictive gene for neuronal response type. The conceptual similarity between their analysis and this past work should be noted. This theme is also discussed in ref. 39. In summary, the data that Bugeon et al present about cholinergic/nicotinic receptors is an attractive hypothesis, but it is not a robust empirical result that deserves the level of emphasis that it is given in the paper.

7. Use of the word 'genetic'. The authors sometimes refer to gene expression and transcripts as being 'genetic'. Genetic is related to the genome, which is not a source of differences here. 'Transcriptomic' or 'gene expression' are more appropriate terms.

8. There is no mention of removing mcherry and GCaMP fluorophores for the retrospective FISH analysis? Wouldn't these components interfere?

9. Some of the figures are small and virtually unreadable without blowing them up on a computer monitor

10. Why are there so many fewer pyramidal cells than interneurons in Fig. 2a? Pyramidal cells should outnumber interneurons by roughly 5:1. Was there some sort of selection process for pyramidal cells? Also, the ordering of which cells are negative PC used for analysis should be noted on the figure.

Author Rebuttals to Initial Comments:

Reviewer 1

This manuscript by Bugeon and colleagues explores how changes in brain state influence the activity of different types of GABAergic interneurons in superficial layers of the primary visual cortex of the mouse. To this end, the authors combine in vivo 2-photon calcium imaging in head-fixed mice with high-resolution in situ transcriptomics ex vivo, with the subsequent registration of the two imaging datasets. This approach provides the most comprehensive dataset to date of the in vivo activity of a large diversity of cortical interneurons, and as such, this is a very important resource that will be of great value to the community. The analysis of this dataset led the authors to propose that the modulation of the activity of interneurons by brain states correlates with their transcriptional profile, as identified by the first principal component of gene expression. This general principle may have limited practical value, but it represents a useful paradigm to describe the interaction between specific patterns of brain activity and neuronal diversity.

I only have a few comments and suggestions, which may contribute to improving the manuscript by clarifying some aspects of the study. I think this might be important to make the manuscript more accessible to a larger audience.

We thank the reviewer for these positive comments, and address specific suggestions below.

1. Why did the authors use 72 genes? What is the minimum number required to distinguish among all 35 t-types found in layers 1-3?

Thank you for this question. As we now state in the text (lines 568-573), this panel is a subset of a panel of 99 genes described in a previous study. The original panel of 99 genes in that other study was picked based on scRNA-seq data, using an algorithm that predicts which gene combinations are required to identify fine transcriptomic subtypes. We then used a retrospective analysis, using in situ transcriptomics to analyze the contribution each gene made to classification accuracy. This revealed that many genes were serving no purpose in accurately classifying cells (figure S16 of Qian et al), leading to their removal from the panel.

We have added a new analysis to evaluate the ability of this 72-gene panel to identify V1 inhibitory neurons subtypes (Extended data Figure 10, lines 850-854). We simulated ground truth data by randomly subsampling scRNAseq data from known classes to match the lower efficiency of our in situ method. We then classified these simulated calls using different gene panels, to create a “confusion matrix” estimating misclassification error rates of this panel, in comparison to hypothetical panels of larger size. This method estimates that the 72-gene panel identifies Family, Type, and Subtype correctly with 98%, 96.4%, and 76.3% accuracy. Increasing the simulated panel size further to 150 genes yielded only slightly better accuracy (99%, 97.3%, and 80.4%) and using all 6000 genes that expressed in at least one cell type actually decreased accuracy (95.4%, 93.6%, and 76.8%) as expected due to overfitting of stochastic expression. We conclude that 72 genes is sufficient to identify inhibitory classes with near-optimal accuracy.

2. Line 90: “the imaged cells were assigned to just 35 of the 60 total t-types defined by the original scRNA-seq study”. Were the 35 t-types preselected based on their laminar location in Tasic et al, or were all 60 t-types used as the reference when using pciSeq?

We used all 109 V1 transcriptomic clusters defined by Tasic et al as reference (including inhibitory cells of all layers as well as non-GABAergic cells), and each cell was assigned to whichever of these 109 transcriptomic clusters matched best, without using anatomical information. Nevertheless, the imaged cells, which are all in superficial layers, were found to transcriptomically match only 35 of these t-types, corresponding to superficial inhibitory classes.

We have clarified this in the text (lines 88-90, 847-850).

3. Line 133: “The hierarchical permutation test revealed significant differences in state modulation at all three levels: Family, Class, and t-type”. It’s unclear what the authors refer to as “significant differences” here. Significant differences in the way that families, classes, and t-types are modulated by brain state? Most of the description that follows this statement corresponds to interneurons classes and t-types. What’s the point of looking at the behaviour of interneuron families when the authors can achieve this level of granularity? Shouldn’t be the point of this analysis to highlight the fact that lumping all PV (or SST or VIP...) cells together (as done in many in vivo studies over the past few years) does not make sense from a functional perspective?

Thank you, we have entirely rewritten this section, including a new analysis showing that correlations are larger within Subtypes than between them [Fig. 2b, Lines 106-119]. In brief, our analysis allows us to assess at which level of the transcriptomic hierarchy different functional properties differ. It shows that lumping cells into Families does not make sense when considering correlations and state modulation, but is not too problematic when considering visual sensory responses.

4. Line 149: “The dependence of state modulation on t-type”. It’s unclear what the authors meant here. Do they mean “the modulation of individual t-types by the brain state...”? Please rephrase.

Thanks, we have rewritten this whole section (Lines 137-155).]

5. It’s been suggested that some of the 60 t-types previously described in the adult mouse cortex may correspond to the same cell type in different brain states. Does the dataset described here support that claim? This would be a useful addition to the current analyses.

We believe the reviewer is referring, for example, to papers that show gene expression in inhibitory cells can change with activity and learning (e.g. Donato et al *Nature* 2013; Dehorter et al *Science* 2015). We have previously conjectured that this might relate to the tPC1 continuum, i.e. that cells could move along this continuum over long enough periods of time (Harris et al, *PLOS Biology* 2018). Nevertheless, our current data provide no

evidence for or against this hypothesis, and we would prefer not to speculate further here without solid evidence.

6. Line 165: “Most inhibitory cell types contained neurons responding to grating stimuli. Do the authors mean “most inhibitory families contained neurons...”? Otherwise, I do not understand this sentence.

Thank you, yes. We have reworded this (Line 172).

More generally, the results on the responses to visual stimuli are surprising. I would have assumed that both the modulation by brain state and the response to sensory stimuli is largely determined by the intrinsic properties of interneurons and their specific connectivity. Why would then the sensory stimuli elicit so little differences (relative to the brain state modulation) in different classes/t-types?

Our present data can't provide a definitive answer to this, and we also cannot rule out that some as-yet untried stimulus class might reveal a difference between subtypes. However, we can formulate a hypothesis: that a neuron's sensory tuning primarily reflects the pattern and tuning of its excitatory and inhibitory input synapses; but that differences in cholinergic receptors between subtypes lead to different state modulation. We now mention this possibility in Discussion (LINES 300-311).

Minor

1. Line 107: “We next asked to what extent fine transcriptomic types affect a neuron's in vivo activity patterns”. It's unclear what the authors meant to say here, please rephrase.

Thank you, we have rewritten this section (Line 106).

2. The term “genetic” as in “a genetic axis” is perhaps too general and overarching to be used in this context. I suggest that the authors describe this as “a transcriptomic axis” instead.

Thank you, this is a good suggestion. We have made the replacement throughout and also replaced “gPC1” by “tPC1”.

3. A comment about the nomenclature used throughout the paper. The five major Families remind me of the New York mafia... Seriously, the use of families/classes/t-types is yet another variation of the always confusing nomenclature employed to describe cortical

interneurons. Considering that they are organised in a hierarchical structure, perhaps using a nomenclature that more evidently reflect that would be useful (class/subclass/type/subtype), as proposed for example by Zeng and Sanes. In that nomenclature, interneurons would be a class of cortical neurons, the five families would be subclasses of interneurons, morphologically defined interneurons would be types, and t-types would be subtypes. Just a suggestion. <https://www.nature.com/articles/nrn.2017.85>

This is a good suggestion and we have taken it on board almost entirely: we now call the second level of the hierarchy Type, and the third level Subtype. As to the top level, this was a difficult decision. If we were to use the Zeng and Sanes convention, the top level of the hierarchy would be called Subclass. Unfortunately, using “Subclass” to describe the top level of a hierarchy would surely confuse our readers. Furthermore, other papers use other conventions, such as Scala et al (*Nature*, 2020) who use Family to describe this level. We have thus retained the name Family for the top level, and switched to the Zeng/Sanes nomenclature for the other two levels.

Reviewer 2

This manuscript presents a powerful new approach to combine in vivo functional imaging with post-hoc cell type decoding. The authors developed the coppaFISH method that allows cellular resolution detection of the expression of dozens of genes simultaneously from thin sections, which they can use to decode fine-grained cell type identities by mapping to scRNA-seq data. They used 2P Ca imaging to record from large populations of neurons in mouse primary visual cortex, and then applied coppaFISH to the imaged tissue to localize mRNAs for 72 genes chosen to identify fine inhibitory t-types in layers 1-3. To achieve high-precision in vivo – ex vivo correspondence for each imaged cell, they performed a series of image alignment tasks on multiple imaging volumes. This approach is among the first few reported in the field, and the most comprehensive and rigorous one I have seen so far. Given the vast amount of cell type information that has been gained from single-cell transcriptomics, the approach reported here will be of tremendous value to the neuroscience field to begin to relate detailed gene expression and cell type information of individual cells with their in vivo activities and function.

The study allowed simultaneous imaging and direct comparison, for the first time, of nearly all the inhibitory cell types in L1-3 for their visual response properties in different brain states, including types that were not identified in vivo before (e.g., Sncg). The authors assigned the cells to a three-level hierarchy of 5 Families, 11 Classes, and 35 t-types. They found that visual responses differed significantly only across Families, but modulation by brain state differed at all three hierarchical levels. They identified a single transcriptomic principal component which could predict the brain state modulation of different cell types. These findings are of great interest as a significant addition of the rich literature on visual cortical physiology and begin to reveal the deeper complexity of the roles the diverse inhibitory cell types may play.

Thank you for this positive evaluation of our work.

Major points:

1. The authors used Tasic et al (2018) transcriptomic taxonomy as the starting point and grouped 35 t-types (with known substantial continuous variations among them) into intermediate-level classes based on their own UMAP analysis, and compared with Gouwens et al (2020) Patch-seq study which also derived intermediate-level triple-modality MET types. There are more recent transcriptomic studies that also attempted to discretize the continuity by defining intermediate-level supertypes or groupings, including Yao et al (*Cell* 2021) isocortex-hippocampus study and BICCN (*Nature* 2021) multimodal primary motor cortex study. It would be important to have a more comprehensive comparison between the current study and these other studies to see how the classes defined here relate to the other definitions and present a consistent view about the intermediate inhibitory cell types, in cortical L1-3, and their associated markers, which will be very useful for the field.

This is an important point, about which we were not sufficiently clear. We used the Tasic et al classifications because they were defined specifically for V1 (while the 2018 paper included data from both V1 and ALM, we used only the V1 data here). The BICCN classifications were defined specifically for M1, and the Yao classifications (*Cell* 2021) were defined for all cortical/hippocampal regions together. While inhibitory neurons are indeed similar across regions, it is still possible that subtle distinctions between subtypes exist in specific regions only, and indeed these classifications are not identical (Figure S1 of Yao et al *Cell* 2021; ED Figure 3 of Yao et al *Nature* 2021). Because our aim was to search for physiological correlates of subtle transcriptomic differences in V1 cells, it was important to use V1-specific definitions. Nevertheless, our results do suggest that the transcriptomic features that matter to both sensory responses and state modulation are features common across cortical areas. We now clarify this and refer the reader to the figures in other papers that identify different classification schemes (Lines 839-843).

2. Related to the above point, for Extended Data Fig 3, it will be really helpful to actually label all relevant t-types on the UMAP itself. Currently with the crowded glyphs and similar colors, it is nearly impossible to tell which t-types are where, so the point of continua and the rationale of class grouping can't be clearly understood.

Thank you, corrected.

3. The differential state modulation of different cell classes/types is intriguing. It is somewhat surprising and disappointing to see that it could be explained by a single genetic PC, and not entirely convincing to me. Given that this gPC1 is strongly correlated with the well-known major difference of excitability and firing pattern of the interneuron

classes, e.g., fast spiking for Pvalb cells, regular spiking for Sst cells and delayed spiking for neurogliaform cells, it could have a dominant effect. Could the relationship between state modulation and gPC1 be just a simple correlation? Similar question goes to the correlation with the expression of cholinergic receptors, which is rather weak. More compelling justification to explain the specific aspects of the state modulation is needed.

Thank you for this important point. We did not mean to say that all aspects of state modulation can be explained by the tPC1 axis. Rather, tPC1 provides a simple organizing principle that explains a substantial amount of state-related variance, but additional factors are likely to contribute further subtle differences between cell types. We have done a new analysis, which shows that tPC1 explains 68% of the variance in state modulation explainable by Subtype (Extended Data Figure 8b; Lines 219-220).

Regarding cholinergic receptors and cellular physiology, we agree that both could contribute, and that our data are correlational, and do not prove causation. We have carefully reframed our statements to make clear that these are hypotheses suggested by our correlational data, but that causality has not been established (Lines 309-311).

Minor points:

1. To make cell type labels consistent, for all figures (e.g., Fig 1, 3, 4, EDFig 8), change all capital labels of PVALB, SST, LAMP5, SNCG and VIP to Pvalb, Sst, Lamp5, Sncg and Vip. Also, according to the initial definition of classes (lines 78-82), there should be a dash between gene symbols, e.g., Pvalb-Tac1, not PVALB Tac1.

Thank you, corrected.

2. Fig 1k, please clarify what the scaled cortical depth on each axis corresponds to. What micrometer of depth, or fraction of the total depth, or something else, does 1 correspond to?

Thank you, corrected.

3. Fig 2a, this may be obvious but please still point out that this shows an example session.

Thank you, corrected.

4. Fig 3b legend says that "Tuning curves were averaged over odd trials, shifted and normalized according to the preferred direction on even trials, and averaged across cells

of the same t-type.” So for the direction tuning plot please clarify if preferred direction is always at 0. For some cell types it doesn't seem to be the case, why?

The reason the preferred direction might not be 0 in these cross-validated analyses is that for weakly-tuned cells, noise might mean firing was randomly strongest at different directions in the training and test sets. We have clarified this in the text [Lines 1075-77 and figure 3 legend].

5. Refs 45 and 54 have now been published.

Thanks, fixed.

6. Line 553, “This means that fewer padlock probes were used for genes with low expression and vice versa” seems to be wrong. It should be fewer padlock probes for genes with high expression.

Thank you for spotting this. Fixed.

7. Extended Data Fig 1a legend, retro-transcribed should be reverse-transcribed.

Thanks, fixed.

8. Extended Data Fig 5b, please label what X-axis stands for, and what the numbers in the plots stand for.

Thanks, fixed.

9. Extended Data Fig 6a-b, provide cell type labels as in Fig 2b.

Thanks, fixed. We have also changed the formats of all these plots.

Reviewer 3

This study by Bugeon et al examines the role of fine transcriptional interneuron types and their higher order groupings in the visual cortex. The study is, for the most part, technically impressive, and the findings are a much needed contribution for better insight into the

functions of these cell types predicted from scRNA-Seq. Specifically, Bugeon et al show different responses of transcriptomic types to brain states related to arousal and locomotion. For visual processing they mostly show that differences in fine transcriptomic types are not evident. Instead, interneurons show differences at the level of broadly defined families. Notably these family definitions largely correspond to a decade of prior work on cortical interneuron classification and function. This has two implications. First, it provides some validation of their methodology. Second, classifying the cells at the family level has been done a variety of ways in the past and it is not clear from the paper how the added gene expression information improves understanding of sensory response over past single gene transgenic or IHC results (by emphasizing the similarity of results, the authors mostly indicate that it doesn't). A main finding that the authors put forward is the genetic principal component (gPC), which correlates nicely with the property differences that are observed in arousal state and visual processing. Some of the conclusions are overly speculative but are presented as empirical facts instead of as untested hypotheses, thus they need to be better qualified and concurrently de-emphasized as primary findings of the study (see below). Overall, the authors should better highlight the concrete differences of their observations with this powerful method relative to extensive prior work by the field.

Thank you, we have added text highlighting the novel discoveries established by this study. In brief:

- The fact that fine Subtypes have different state modulation could not have been shown with previous methods, which only identified coarse cell families (Lines 46-47).
- The fact that Subtype-level differences in state modulation reflect transcriptomic continua, rather than discrete cell groups, and indeed can correlate with the analog expression levels of individual genes (Figure 2d-f, Lines 156-167, ED Figure 6e, 8a)
- The fact that fine subtypes do not have substantially different visual responses also could not have been shown with previous methods. Indeed, showing that fine subtypes have similar responses requires first distinguishing these subtypes (Lines 169-192).
- Data on Sncg cells of visual cortex had not been previously reported. The fact that they are uniformly suppressed by visual stimuli is a novel, unexpected result (Lines 174-176).
- The fact that the ordering of Subtypes defined by in vivo state modulation matches the ordering defined by in vitro physiology and cholinergic receptor expression could not have been discovered without measuring all these fine Subtypes (Lines 221-247)
- Analysis of correlation between cell types (Figure 4d, ED Figure 8c-e; New analysis in Figure 2b) could not have been performed without simultaneous measurement of multiple cell classes, which is impossible using transgenic strategies.

We agree also that some of our results (such as the relation of state modulation to in vitro spiking patterns and cholinergic receptors) are correlative, and have clarified that a causal role for these factors remains a hypothesis to be tested in future work (Lines 309-311). We have also moved all these relating our data to previous scRNAseq or Patch-seq data to a separate figure (Figure 5), to distinguish them from the experimental results specific to our study (Figures 1-4).

1. The method for retrospective FISH after calcium imaging in vivo. This is an incremental advance over past methods (previously called either CaRMA imaging or MultiMap imaging) with the advance primarily being based on the number of genes that have been examined (72 genes) which is considerably greater than the 12 genes in Xu et al., the 6 genes in Lovett-Baron, and the 6 genes in Chesler/Ryba. Note that Chesler/Ryba also demonstrated this method (neuron 2020), and they should be cited by the authors. Because of this prior work, the main advance of incorporating a greater number of genes into the CaRMA/multimap method should be mentioned.

Thank you, we have added this to the text (Lines 44-47). Also thank you for the Chesler/Ryba citation, which we have added (Line 41).

2. Biologically, the advance is to look at fine transcriptional types and their grouping into classes and families in the context of one another. There are some confirmatory results here as well as some new observations. This is definitely valuable and unique data. The results are important but also a little disappointing because there doesn't appear to be much detectable significance to the fine interneuronal transcriptomic types in visual processing. This is surprising because there do appear to be differences in responsiveness at the level of internal states. I think many readers will be left wondering if there are tasks or stimuli that would potentially draw out a more specific role for interneuron types.

This is a good point, which we have added to discussion (Lines 323-327).

3. Cell type assignment. There appear to be substantial undiscussed technical issues with the gene set, the gene detection method, and characterization of the reliability with which these types (defined from the Allen Institute (Tasic et al)) can be predicted by the gene set selected by Bugeon et al. The genes used for cell type assignment in this paper often do not correspond to the cell type designators from Tasic et al. This is confusing and never really discussed. Moreover, analysis of the reliability with which the chosen gene set predicts the scRNA-seq clusters is also not presented in this paper. The authors do show a supplemental file reporting that typically less than 50% of interneurons can be assigned to a cell class. This is surprising given their large number of genes. Does this reflect RNA degradation or a methodological limitation of padlock probes, which are not efficient for gene detection, or is it a limitation of the gene set? This raises questions about

the robustness of the methodological advance associated with the larger number of genes.

We thank the reviewer for this comment, which indicates that we were not sufficiently clear. As we now clarify in text (Line 847; Extended data Figure 4c), 99% of cells analyzed *in situ* were assigned a cell class; the Bayesian assignment algorithm gives a posterior probability reflecting confidence in this assignment, which was usually very high. (We now show a histogram of these confidence levels as Extended Data Figure 4c). Instead, 50% is the fraction of cells in the *in vivo* imaging volume that could not be confidently identified in the *ex vivo* tissue sections. Failures of *in vivo* to *ex vivo* identification happen for many reasons such as loss or damage to some slices in processing, or tissue slice warping that made volume registration ambiguous. We took a conservative stance to this problem, manually curating all matches and excluding any cells for which the match was not certain (without reference to the *in vivo* physiology). This conservatism resulted in the 50% rate of match acceptance (Lines 939-940)..

Regarding gene set choice: it is now clear from scRNAseq analyses that the transcriptomic signatures of cell types are highly redundant. Because each cell type is characterized by combinatorial expression of many genes, one may remove individual genes from the panel and still be able to identify cells with high confidence. The genes chosen for the names of the clusters used by Tasic et al were arbitrary: they are genes that are found in the cell class, but they are not the only genes that can identify the class, and are often also found in other classes but in different combinations. In previous work (Qian et al *Nature Methods* 2019, figure S16), we showed that classification of fine interneuron subtypes in CA1 could be made accurately with around 50 carefully-chosen genes. We have now added a similar analysis for V1 data (Extended Data Figure 10), using simulated ground truth obtained by subsampling Tasic et al.'s scRNA-seq data. This suggests that using 150 genes instead of 72 would result in only a minor improvement, and that using 6000 would actually result in decreased performance due to overfitting of random expression fluctuations.

4. Similarity of responses within a type. One of the hoped for advantages of transcriptomic types is that they will correspond to groups of cells that respond similarly, as implied on line 32-34 of the manuscript. However, there is not very clear quantification of this. There is some information about proportions excited or inhibited, but it is hard to keep track of. The authors should consider an approach outlined in ref. 40, which defined a metric called purity that could be used here. Regardless, there should be much more extensive documentation of the similarity of the responses within transcriptomic types under different conditions (states, stimuli, etc). For example, Fig. 2a could be reproduced as the purity of the individual types/groups/families while maintaining the temporal axis of the figure. Other purity temporal averages should also be provided.

Thank you for this excellent suggestion. We have done a new analysis showing that neurons of the same transcriptomic Family, Type, or Subtype are more correlated than

neurons belonging to different groupings. This fundamental finding is now presented as the first conclusion of the paper (Figure 2b). To do this analysis, we used a new type of nested permutation test rather than the purity measure, which allows us to implement a conservative statistical test while maintaining our hierarchical classification of cell types. In other words, this test can ask whether even within neurons of the same Type, neurons classified as belonging to the same Subtype are more correlated than neurons belonging to different Subtypes.

5. Genetic PCA. The authors use this dimensionality reduction approach as a convenient descriptor of the gene expression continuum of interneuron cell types. This is an appropriate choice, but it seems to primarily reflect the already established family differences that are widely used in the literature (e.g., PV, SST, VIP). This is somewhat difficult to fully evaluate because not all the family markers used to describe the cell types were actually used in the gene set (e.g., *Sncg* is missing). Thus, the emphasis on this analysis is appropriate but the level of surprise communicated in the manuscript (line 263) at its usefulness without putting it in the context of known interneuron family differences seems unwarranted (unless the authors can explain other reasons to be surprised by this). It is also odd that the authors emphasize the weighting of *Slc6a1* and *Gad1* for the negative PC weights instead of *Pvalb*, which is the most negative weight. The authors also introduce unsupported claims that *Slc6a1* is reflective of elevated metabolic activity and *Gad1* reflects more GABA release. Possibly, but the only citation (13) is to a paper by some of the authors on this study that just proposes this interesting hypothesis but does not test it. This speculation appears to have the aim of trying to provide some cell biological meaning to their data, but it is not sufficiently well supported to be considered a conclusion of this paper as opposed to an interesting observation to be tested later.

This is a very important point, and we were not sufficiently clear. Transcriptomic PCA is *not* just a restatement that there are 5 main Families with different properties. It is a way of ordering these Families, and the fine Subtypes within them, along a one-dimensional continuum. While different Families occupy different mean positions on the continuum defined by tPC1, a cell's Family is not sufficient to determine its place on the continuum. For example, the Subtypes from the *Lamp5* Family are spread all along the continuum. We have added a new figure (Figure 4b) that illustrates this graphically.

The surprising result of this analysis is that multiple cellular properties, which one might have thought would have no systematic relationship (*in vivo* state modulation; *in vitro* physiology; axonal laminar structure; and expression of genes unrelated to family definition such as *Gad1* and cholinergic receptors), all follow this same one-dimensional ordering. This need not have been the case - a priori, each of these different cellular properties might have defined a different ordering of families and subtypes.

The reason we focused on *Gad1* and *Slc6a1* is precisely because they are not canonical Family markers, but genes that all inhibitory neurons express to some degree or other. Indeed, Extended Data Figure 8a directly shows a correlation between state modulation and expression of these two genes, in a manner that does not even use the tPCA weights.

We have clarified these points by adding a new panel (Figure 4b) that graphically shows that the tPC1 axis is not just a function of Family, and by performing additional ANCOVA

analyses that show that tPC1 predicts state modulation even when accounting for a common effect of Family (LINES 215-216).

6. The role of cholinergic signaling. The authors get more speculative by introducing a correlation with the expression level of excitatory or inhibitory cholinergic or nicotinic receptors. This data is derived from level of expression in scRNA-Seq (Tasic et al), and it is a relationship that was never empirically determined by experiments from the neurons recorded in vivo by Bugeon et al. This is not clearly stated in the main text, and I was surprised to discover the lack of in vivo testing of these transcripts while I was trying to understand the discrepancy of the gene sets from the transcript-type naming convention. The role of cholinergic modulation is an attractive hypothesis, but the correlation coefficients are low, and the authors should have included these genes in their gene set to establish if these low correlation coefficients are weaker or stronger when measured from the actual neurons recorded in vivo. It is important to note that these neuromodulatory receptors are not necessarily expressed in every cell of a particular type and thus it needs to be shown directly that their expression level predicts the neuronal response type of the interneurons. It also should be kept in mind that, if authors are going to go back to the scRNA-Seq data without pre-planning the hypotheses that they will test, then all sorts of correlations might be observable with different gene sets. Finally, these findings appear to be similar to the tested hypothesis in reference 40 that the neuromodulatory gene *Npy1r* is expressed across multiple hypothalamic cell types and is the most predictive gene for neuronal response type. The conceptual similarity between their analysis and this past work should be noted. This theme is also discussed in ref. 39. In summary, the data that Bugeon et al present about cholinergic/nicotinic receptors is an attractive hypothesis, but it is not a robust empirical result that deserves the level of emphasis that it is given in the paper.

We agree, and have rephrased this in the discussion accordingly (Lines 300-311, 241-251).

7. Use of the word 'genetic'. The authors sometimes refer to gene expression and transcripts as being 'genetic'. Genetic is related to the genome, which is not a source of differences here. 'Transcriptomic' or 'gene expression' are more appropriate terms.

Agreed. We have changed this and also now use "tPC1" rather than "gPC1".

8. There is no mention of removing mcherry and GCaMP fluorophores for the retrospective FISH analysis? Wouldn't these components interfere?

The intrinsic fluorescence of these molecules fades during tissue processing. We added this to the methods section (Lines 632-633).

9. Some of the figures are small and virtually unreadable without blowing them up on a computer monitor

Thanks, this is a very important point. We have redesigned all figures and hope this has improved things.

10. Why are there so many fewer pyramidal cells than interneurons in Fig. 2a? Pyramidal cells should outnumber interneurons by roughly 5:1. Was there some sort of selection process for pyramidal cells? Also, the ordering of which cells are negative PC used for analysis should be noted on the figure.

We edited the figure legend to explain that around two thirds of our recorded cells are excitatory. (This is lower than the overall fraction of excitatory cells in cortex, likely because we record in layers 1-3, which have a higher fraction of inhibitory cells, and also potentially because cell detection based only on GCaMP activity may miss some sparsely active excitatory neurons). Nevertheless, because our analysis focuses on interneurons, we used a compressed y-axis scale for excitatory neurons in figure 2a, which might have made it appear there were more inhibitory than excitatory neurons. Although the figure had scale bars showing this, they might not have been prominent enough. We have also added an arrow to indicate the direction of sorting of excitatory cells in the raster.

Reviewer Reports on the First Revision:

Referees' comments:

Referee #1 (Remarks to the Author):

The authors have appropriately addressed all my comments and suggestions. I particularly like the sections that they have rewritten; I think they are much clearer now. I have no further comments or suggestions.

Referee #2 (Remarks to the Author):

In the revised manuscript the authors have addressed my concerns satisfactorily. The statistical tests used are appropriate and the results are presented accurately.

I just have one more comment: I agree with Reviewer 1's comment on nomenclature of cell types. I think Class/Subclass/Type/Subtype is appropriate to be used to describe cortical cell types including GABAergic inhibitory neuron types, because Class is the top level of the hierarchy rather than Subclass or Family. Although it is true that Scala et al paper also used the term Family, please note that all the other papers in the same BRAIN Initiative Cell Census Network (BICCN) Nature 2021 publication package, including the flagship BICCN paper itself (aka Callaway et al paper), used the Class (e.g., GABAergic neurons), Subclass (e.g., Pvalb, Sst, Vip, Sncg and Lamp5) and Type (clusters within each subclass) definitions.

Referee #3 (Remarks to the Author):

The author's revisions have addressed my primary comments. The specific advances of the work are clarified and significant.

As a very minor consideration, I would recommend either making figure 2 extend down an entire page or break it into two separate figures to improve readability.